# An integrated resource for functional and structural connectivity of the marmoset brain

Xiaoguang Tian [1] ✉, Yuyan Chen[2], Piotr Majka [3,4], Diego Szczupak[1], Yonatan Sanz Perl [5,6], Cecil Chern-Chyi Yen [7], Chuanjun Tong [2], Furui Feng[2], Haiteng Jiang[8,9], Daniel Glen[10], Gustavo Deco [5,11,12,13], Marcello G. P. Rosa [4] ✉, Afonso C. Silva [1] ✉, Zhifeng Liang [2,14] ✉ & Cirong Liu [2,14,15,16] ✉

Comprehensive integration of structural and functional connectivity data is required to model brain functions accurately. While resources for studying the structural connectivity of non-human primate brains already exist, their integration with functional connectivity data has remained unavailable. Here we present a comprehensive resource that integrates the most extensive awake marmoset resting-state fMRI data available to date (39 marmoset monkeys, 710 runs, 12117 mins) with previously published cellular-level neuronal tracing data (52 marmoset monkeys, 143 injections) and multi-resolution diffusion MRI datasets. The combination of these data allowed us to (1) map the fine-detailed functional brain networks and cortical parcellations, (2) develop a deep-learning-based parcellation generator that preserves the topographical organization of functional connectivity and reflects individual variabilities, and (3) investigate the structural basis underlying functional connectivity by computational modeling. This resource will enable modeling structure-function relationships and facilitate future comparative and translational studies of primate brains.

Mapping brain architecture is critical for decoding brain functions and understanding the mechanisms of brain diseases[1]. Non-human primate (NHP) neuroimaging provides a granular view of the evolution of the brain[2] and could overcome the constraints of human neuroimaging by integration with "ground truth" data from cellular-resolution tracing[3].

As one of the few non-invasive imaging techniques capable of mapping whole-brain functional activity patterns, resting-state fMRI (rs-fMRI) provides insights into large-scale functional architecture[4]. However, data-sharing initiatives of NHP neuroimaging are still at an early stage, with existing open datasets of rs-fMRI data originating in different laboratories and collected for different purposes[5]. This leads to inconsistent imaging protocols and data quality, which hinder analyses across datasets. In addition, most presently available rs-fMRI datasets have been acquired in anesthetized animals, resulting in

difficulties for cross-species studies, particularly relative to awake human brains[6]. The final barrier is the practical difficulty of training large numbers of NHPs to be fully awake during MRI scans[7,8]. A platform for international collaborative research (PRIMatE RESOURCE EXCHANGE) was initiated to address these problems and promote open resource exchange and standards for NHP neuroimaging[5,9].

The common marmoset monkey (Callithrix jacchus) has drawn considerable interest as an NHP species, offering many practical advantages for neuroscience research, including neuroimaging[10–12]. Previous work from our groups has contributed ultra-high-resolution ex vivo diffusion MRI data[13], mesoscale neural tracing data[14], and structural atlases[15–17], which have enabled unprecedented precision in analyses of NHP brain anatomy. However, an essential component for understanding brain architecture has been missing: integrating these

A full list of affiliations appears at the end of the paper. ✉e-mail: txgxp88@gmail.com; marcello.rosa@monash.edu; afonso@pitt.edu; zliang@ion.ac.cn; crliu@ion.ac.cn

anatomical datasets with rs-fMRI. To address this limitation, and in alignment with a strategic plan developed by the NHP imaging community[8], we developed standardized protocols for imaging awake marmosets, which were implemented across two institutions, the National Institutes of Health (NIH), USA, and the Institute of Neuroscience (ION), China. This effort resulted in the largest awake NHP rs-fMRI dataset to date, which is being made available through an open-access platform. Furthermore, we integrated neuronal tracing and different diffusion MRI datasets into the same MRI space, resulting in a comprehensive resource that allows us to explore the relationships between the structural and functional connectomes by computational modeling.

## Results

The resource reported in this paper, summarized in Fig. 1, is supported by a publicly available standardized dataset. Following the same protocols for animal training and MRI imaging, including the designs of the radiofrequency coil and MRI pulse sequences, we acquired an extensive awake resting-state fMRI dataset to date from 39 marmosets of two research institutes (13 from ION, age $3 \pm 1$ years old; 26 from NIH, age $4 \pm 2$ years old; 12117 mins in total scanning, Supplementary Table 1 for details). This is also the same range of ages used in our previous studies of structural connectivity[13,14]. For test-retest evaluation, we scanned multiple runs (17 mins/run) for each marmoset, resulting in an essentially similar data quantity of two institutes (346 ION runs and 364 NIH runs) and included two "flagship" marmosets with many runs (64 runs from the ION and 40 runs from the NIH). Besides similar quantity, we comprehensively evaluated the data quality across two datasets, which is essential for data harmonization[18]. We compared different metrics in the SNR, temporal SNR (tSNR), spatial CNR, and head motions across the sites/scanners (Supplementary Figs. 1–3, Supplementary Table 2). We found no significant differences between the two datasets. We also compared the similarity of functional connectivity across subjects (Supplementary Fig. 4A) and sessions (Supplementary Fig. 4B), which still revealed no significant differences. The quality assessments and quality control (QA/QC) measurements demonstrated consistency and interpretability across datasets, which makes them suitable for further analysis.

Using these datasets, we created a comprehensive map of resting-state brain networks and a fine-grained functional cortical parcellation based on resting-state functional connectivity. Furthermore, we developed a deep-learning-based approach to map the population-based functional cortical parcellation onto individual brains. This allowed investigation of the structural basis underlying functional connectivity. For this purpose, we sampled the most extensive collection of NHP neuronal tracing data available (52 marmosets and 143 injections) onto the same MRI space at the voxel or vertex level and integrated it with the same functional MRI data space mentioned above. In addition, further enhancing the capacity of our resource, we also integrated extra high-resolution ex vivo diffusion MRI and in vivo diffusion MRI data obtained at 25 marmosets from the same cohort. On this basis, we investigated the relationship between structural and functional connectivity using a whole-brain computational model. To facilitate the user to explore connectomes reported in this paper and compare them with other connectomes[13–15], we make online connectome viewers (connectome.marmosetbrainmapping.org).

### Mapping functional brain networks

Identifying functional networks of areas showing highly correlated fMRI signals is critical to characterizing the brain architecture. Using the independent component analysis (ICA), a data-driven approach for separating independent patterns in multivariate data, we identified 15 cortical networks from awake resting-state fMRI data (Fig. 2; Supplementary Figs. 5–6 include the power spectrum). All specified components showed clear neural-like

patterns spatially (all peaks located in the cortical gray matter) and temporally (no patterns of artifacts or noises), as shown in Supplementary Fig. 6. We also conducted the ICA separately on each dataset. We found that both the ION and the NIH data revealed these components with similar spatial patterns and temporal CNR (tCNR, Supplementary Fig. 7), demonstrating the reproducibility of networks and the consistency of the two datasets.

The details of the 15 cortical networks were as follows. Six functional networks were characterized by short-range connectivity, including the ventral somatomotor (Fig. 2A), the dorsal somatomotor (Fig. 2B), the premotor (Fig. 2C), the frontopolar (Fig. 2D), the orbitofrontal (Fig. 2E), and the parahippocampal/ temporopolar cortex (Fig. 2F) networks. The next two components are the auditory and salience-related networks, the first being primarily located in the auditory and insular cortices, and weakly coupled with the anterior cingulate cortex (Fig. 2G), and the second (Fig. 2H) encompassing the anterior cingulate cortex. In addition, we also identified two transmodal networks (Fig. 2I–J), including association areas in the dorsolateral prefrontal cortex (dlPFC), rostral premotor cortex, lateral and medial parietal cortices, and temporal cortex. One of these is most likely the frontoparietal-like network (Fig. 2I), which has not been identified in previous studies[19,20], and the other is the default-mode network (DMN, Fig. 2J), which we characterized extensively in a previous study[21]. The remaining five networks represent the first complete mapping of visual-related functional networks of the marmoset cortex (Fig. 2K–O). Three networks included the primary visual cortex and parts of extrastriate areas related to far peripheral vision (Fig. 2K), near-peripheral vision (Fig. 2L), and foveal vision (Fig. 2M). The other two networks involve hierarchically higher visual areas (Fig. 2N–O), such as V3, V4, the inferior temporal cortex, the adjacent polysensory temporal cortex, and vison-related frontal regions.

Based on their spatial overlap patterns and connectivity strengths (normalized Z scores), we combined the 15 cortical networks into network-parcellation maps (Fig. 2P, Q). Due to local connectivity being stronger than long-range connectivity, the primary map (Fig. 2P, Q, top rows) is dominated by the short-range networks (i.e., Fig. 2G, I–L, I, and O). Thus, we created a second connectivity map (Fig. 2P, Q, bottom rows) to cover the long-range connectivity not captured by the primary map. The two network-parcellation maps characterized the entire cortical networks, and will likely be of value for future functional connectivity studies of the marmoset brain.

### Mapping functional connectivity boundaries

The brain network maps provided a global view of cortical functional organization. Next, we aimed to characterize the cortex at a finer local scale. For this, we used a functional connectivity boundary mapping approach to identify putative borders for the functional parcels[22–24] as an efficient way to map transitions in functional connectivity. Population boundary maps based on the ION, the NIH, or combined datasets are visually similar, presenting clear functional connectivity borders (Fig. 3A), and were highly reproducible with average Dice's coefficients for both hemispheres: 0.7 (ION-NIH), 0.71 (ION-Both), and 0.69 (NIH-Both), respectively (Supplementary Fig. 8). However, although consistent at the population level, boundary maps indicate variability across individuals (Fig. 3B), with an average Dice's coefficient of 0.3842 for both hemispheres (Fig. 3C, D), which is significantly lower than those for the population. We also found across-session variability in the same individual, but more scanning runs efficiently enhanced the reproducibility (Fig. 3E). Therefore, these results suggest that both individual and across-session variability contribute to the low consistency of individual boundary maps and that using test-retest data is essential for improving the reliability of maps.

## Marmoset Brain Mapping | Version 4

**Data**

*New Data*
**Awake Resting-state fMRI**

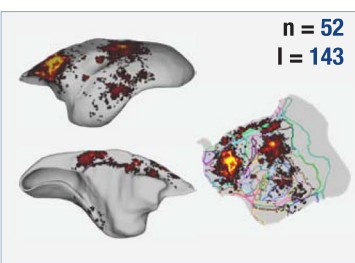

n = **39**
**710** runs
**12117** mins

*Major Updates*
**Voxel-level Neuro-tracing Data**

n = **52**
l = **143**

*New Data*
**In-vivo Diffusion MRI**

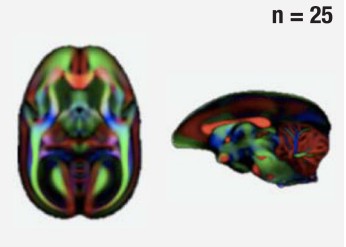

n = 25

**Mapping**

*ICA-based*
**Functional Network Parcellation**

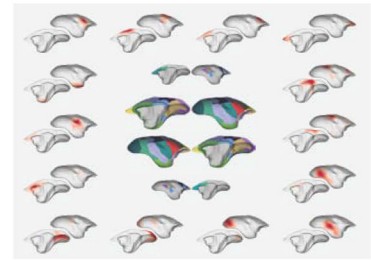

*Gradient-boundary-map based*
**Group Cortical Parcellation**

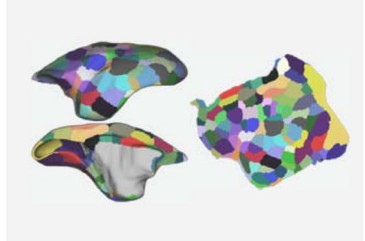

*Deep-learning based*
**Individual Parcellation Generator**

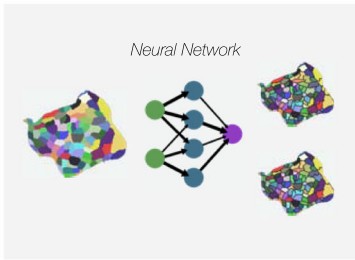

*Neural Network*

**Evaluation**

*Compared with*
**Existing Cortical Parcellations**

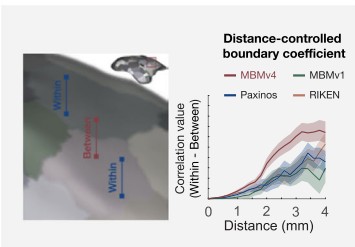

*Evaluated by*
**Functional Gradients and Activations**

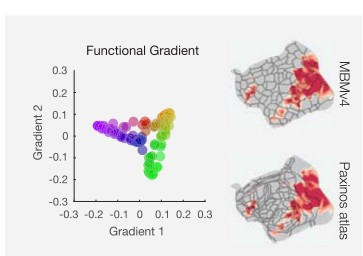

*Computational Modeling by*
**Structural and Functional Connectivity**

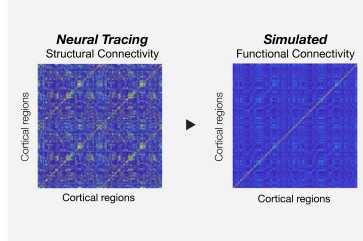

**Online**

*Interactive Atlas Viewer*
**Parcellations and Atlases**

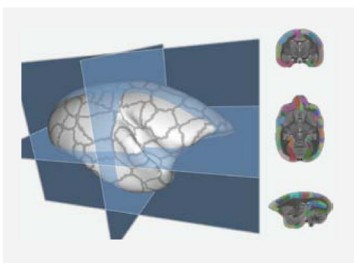

*Interactive Connectome Viewer*
**Connection Graphs**

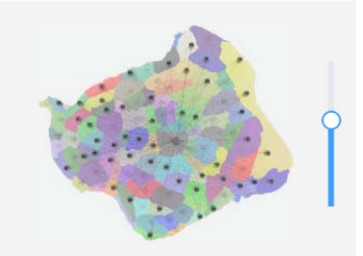

*Interactive Connectome Viewer*
**Connectivity Matrices**

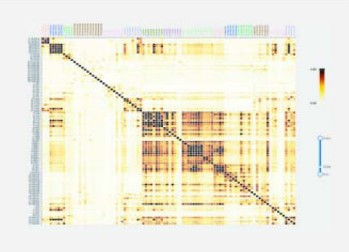

**Fig. 1 | Outline of Marmoset Brain Mapping resource.** The resource provides the awake test-retest resting-state fMRI data, in vivo diffusion MRI data from the same marmoset cohorts, and the neuronal tracing data mapped onto the same MRI space at the voxel/vertex level. In addition to the datasets, it also supports the study of whole-brain functional networks and computational modelings, as well as functional connectivity-based parcellation of the cortex (Marmoset Brain Mapping Atlas Version 4) using a deep neural network for accurate individual mapping. As a comprehensive multimodal resource for marmoset brain research, we also provide an online platform to explore the relationship between structural and functional connectivity. This functionality is embodied in online interactive viewers.

### Generation of functional connectivity parcels (Marmoset Brain Mapping Atlas Version 4, MBMv4)

Because the population boundary maps are more reproducible than individual maps, we used the combined ION-NIH population boundary map to generate cortical functional connectivity parcels. By the detection of the local-minima[22], "watershed-flood" region growing[25], and semi-manual optimization of parcel boundaries (Fig. 4A), we obtained 96 parcels per hemisphere (Fig. 4B). Since we processed each hemisphere independently, we compared the similarity of the parcellations of the two hemispheres. The hemispherical parcellations are

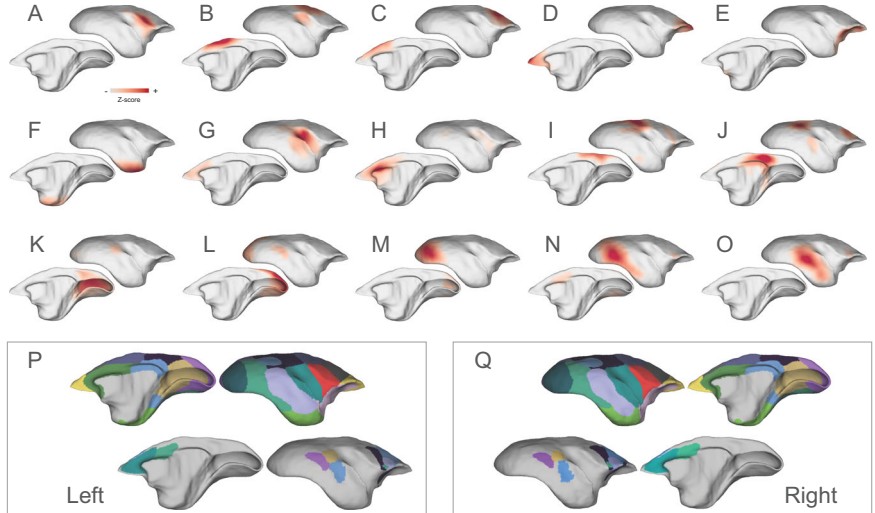

**Fig. 2 | Functional cortical networks and their parcellation maps.** The identified networks include: **A** the ventral somatomotor, **B** the dorsal somatomotor, **C** the premotor, **D** the frontal pole, **E** the orbital frontal cortex, **F** the parahippocampus, and temporal pole, **G–H** the auditory and salience-related network, **I–J** two transmodal networks, including a putative frontoparietal network and the default-mode network, and **K–O** visual-related networks from the primary visual cortex to higher-order functional regions. These networks were combined to form two network-parcellation maps (**P–Q**), which are dominated by the networks with short-range connectivity (**P–Q**, top rows) and with long-range connectivity (**P–Q**, bottom rows), respectively.

similar in the parcel sizes (Supplementary Fig. 9A; Wilcoxon paired signed-rank test, $N = 96$, $p = 0.7981$; Supplementary Fig. 9C for the size comparison in the subject's native space) and functional connectivity patterns between vertices within the same parcel (Supplementary Fig. 9B; Wilcoxon paired signed-rank test, $N = 96$, $p = 0.411$). For continuity with previously released resources[13,15,16], we named this functional connectivity-based parcellation of the cortex "Marmoset Brain Mapping Atlas Version 4". We also provide an online viewer to visualize parcellations, including previous versions (atlasviewer. marmosetbrainmapping.org).

To estimate the validity of the generated functional parcels, we used the distance-controlled boundary coefficient (DCBC)[26]. The basic idea of DCBC is that when a boundary divides two functionally homogenous regions, for any equal distance on the cortical surface, the functional connectivity pattern between vertices within the same parcel should be higher than that between vertices in different parcels (Fig. 4C). In other words, a higher DCBC (within-between) means higher within-parcel homogeneity and higher between-parcel heterogeneity. We calculated the DCBC between the vertex pairs using a range of spatial bins (0–4 mm) with a 0.5-mm step (the spatial resolution of the rs-fMRI data). Here, we compared the fit of the functional map represented by MBMv4 with existing structural cortical parcellations, including MBMv1 atlas[15], the digital reconstruction of the Paxinos atlas[15,27], and the RIKEN atlas[28]. The result of DCBC in Fig. 4C demonstrates that MBMv4 has the best performance for the presentation of functional connectivity (the average DCBC values were 0.0186, 0.0135, 0.0177, 0.0330 for RIKEN, MBMv1, Paxinos, and MBMv4 atlas; multiple comparisons for One-Way ANOVA $F_{(3,8556)} = 22.44$, $p = 1.81 \times 10^{-14}$).

**Mapping MBMv4 in individual brains by deep neural networks**
To overcome the limitation of variable individual boundary maps (see Fig. 3C–E), we employed a deep-learning approach for the individual Mapping from MBMv4 (Fig. 5A). First, based on the population-level whole-brain functional connectivity, we trained a deep neural network classifier for each parcel to learn the associated fingerprint of functional connectivity. Then, the trained networks distinguished the goal parcel for every marmoset based on the corresponding functional connectivity of the searching area, consisting of the goal parcel and its neighbors. Due to the overlap of searching

areas, vertices could belong to multiple parcels. Therefore, we only kept these vertices attributed to a single parcel as the seeds for regional growth by the "watershed" algorithm. This iterative region-growing procedure would assign all vertices to a parcel, resulting in an individual cortical parcellation.

Since individual parcellations should be close to the population definition[29,30], we compared the population-based MBMv4 parcellation and the automatically generated individual parcellations. By calculating the percentage of vertices sharing the same labels from both hemispheres (a metric of concordance), we found that the individual parcellations from all marmosets are similar to MBMv4 with an average of 90% concordance (Fig. 5B, the violin/box plot on the left, the examples on the right). Using the test-retest dataset, we revealed the consistency of the individual parcellations across different sessions (Fig. 5C, the violin/box plot on the left, and examples on the right). The across-session analysis yielded an average of 86.7% concordance, lower than the average value of 91.3% across-individual similarities. Furthermore, we observed that the lateral prefrontal cortex and occipital-temporal cortex had higher across-individual and across-session mapping variabilities (Supplementary Fig. 10), consistent with previous findings in humans[22,31]. Thus, the deep-learning approach efficiently adjusts the parcel borders to reflect the individual variabilities while maintaining high consistency with the population parcellation.

We also used the DCBC to evaluate whether the border adjustment of the individual parcellation captured the specific features of individual functional connectivity patterns. We assumed that the deep-learning-based method (DNN-reg) should result in a higher DCBC than the direct spatial registration of MBMv4 (Spatial-reg). Figure 5D (Top panel) presents the functional connectivity for the pairs of vertices within the same parcel (average correlation values within the same surface length 0–4 mm were 0.8331 and 0.8172 for DNN-reg and Spatial-reg) and between different parcels (average correlation values were 0.8256 and 0.8171 for DNN-reg and Spatial-reg). Thus, the DNN-reg had higher DCBC (within-between) than the Spatial-reg (Fig. 5D, bottom panel; the average DCBC values were 0.0167 and 0.0085 for the DNN-reg and the Spatial-reg, respectively; multiple comparisons for one-way ANOVA $F_{(1,2512)} = 20.35$, $p = 6.74 \times 10^{-6}$). In sum, the border adjustment by the proposed deep-learning network reflects individual functional connectivity patterns.

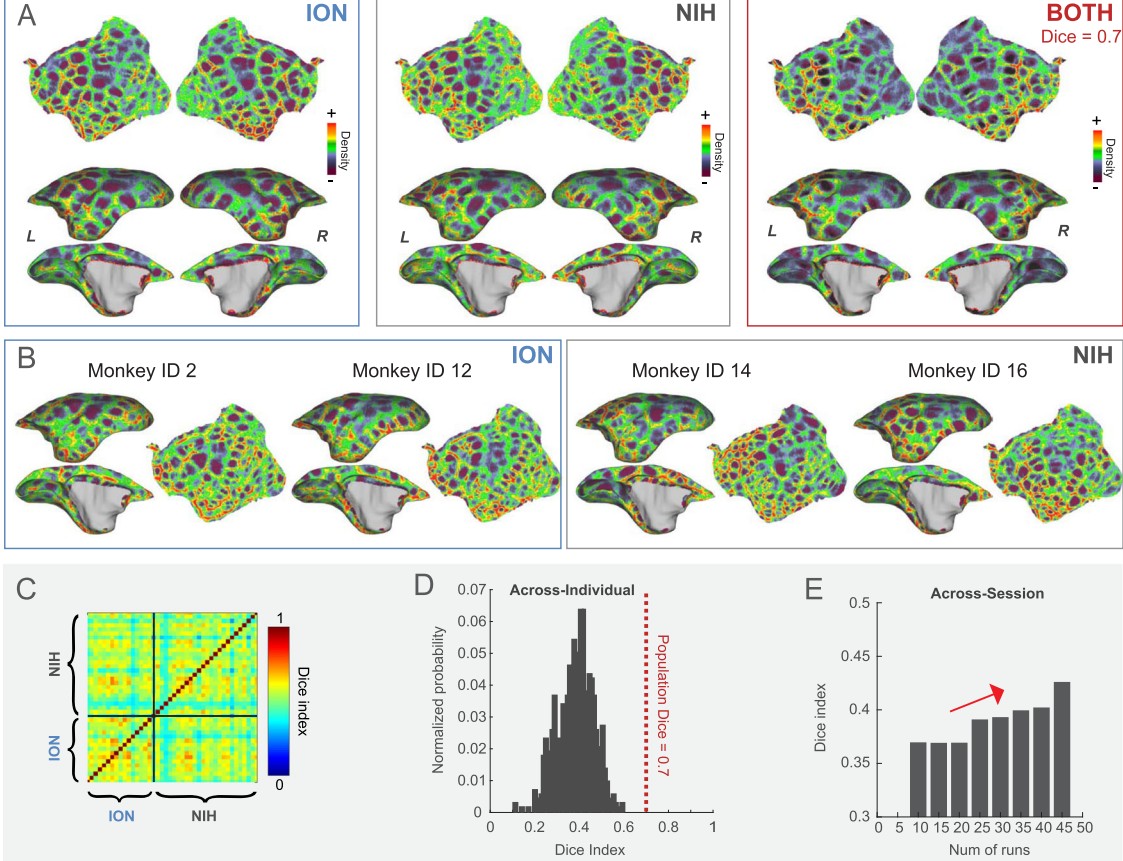

**Fig. 3 | The functional connectivity boundary maps. A** The population-based boundary maps from the ION, the NIH, and the combined datasets. These maps are highly consistent, with an average Dice coefficient of 0.7. **B** Boundary maps in the left hemisphere from four exemplar marmosets (two from the NIH cohort and two from the ION, including the flagship marmosets). **C, D** The heatmap of the average Dice's coefficients for both hemispheres between individuals and its distribution histogram. **E** The average Dice's coefficients change for both hemispheres with the number of runs in the same individuals.

## MBMv4 reflects accurate functional and topographical organizations

As evaluated from functional connectivity, MBMv4 provides a more accurate reflection of the MRI-based functional parcellation of the cortex than current histology-based atlases. To further verify this reliability, we took a task-activation map during the presentation of movie[32] which encompassed 10 deg × 8 deg of the visual field. This activation map was then registered onto the same individual MBMv4 map and the histology-based Paxinos atlas to examine the spatial overlap between the activations and functional parcels. As a result, we found that the MBMv4 has a good correspondence with task activations by visual inspection, such as the co-activation of foveal areas V1 and MT, and the temporal parcels (Fig. 6A, flat maps). Additionally, by measuring the shortest distances from every vertex in the boundary of the activation map to the atlas boundaries (MBMv4 or Paxinos boundaries), we found that the parcel borders of MBMv4 have higher consistency with the activation map than the Paxinos atlas (Fig. 6A, the scatterplots; Wilcoxon paired signed-rank test: Monkey ID 25, $N = 878$, $p = 3.07 \times 10^{-40}$ for the left hemisphere; $N = 816$, $p = 6.11 \times 10^{-26}$ for the right hemisphere. Monkey ID 15, $N = 826$, $p = 2.22 \times 10^{-25}$ for the left hemisphere; $N = 850$, $p = 2.95 \times 10^{-53}$ for the right hemisphere). Thus, MBMv4 reflects functional differences that cytoarchitectonics does not capture, possibly because the latter contains the full visual field representations. Thus, the MBMv4 may provide functional-related contrast that can help enhance the precision of cross-species studies[33].

Besides the clear functional boundaries, MBMv4 also preserved the topographical organization of the functional connectivity. Recent methodological developments have allowed complex brain features to

be mapped to the low-dimensional representations as gradients[34], and these gradients characterized the topographical organization of the functional brain connectome from unimodal to transmodal networks. If the atlas complies with this topographical organization, it should be able to identify such gradients. As shown in Fig. 6B left panel, MBMv4 results in a pattern of gradient spectrum for functional connectivity. In contrast, we did not find a gradient pattern based on the Paxinos et al. (2012) atlas (right panel in Fig. 6B). Therefore, MBMv4 offers an alternative view to understanding the functional connectome of the marmoset brain by reflecting the characteristics of functional connectivity.

## MBMv4 is an essential link between the functional and structural connectivity

Since MBMv4 offers a more comprehensive scheme to study the functional connectome, it is worth linking it to structural information to investigate the relationships between structural and functional connectivity. We have also provided online viewers to facilitate the comparison between them (connectome.marmosetbrainmapping. org). Furthermore, we used a whole-brain computational model[35–37] to explore structural and functional connectivity relationships. The processing procedure is shown in Fig. 7A. First, we established structural connectivity values using the in vivo diffusion MRI, the ex vivo ultra-high-resolution diffusion MRI, or the neuronal tracing datasets fitted to either the MBMv4 or the Paxinos et al.[14] atlas parcellations. Then, we used a Hopf bifurcation hemodynamic model[36] to simulate the functional connectivity of every brain parcel or region based on their structural connectivity and compare it with the

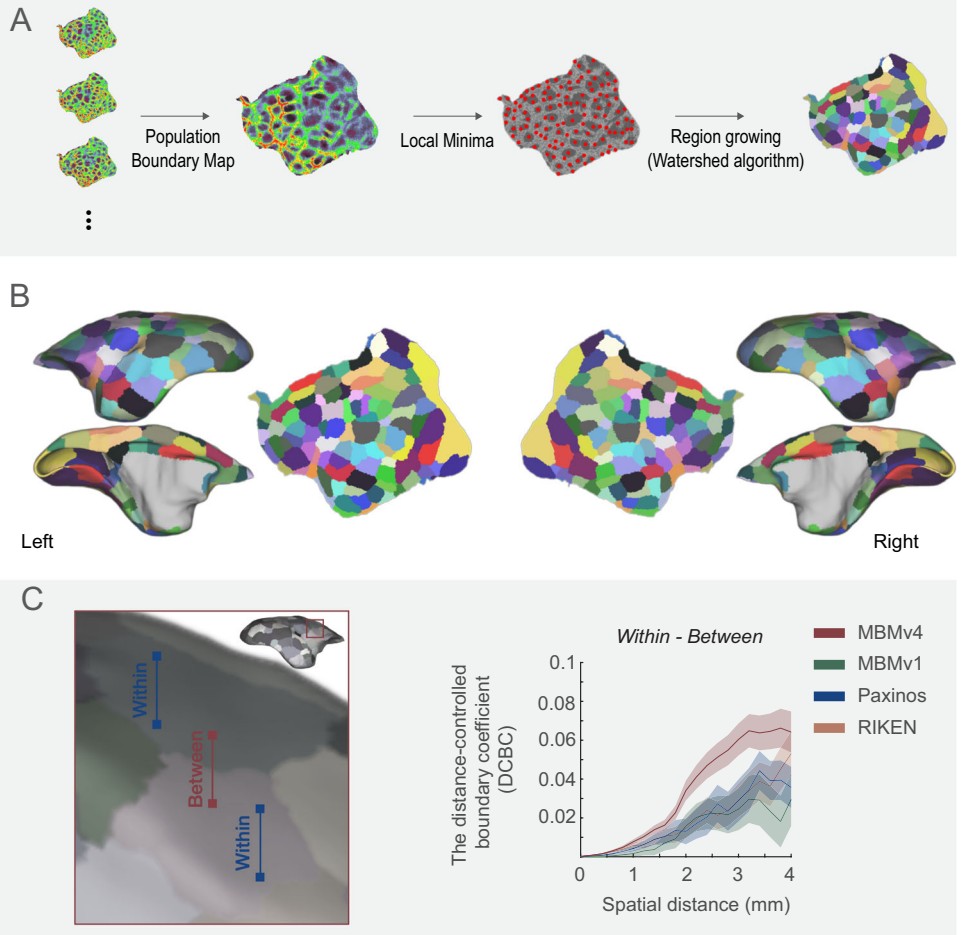

**Fig. 4 | Marmoset Brain Mapping Atlas Version 4 (MBMv4). A** The processing procedure includes generating the population functional connectivity boundary maps, defining the local minima for seeding, and generating parcels by the "watershed" algorithm. **B** The resulting 96 functional connectivity parcels per hemisphere are overlaid on the white matter surface and flat map of MBMv3[16]. **C** Distance-controlled boundary coefficient (DCBC) evaluation metric. All pairs of voxels/vertices were categorized into "within" or "between" parcels (left panel) according to different brain parcellations (MBMv1, MBMv4, Paxinos, and RIKEN atlases, right panel), and the DCBC metric was calculated by the differences (within-between) in functional connectivity as the function of distance on the surface (0–4 mm in steps of 0.5 mm). Data are presented in mean ± s.e.m.

empirical functional connectivity from the actual resting-state fMRI data. We used Pearson's correlation to measure the similarity between the simulated and the empirical functional connectivity. Additionally, we used group-average functional connectivity as an empirical observation for the ex vivo diffusion MRI and neuronal tracing dataset and individual functional connectivity for the individual in vivo diffusion MRI.

The modeling results in Fig. 7B, C demonstrate that: (1) the MBMv4 parcellation provides the best fitting value ($R = 0.721$) compared to the Paxinos atlas ($R = 0.638$) for the extra high-resolution ex vivo diffusion MRI dataset (the polygon in Fig. 7B and results in Fig. 7C); (2) the MBMv4 parcellation fits the in vivo diffusion MRI datasets better than the Paxinos atlas in a variety of individual simulations (all circles in Fig. 7B: the average fitting values from 25 animals were $R = 0.4707$ for MBMv4 and $R = 0.3659$ for Paxinos atlas); and (3) both parcellations performed equally ($R = 0.525$ for MBMv4 and $R = 0.472$ for Paxinos atlas) fitting the cellular connectivity data from the aggregated neuronal tracing (the star in Fig. 7B and results in Fig. 7C). However, no matter which structural data was used for estimating functional connectivity, we found that the modeling predicted by MBMv4 fits the empirical functional connectivity data better than the Paxinos atlas (Fig. 7B; Wilcoxon paired signed-rank test: $N = 27$, $p = 0.002947$).

Due to its inherent limitation for tracking long connections, the diffusion tractography was more affected by distance, with longer connections having lower FC fitting values (Fig. 7D, blue lines). Based on studies demonstrating that cellular-resolution connections follow an exponential distribution of projection lengths[38–40], we introduced an exponential distance rule (EDR) to our model to compensate for the distance effect (Fig. 7D, redlines). Notably, the EDR fully corrected the distance effect in neural tracing data (Fig. 7D), suggesting that the neural tracing data might be a more reliable bridge to link structural and functional connectivity of long-range connections. Still, as shown in Fig. 7E, F, the EDR-constrained modeling based on MBMv4 fits the empirical functional data better than the Paxinos atlas (Wilcoxon paired signed-rank test: $N = 27$, $p = 0.01947$), similar to the modeling without EDR (Fig. 7B), showing that MBMv4 is better suited for investigating the relationship between structural and functional connectivity.

## Discussion

There are challenges in adapting well-established approaches for human neuroimaging to NHP neuroimaging[5]. The present study used effective and practical animal training and imaging protocols to scan a large cohort of marmosets. Despite the different MRI scanners (7T and 9.4T) in two institutions (ION and NIH), the protocol produced similar

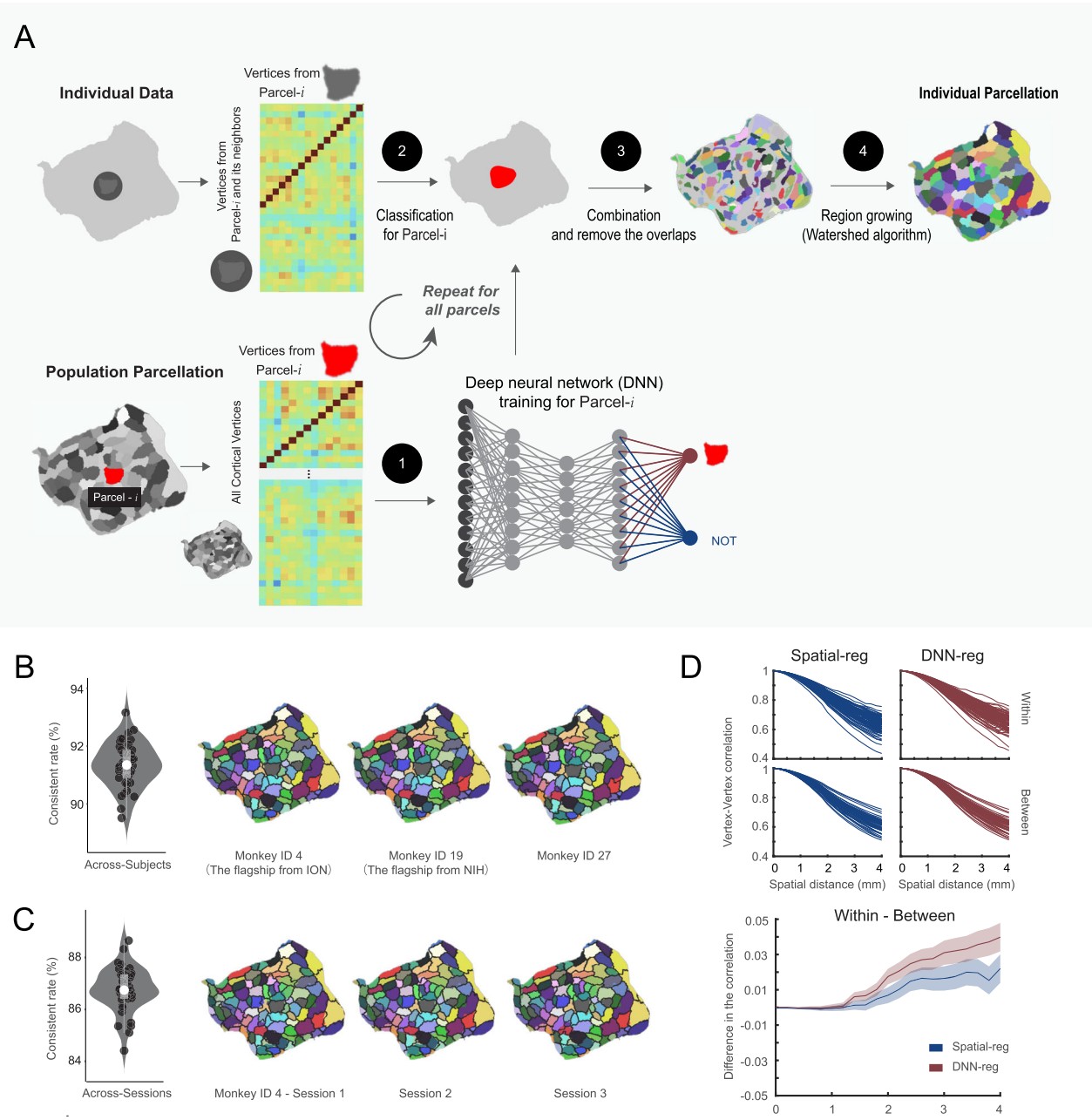

**Fig. 5 | Mapping individual functional connectivity parcellation. A** An overview of individual Mapping based on the deep neural network approach. **B** MBMv4 Mapping of each individual. Left panel: the concordance between the population MBMv4 and individual parcellations (*N* = 78, all hemispheres from 39 subjects). Data are presented by the violin and the box plots (the 25th percentile: 0.9068; the 75 percentile: 0.92448), in which the white point represents the average value 0.915 (the maximum value: 0.931; minimum 0.900); Right panel: three examples of individual parcellations. The underlay (color-coded) presents the population MBMv4, and the overlay (black border) shows the individual parcellations. **C** Mapping of MBMv4 per session. Left panel: The concordance between every individual parcellation and the corresponding parcellation using one session data (*N* = 78, all hemispheres from 39 subjects). Data are presented by the violin and the box plots (the 25th percentile 0.865; the 75 percentile 0.877), in which the white point represents the average value 0.874 (the maximum: 0.878; the minimum 0.859); Right panel: representative parcellations of three sessions from one marmoset. The color-coded underlay represents individual parcellation, while the black border overlay shows the session-based parcellation. **D** The distance-controlled boundary coefficient (DCBC) for the individual parcellation generated by the spatial registration (Spatial-reg, blue) and the deep neural network (DNN-reg, red). Top panel: the functional connectivity for all pairs of vertices within the same parcel and between parcels for DNN-reg and Spatial-reg, respectively. Bottom panel: the comparison of DNN-reg and Spatial-reg by DCBC. Data are presented in mean ± s.e.m.

data quality, suggesting the compatibility of our approach (see Methods for details, Supplementary Figs. S1–S4). Therefore, we pooled the in vivo resting-state fMRI datasets from both institutions (details in Supplementary Table 1) to create the most comprehensive functional connectivity dataset of the NHP brain to date. We integrated this dataset with in vivo diffusion MRI of the same cohort, high-resolution ex vivo diffusion MRI[13], and the most extensive mesoscale retrograde neuronal tracing data[14]. This resource expedites the mapping of marmoset brains and will allow cross-species comparisons.

Like the human cortex, the marmoset cerebral cortex comprises large-scale functional networks. However, the first awake resting-state fMRI study of the marmosets[19] found only 12 functional networks

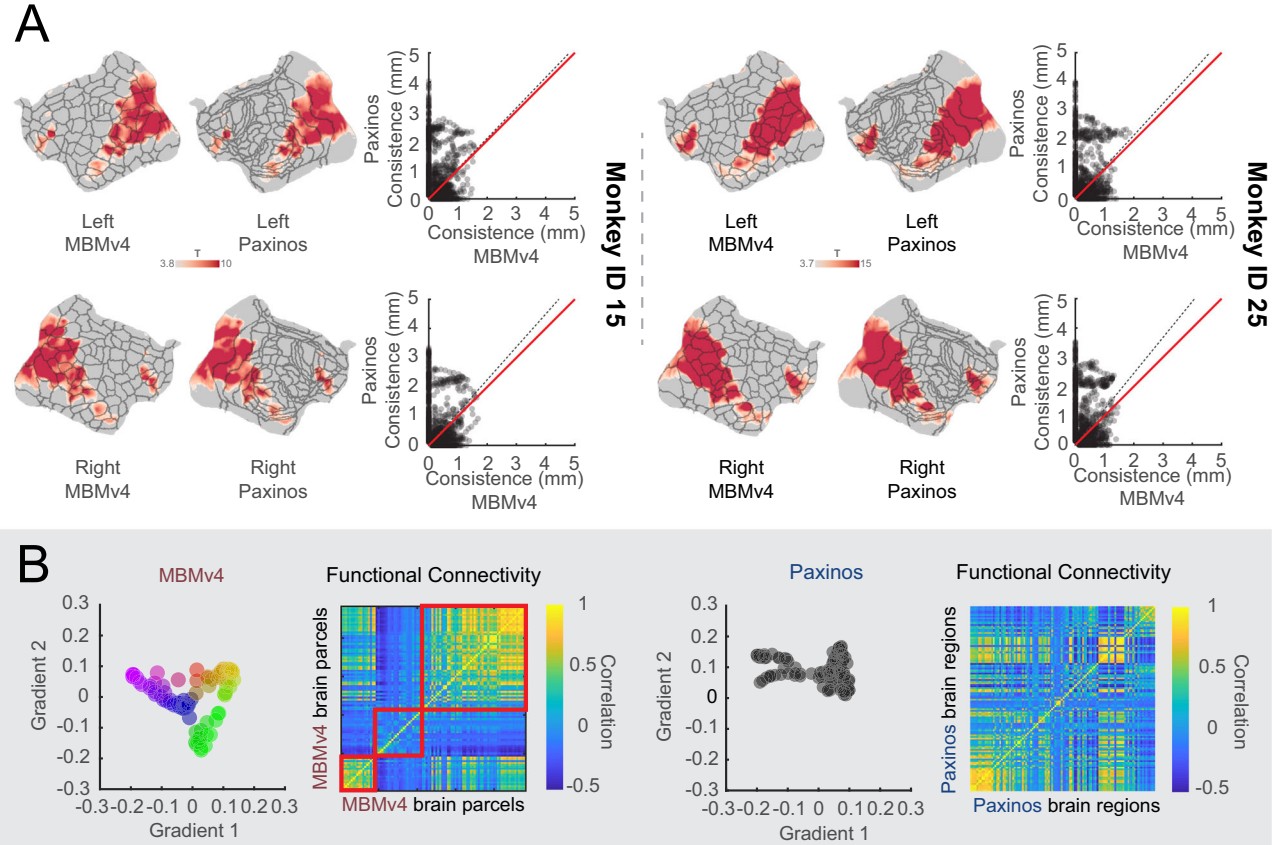

**Fig. 6 | MBMv4 matches functional boundaries and preserves the topographical organization of the functional connectivity. A** The visual activation maps from two monkeys are overlaid on the parcel boundaries from individual MBMv4 parcellation and Paxinos atlas (Left panel: monkey ID 15; Right panel: monkey ID 25). The scatter plots compare the boundary matching of the MBMv4 and the Paxinos atlas with the activation maps, measured by the shortest distance from every voxel in the borders of the activation maps to the parcel borders of the MBMv4 or the Paxinos atlas. The dashed black line represents the diagonal line, and the red line represents the linear fitting line. **B** The scatter plots in the left panel are the first two axes of gradients (the color scale of dots represents the scores of the first axis for every gradient), decomposed by the functional connectivities of the MBMv4 and the Paxinos atlas (the spectrum colors denote the gradient position in this 2D space). The heat maps of functional connectivity sorted by the scores of the first axis (gradient 1) are shown in the right panel.

(10 cortical networks), and another ICA-based study described 8 brain networks, possibly due to the influence of anesthesia[20]. Based on a more extensive awake rs-fMRI dataset, the present study mapped the large-scale functional networks and built the first network-based parcellation, providing a more comprehensive description of functional networks in the marmoset brain, including a total of 15 cortical networks. Moreover, based on functional connectivity boundary maps, we created a population-based cortical parcellation on a fine scale (MBMv4; Fig. 4) with 192 distinct parcels (96 per hemisphere). We further verified hemispheric symmetry by warping the standard average surfaces to every subject's native space in Supplementary Fig. 9C. A previous study of the human cerebral cortex identified 422 discrete functional connectivity parcels using the same approach, 206 in the left hemisphere and 216 in the right hemisphere[22]. Therefore, our results may reflect that the asymmetry in marmosets may be smaller than in humans, as expected from previous analyses based on anatomical measurements[41] and the evidence that the number of cortex subdivisions increases with brain volume[42].

It is important to emphasize that our functional parcels do not correspond to the traditional cytoarchitectonic definition of the cortical areas[43,44]. Consistent with many brain parcellations generated by non-invasive neuroimaging[22,29,45–47], our functional parcels most likely reflect a different type of computational sub-units, agreeing with the idea that the brain is organized in multiple scales[48,49]. Therefore, compared with other available structural atlases, MBMv4 captures the organization of functional connectivity accurately. For example,

MBMv4 achieved better task correspondence (Fig. 6A) due to a strong link between task-fMRI and rs-fMRI[50–52]. Another piece of evidence for the accuracy of the MBMv4 parcellation is the topographical gradient organization of functional connectivity (Fig. 6B). Finally, MBMv4 better links the structural and functional connectivity, as demonstrated by our modeling simulation (see Fig. 7).

Consistent with previous findings in humans[22,29,47], the parcels defined in MBMv4 do not follow the boundaries of cytoarchitectonic areas, thus demonstrating an essential difference between anatomical features and functional connectivity. For example, the somatomotor cortex is parcellated into subregions that appear to correspond to representations of the face, forelimbs, hindlimbs, and trunk musculatures across multiple areas, and areas such as V1 and V2 are subdivided into several functional parcels according to the representation of eccentricity in visual field representation, which is contiguous across areas[53] but may include discontinuities[54]. Previous studies revealed that some topographically organized cytoarchitectonic areas could be dissociated from the resting-state functional responses[55,56]. Therefore, the present MBMv4 should be considered a functional connectivity description, providing complementary information about a type of organization that cannot be observed via classical anatomy.

An essential goal of this study was to reflect individual characteristics by creating parcels from each individual's data. Although the boundary map-derived parcels could be used for individual analysis, we found that the subject boundary maps had significant

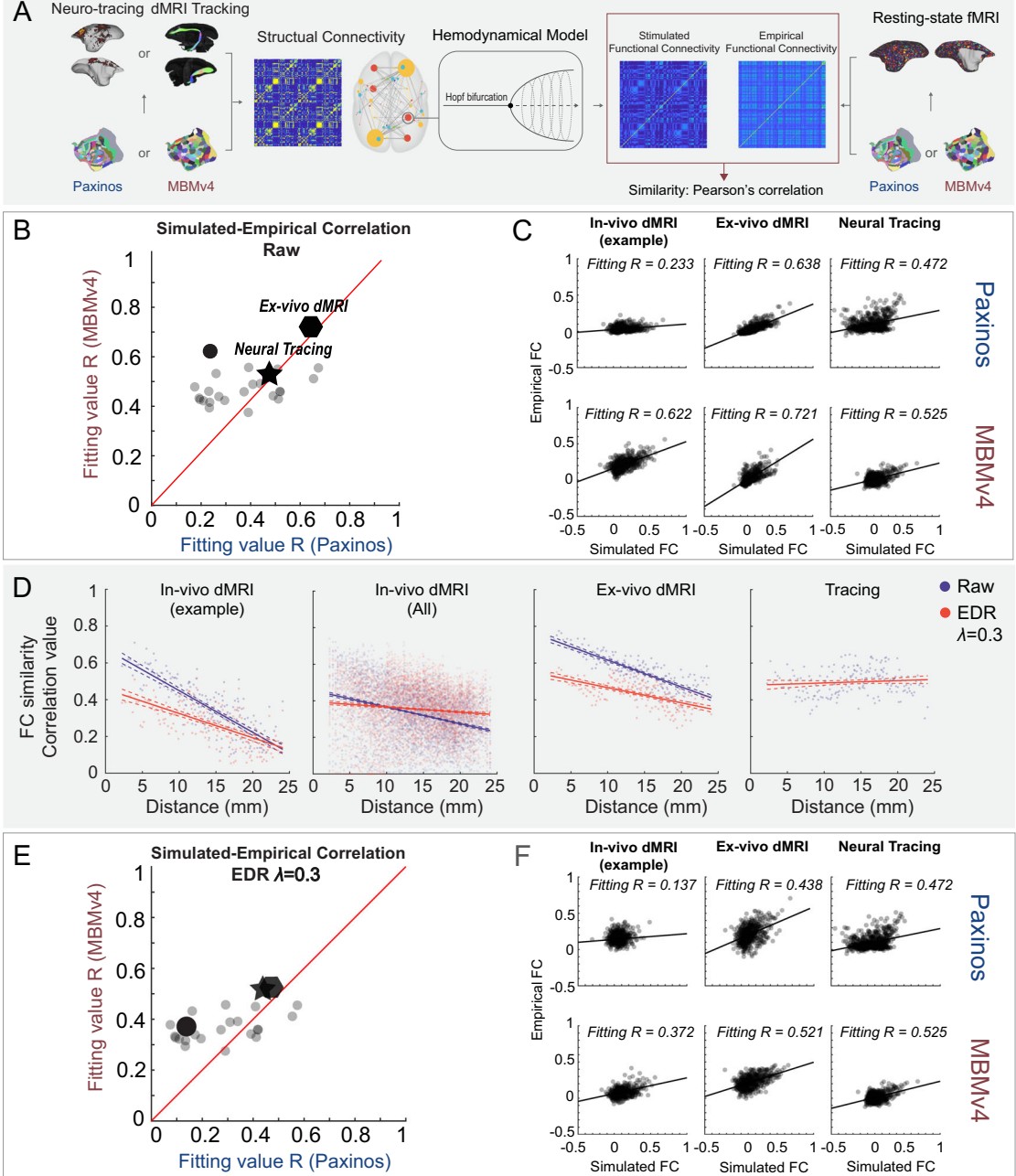

**Fig. 7 | A computational framework links the structural-functional connectivity according to different parcellations. A** Application of whole-brain modeling, including an estimation of structural connectivity from neuronal tracing, diffusion MRI (in vivo or ex vivo) according to the Paxinos atlas or MBMv4, simulation of functional connectivity from structural connectivity by the Hopf bifurcation hemodynamic functions, and a similarity measure with empirical connectivity from resting-state fMRI. **B** Model fitting comparison in different spatial scales using the Paxinos versus the MBMv4 atlases. Individual examples using in vivo diffusion MRI (round dots; an example animal marked in black dot), ex vivo diffusion MRI (polygon), and neuronal tracing (star) show higher fitting values for MBMv4 versus the Paxinos atlas. **C** Examples of correlations between the simulated and empirical functional connectivity from **B**; solid black lines represent marginal regression

lines. **D** Model fitting comparisons with (red) and without (blue) an exponential distance rule (EDR) correction. From left to right, the plots present the simulation results obtained with in vivo diffusion MRI from the example animal (the linear regression values $R = 0.689$, $p = 9.27e\text{-}50$ for the blue line; $R = 0.413$, $p = 1.28e\text{-}23$ for the red); in vivo diffusion MRI from all animals ($R = 0.11$, $p = 6.05e\text{-}114$ for the blue; $R = 0.011$, $p = 4.17e\text{-}13$ for the red); the ex vivo diffusion MRI ($R = 0.7$, $p = 2.69e\text{-}51$ for the blue; $R = 0.395$, $p = 2.19e\text{-}22$ for the red); and the neuronal-tracing dataset ($R = 0.009$, $p = 0.02$). The line plots are presented as mean values ± 95% C.I. Dashed lines represent the 95% confidence interval. **E** Model fitting comparison with EDR correction between the Paxinos atlas and MBMv4 in different spatial scales. **F** Examples of correlation between the simulated and empirical functional connectivity from **E**.

variations and that the reproducibility became lower than the group-level map (Fig. 3). This finding emphasizes the need to acquire large amounts of data for the reliable test-retest of the individual boundary map. Accordingly, we developed a deep neural network to map reliable population-level MBMv4 into every individual nonlinearly. As a result,

we demonstrated good reliability in the test-retest dataset (across sessions from the same individual; see Fig. 5C) and the applications of task-fMRI activation mapping from the same individuals (Fig. 6A). Importantly, the locations of the most variable functional parcels are in the lateral prefrontal cortex and lateral temporal-occipital cortex (see

Supplementary Fig. 10), corresponding to previously reported exceptionally high inter-subject variability resting-state functional connectivity patterns[31]. Moreover, these regions co-locate, expanding preferentially in primate evolution[57] and maturing later in postnatal development[58]. As resting-state functional connectivity can be altered by many biological features, including behavioral context during development associated with phenotypic correlations, a better understanding of the causes of inter-subject parcel variation will be an important focus for our future work.

In addition to the functional connectome mapping, we integrated all currently available structural connectome datasets, including the in vivo diffusion MRI, the ex vivo high-resolution diffusion MRI[13], and the mesoscale tracing dataset[14]. Using multimodal data allowed us to investigate the relationship between functional and structural connectivity with unprecedented detail. Using whole-brain modeling[36,37], we observed that the structural connectivity simulated functional connectivity based on MBMv4 had a high coherence with empirical data, no matter which types of structural connectivity were used (Fig. 7B, C and examples Fig. 7E, F). The finding corroborates the conclusion that MBMv4 reflects meaningful computational sub-units from the view of whole-brain functional connectivity. Meanwhile, we also found room for modeling performance improvement through the detailed estimation of structural connectivity. For example, compared with in vivo diffusion MRI data, the ultra-high-resolution ex vivo diffusion MRI data from a brain sample could provide more thorough structural information. Notably, we should also be careful about the influence of the distance when we use the diffusion dataset to simulate functional connectivity because the diffusion tract may generate the fiber cut and so on in the long-distance tract (Fig. 7D, blue curves). To overcome this shortcoming, we introduced a correction factor for structural connectivity based on the EDR (Fig. 7D, red curves). Nevertheless, the simulation results in Fig. 7B, E demonstrated the best fitting by the present MBMv4, no matter which datasets were used. Furthermore, since our modeling is simple, with only two parameters that avoid overfitting simulation (see our method description), the whole-brain model could be an efficient tool with broad applications to link structure and function for future studies.

Although we provided the most comprehensive multimodal data resource for mapping the marmoset connectome, our current study still faced limitations that need to be addressed by future work, which is essential for both experimenters and users of non-human primate neuroimaging. First is the data collection. The population used to generate the MBMv4 was sex-biased (31 males vs. 8 females) due to the priority of colony expansion worldwide. The neuronal tracing data were also limited, not covering all cortical regions and missing subcortical information. Including neuronal-tracing data is critical for accurately mapping the future fully structural connectome. Finally, our parcellation only used the resting-state functional connectivity information, as in many human studies[22,45]. Therefore, multimodal brain parcellation incorporated structural contrasts, especially the T1w/T2w myelin map, multiple task-fMRI data, gene expression data, etc., will be necessary to fully capture the anatomical and functional architectures of the marmoset brain.

## Methods
### Data collection and preprocessing
**Animals and MRI scanning.** Experimental procedures followed policies established by the Chinese Laboratory Animal – Guideline for Ethical Review of Animal Welfare (ION data) or the US Public Health Service Policy on Humane Care and Use of Laboratory Animals (NIH data). All procedures were approved by the Animal Care and Use Committee (ACUC) of the Institute of Neuroscience, Chinese Academy of Sciences (ION data), or the ACUC of the National Institute of Neurological Disorders and Stroke, National Institutes of Health (NIH data). The respective ACUC-approved protocols specify group size numbers based on a power analysis to detect differences between animals to ensure rigor and reproducibility of the results while minimizing the number of animals used in the study. Our studies are powered to detect inter-individual differences. The experimental designs are typically two or 3-factor ANOVAs. Values of $p < 0.05$ are considered statistically significant. The number of animals used is the minimum necessary to provide reliable estimates of inter-individual effects based on power considerations. Typically, sample size estimates are based on the number of animals needed to achieve a power of 0.80 for moderate effect size and 0.99 for large effect size. Both marmoset colonies are socially housed to ensure psychosocial well-being and are offered a varied diet that includes food treats. In addition, dedicated husbandry and veterinary teams interact with the animals daily as part of the psychological enrichment plans approved by the ACUCs of both institutions.

The data acquisition procedure from both centers followed the same animal training protocol, 8-element radiofrequency (RF) coil design[59], and MRI scanning protocols. Thirteen marmosets (12 males and 1 female) were recruited from the ION cohort, from which we generated 62 awake resting fMRI sessions and 349 runs (17 min per run). As three of the 349 runs had extensive head motions (>10% time points were motion censored based on the preprocessed pipeline described below), we excluded the three runs from the analysis, resulting in a total of 346 runs (see Supplementary Table 2 for the summary of the head-motion per run). Twenty-six marmosets (19 males and 7 females) were recruited from the NIH cohort to produce 51 awake resting-state fMRI sessions and 364 runs. Therefore, the NIH and ION data had a comparable number of valid runs. The two datasets included 39 marmosets with 113 sessions, 710 valid fMRI runs, and 12117 mins of total scan time. Detailed demographic information is provided in Supplementary Table 1. All marmosets underwent a 3–4-week acclimatization protocol as previously described[60]. After completing the training, all marmosets were properly acclimated to laying in the sphinx position in an MRI-compatible cradle. Their heads were comfortably restrained with 3D-printed anatomically conforming helmets that allowed the resting-state fMRI (rs-fMRI) data acquisition as the animals lay relaxed in their natural resting position.

All 39 marmosets were imaged using identical rs-fMRI protocols and pulse sequences, except for a minor adjustment in the echo time (TE) to accommodate hardware differences between the ION and the NIH gradient sets. The ION marmosets were scanned in a 9.4T/30 cm horizontal MRI scanner (Bruker, Billerica, USA) equipped with a 20-cm gradient set capable of 300 mT/m gradient strength and an 8-channel phased-array RF coil[59] custom-built for marmosets (Fine Instrument Technology, Brazil). Multiple runs of rs-fMRI data were collected in ParaVision 6.0.1 software using a 2D gradient-echo (GE) EPI sequence with the following parameters: TR = 2 s, TE = 18 ms, flip angle = 70.4°, FOV = 28 × 36 mm, matrix size = 56 × 72, 38 axial slices, slice thickness = 0.5 mm, 512 volumes (17 min) per run. The GE-EPI fMRI data were collected using two opposite phase-encoding directions (LR and RL) to compensate for EPI distortions and signal dropouts. Two sets of spin-echo EPI with opposite phase-encoding directions (LR and RL) were also collected for EPI-distortion correction (TR = 3000 ms, TE = 37.69 ms, flip angle = 90°, FOV = 28 × 36 mm, matrix size = 56 × 72, 38 axial slices, slice thickness = 0.5 mm, 8 volumes for each set). After each rs-fMRI session, a T2-weighted structural image (TR = 8000 ms, TE = 10 ms, flip angle = 90°, FOV = 28 × 36 mm, matrix size = 112 × 144, 38 axial slices, slice thickness = 0.5 mm) was scanned for co-registration purposes.

The NIH marmosets were scanned in a 7T/30 cm horizontal MRI (Bruker, Billerica, USA) equipped with a 15 cm customized gradient set capable of 450 mT/m gradient strength (Resonance Research Inc., Billerica, USA) and an 8-channel phased-array RF coil custom-built for marmosets[59] with identical coil geometry to the one used by ION. Multiple runs of rs-fMRI data were collected during each scanning

session in ParaVision 6.0.1. software using a 2D gradient-echo (GE) EPI sequence with the following parameters: TR = 2s, TE = 22.2 ms, flip angle = 70.4°, FOV = 28 × 36 mm, matrix size = 56 × 72, 38 axial slices, slice thickness = 0.5 mm, 512 volumes (17 min) per run. The GE-EPI fMRI data were collected using two opposite phase-encoding directions (LR and RL) to compensate for EPI distortions and signal dropouts. Two sets of spin-echo EPI with opposite phase-encoding directions (LR and RL) were also collected for EPI-distortion correction (TR = 3000 ms, TE = 36 ms, flip angle = 90°, FOV = 28 × 36 mm, matrix size = 56 × 72, 38 axial slices, slice thickness = 0.5 mm, eight volumes for each set). After each rs-fMRI session, a T2-weighted structural image (TR = 6000 ms, TE = 9 ms, flip angle = 90°, FOV = 28 × 36 mm, matrix size = 112 × 144, 38 axis slices, slice thickness = 0.5 mm) was scanned for co-registration purposes. Furthermore, multishell diffusion MRI (DTI) datasets were collected using a 2D diffusion-weighted spin-echo EPI sequence with the following parameters: TR = 5.1 s, TE = 38 ms, a number of segments = 88, FOV = 36 × 28 mm, matrix size = 72 × 56, slice thickness = 0.5 mm, a total of 400 DWI images for two-phase encodings (blip-up and blip-down) and each has 3 b values (8 $b = 0$, 64 $b = 2400$, and 128 $b = 4800$), and the scanning duration was ~34 min. The multishell gradient sampling scheme was generated using the Q-shell sampling method[61].

**Data preprocessing.** The rs-fMRI datasets were preprocessed by the customized script involving AFNI[62], FSL[63], ANTs[64], and Connectome Workbench[65]. In brief, the rs-fMRI data were slice-timing-corrected and motion-corrected by the "3dTshift" and "3dvolreg" commands of AFNI and corrected for EPI distortions by the "top-up" command of FSL (see our examples in Supplementary Fig. 11). The rs-fMRI datasets were further preprocessed by regressing linear and quadratic trends, motion parameters and their derivatives, and motion-sensor regressors (any TRs and the previous TRs were censored if the detection motion was >0.2 mm and temporal outlier >0.1). Note that, for the motion measurements, we calculated the weighted euclidean norm of six motion parameters with a 0.25 weight for the three rotation degrees (yaw, pitch, and roll) according to the relative head radius of the marmosets compared to humans. White matter and cerebrospinal fluid signals were removed, and the rs-fMRI datasets were bandpass filtered (0.01–0.1 Hz). The above nuisance signal regression and bandpassing filtering were carried out by the "3dDeconvolve" and "3dTproject" commands in AFNI. Next, the preprocessed data were spatially normalized to the template space of our Marmoset Brain Atlas Version-3 (MBMv3) by the "antsRegistration" routine of ANTs[16]. The spatial normalization concatenated multiple transformations, including (1) rigid-body transformation of each fMRI run to the T2-weighted image acquired at the end of each session, (2) rigid-body transformation of T2-weighted images from each session to a cross-session averaged T2-weighted image from each animal, (3) affine and nonlinear transformation of the averaged T2-weighted image from each animal to the T2w template of our MBMv3 space. Finally, all preprocessed data were mapped to 3D brain surfaces of the MBMv3 using the Connectome Workbench (wb_command -volume-to-surface-mapping function and ribbon constrained mapping algorithm), normalized (subtract mean and divide by standard deviation) and concatenated per session before the boundary mapping described below. The preprocessed data were smoothed with 1mm FWHM using 3dBlurInMask (for volume data) and wb_command -cifti-smoothing (for surface data), respectively, before the network analysis and cortical parcellation.

The in vivo diffusion MRI dataset was preprocessed by the DIFF_PREP, DR_BUDDI, and DR_TAMAS pipelines of TORTOISE[66]. The DIFF_PREP and DR_BUDDI routines incorporated correction for eddy-currents- and EPI-induced distortions using pairs of diffusion data acquired with opposite phase encoding (blip-up and blip-down) and the T2-weighted image and merging the preprocessed pairs into one

dataset. The nonlinear spatial registration from the individual space to the DTI template of our MBMv3 space[16] was carried out using the DR_TAMAS routine of TORTOISE. The registration information was then used to transform multiple atlases into the individual space for diffusion tractography.

All diffusion trackings were performed using the iFOD2 method of the software Mrtrix3[67]. The response function of each preprocessed diffusion MRI data was calculated by the "dhollander" method of the "dwi2response" command, and then the fiber orientation distributions (FOD) were estimated using spherical deconvolution by the multishell multi-tissue CSD method of the "dwi2fod" command. Finally, region-to-region tractography was performed using the iFOD2 method of the "tckgen" command. For each pair of cortical regions, diffusion tractography was conducted by using one region as the seed and the other as the target, and vice-versa. Thus, each pair of regions generated two sets of tracking probability maps, which were normalized by the total streamlines selected, and the two probability maps were averaged into a single map to represent the final map of the connection between the two regions. Finally, all pairs of connections formed the whole cortical structural connectome for computational modeling.

The neuronal tracing data were mapped onto the histological NM template from our previous study[17]. The NM template is a population-based 3D cortical template generated from Nissl-stained serial sections of 20 marmosets. Since the NM template only covers the cortex and has Nissl-stain contrast and a 75μm isotropic high spatial resolution, its direct spatial transformation to our in vivo MBMv3 template is inaccurate. Thus, we modified the 80μm isotropic ultra-high-resolution MTR template of our Marmoset Brain Atlas Version-2 (MBMv2) atlas[13] to remove the parts of the brain that were not covered in the NM template, including the cerebellum, brainstem, and parts of subcortical structures. This step increased the accuracy of registration between the NM template and the MBMv2 template. Then, the ex vivo MTR template of the MBMv2 was transformed into the in vivo myelin-map template of our MBMv3. By concatenating the two transformations (the NM-to-MBMv2 and the MBMv2-to-MBMv3), we accurately converted the neuronal tracing data from the NM template to the MRI template. We then mapped the neuronal tracing data onto the MBMv3 cortical surfaces. For the above registrations, we used the CC similarity metric as the cost functions and three-stage alignments (rigid alignment, affine alignment, and nonlinear SyN transformations), which were also the default options antsRegistrationSyN.sh. An example of registration results is shown in Supplementary Fig. 12.

The instruction and example code for the data preprocessing pipeline is provided via the resource webpage https://marmosetbrainmapping.org/data.html (ReadMe and Codes sections), allowing the user to replicate our protocols.

**Data harmonization across NIH and ION datasets.** We calculated a series of indices to test the data harmonization across different datasets (NIH and ION). They included the single time points SNR, mean images (average across time for one fMRI run), SNR, tSNR (from one fMRI run), Contrast to Noise Ratio (CNR, the mean of the gray matter intensity values minus the mean of the white matter intensity values divided by the standard deviation of the values outside the brain), temporal contrast to Noise Ratio (tCNR, the variance of optimal resting-state fMRI components after ICA contrast to the noisy component), the Fiber (Foreground to Background Energy Ratio: the variance of voxels inside the brain divided by the variance of voxels outside the brain), head motion and the whole-brain functional connectivity across subjects and sessions.

### Functional networks, cortical parcellation and network modeling
**Brain network identification by the Group-ICA.** ICA was performed by the Group-ICA routine of the GIFTI software (https://trendscenter.

org/software/gift/) to identify the brain networks using a number of different component settings. First, preprocessed data without bandpassing and regression of nuisance covariates were group-ICA analyzed with increasing numbers of ICA components from 20 to 80 in a step of 10. We tested the reliability of different ICA methods, including the default "Infomax" ICA algorithm or "ICASSO" group-ICA method, on different datasets (the NIH dataset, the ION dataset, or combined both datasets) and obtained consistent results regardless of the ICA setting or dataset used. Finally, every resulting component from Group-ICA analyses was visually inspected and sorted according to its neuroanatomical features. Since the sorted elements were highly consistent across different settings of ICA-component numbers, we selected the best component to represent every labeled network. We also did manual correction before creating the final network parcellation; for example, when the left and right parts of the same network were separated into two components, we merged them into one network by averaging their maps. We identified 15 cortical resting-state networks from the group-ICA analysis (Fig. 2A–O and Supplementary Fig. 5).

We combined the 15 cortical networks from Fig. 2A–O according to their normalized Z scores from ICA to create a cortical-network parcellation. The details included (1) the combination of networks according to their spatial locations; (2) if they had spatial overlapping, we took the highest value according to their normalized Z scores from ICA; (3) short-range (local) connectivity is usually stronger than long-range connectivity, so the single map cannot cover long-range connectivity due to the spatial overlapping. Therefore, we created a second map to cover the components with long-range connectivity that were missed in the first map. We repeated the above step but only applied to networks with long-range connections (such as Fig. 2I–K) to obtain the second map. The primary map (Fig. 2P, Q, top rows) was mostly contributed by the short-range networks (i.e., Fig. 1G, L, I, O), and the second one (Fig. 2P, Q, bottom rows) was to cover the long-range connectivity that was not captured by the primary map.

Since the human connectome project released a pipeline for denoising by ICA-FIX, it has recently been applied to some animal research studies. Therefore, we also explored whether adopting this processing would affect the identification of brain networks. Before using ICA-FIX, we had to create a training dataset as a standard. For this, we ran first-level ICA on each fMRI run, randomly selected 24 runs from 24 animals (12 runs from the ION dataset and 12 runs from the NIH dataset), and manually classified the noise components based on their spatial patterns and power spectrums. Since the recommendations for the use of ICA-FIX suggest a training dataset of at least 10, we created a total of 24 datasets (trainingMBMv4.RData) to improve significantly our ICA-FIX classifier, which included both the ION dataset (trainingION.RData, using 12 ION training datasets) and the NIH (trainingNIH.RData, using 12 NIH training datasets) respectively. Then, a trained-weighted file (trainingMBMv4.RData) was used to clean all fMRI data based on three sensible-value thresholds (5, 10, and 20) since ICA-FIX recommends a threshold in the range of 5–20. Thus, we created three different versions of ICA-FIX cleaned datasets. In addition, we provided the mask files to allow ICA-FIX to work on the marmoset data since ICA-FIX had several steps that use human-default settings and files incompatible with the marmoset data. Regardless of which training dataset was used, we still obtained the same functional network results shown in Fig. 2A–O.

**Boundary map generation.** Following similar procedures to those described previously in a human imaging study[22], the boundary mapping of resting-state functional connectivity data was implemented in the Connectome Workbench and using customized Matlab codes (Mathworks, Natick, USA, Version 2019b; see the scripts in our open resource). First, the time course of every surface vertex for each brain hemisphere of each subject was correlated with every other surface

vertex to make a correlation map. Then, a similarity map was created for every vertex by calculating pairwise spatial correlations between all correlation maps. Thirdly, the first spatial derivative was applied on the similarity map by the Connectome Workbench's function "cifti-gradient" to generate gradient maps for each brain hemisphere of each subject. Next, the gradient maps were averaged across subjects to produce the group gradient maps for each brain hemisphere. Lastly, the "watershed by flooding" algorithm was applied to identify boundaries in the gradient maps.

**Test-retest evaluation of the boundary map.** To compare the reliability of the boundary maps between the ION and the NIH datasets (Fig. 3A and Supplementary Fig. 8), between the individuals (Fig. 3C–E), and between runs from the same individual (Fig. 3E), we first thresholded two resulting boundary maps for each hemisphere to retain the cortical vertices most likely to be boundaries (i.e., retaining the top quartile of boundary values for a cumulative probability of 0.75) and assessed the overlap of the two thresholded boundaries by calculating the Dice's coefficient. The Dice's similarity coefficient of two thresholded boundaries, A and B, is expressed as:

$$\text{dice(A,B)} = 2 * \left| \frac{\text{intersection(A,B)}}{|A| + |B|} \right| \tag{1}$$

The average Dice's similarity coefficient is the mean of Dice's similarity coefficients across hemispheres.

**Cortical parcellation based on the population-level boundary map.** The creation of parcels was implemented by the customized Matlab scripts (see our open resource). Firstly, based on the vertices with values smaller than their neighbors that were <5 vertices away, we identified all local minima of vertices on the boundary map as seeds for parcel creation. Then, the parcels were grown from these seeds using the "watershed algorithm" procedure as above, allowing them to expand outward from the seed until they met other parcels. Because the whole process depends on the number of seeds for parcel creation, this might result in a large number of parcels. Therefore, according to the performance, we manually defined a threshold for merging adjacent parcels, which is the 60th percentile of the values in the boundary map[22]. It means that any two adjacent parcels with an average value below this threshold were considered not sufficiently dissimilar and should be merged. Finally, according to the population-level boundary map, we visually examined the remaining parcels to identify those needing further adjustment, including eliminating vertices and spatial smoothing. The detailed manual processings for the post-optimization included (1) manually adjusting the parcel borders, (2) manually correcting wrong areal attributions of the region growing, and (3) spatial smoothing of the parcel borders by 8-neighbor vertices. We finally found the resulting cortical parcellation with 96 functional parcels in each hemisphere as our MBMv4 in Fig. 4B.

**Evaluation of cortical parcellation by the DCBC.** Following a previous study[26], we used the DCBC as a metric to evaluate functional boundaries between our parcels. The rationale for this method is that any two points belonging to any given parcel should have more similar functional profiles than those belonging to different parcels. Furthermore, because the functional organization varies smoothly, the correlation between two points will weaken with increasing spatial distance. Thus, we calculated the correlation coefficients for all pairs of points separated by a specific surface Euclidean distance, using 0.5-mm spatial bins (same as fMRI imaging spatial resolution) ranging from 0 to 4 mm for pairs of points residing within parcels or across different parcels (between). The DCBC defines the difference between the within-parcel and between-parcel pair correlations. A higher DCBC reflects that pairs within the same region are more functional, serving

as a global parcellation measure. For the group comparison across atlases (Fig. 4C), the DCBC metrics were calculated for each participant in each spatial bin and then averaged. For the same participant comparison across atlases (Fig. 5D), the DCBC metrics were calculated for each session in each spatial bin and then averaged.

**Comparison with alternative atlases.** We compared our parcellation against alternative digital parcellations created by various approaches. These alternative parcellations included: (1) Paxinos atlas[68], the most commonly used atlas in marmoset brain research, which is cytoarchitectonic characterization by immunohistochemical sections, and here we used its 3D digital version;[15,27] (2) RIKEN atlas:[28] The atlas is cytoarchitecture based on Nissl-staining contrast. (3) The first atlas version of Marmoset Brain Mapping (MBMv1):[15] The borders were delineated based on the high-resolution diffusion MRI contrast and parcellated by a structural connectivity-based approach.

**Deep-learning-based individual parcellation generator.** The group-average parcellation described in the preceding sections is desirable for generating parcellations of individual animals. Although applying our group-level parcellation to individual animals is feasible, as demonstrated in a previous human study[22], we still found misalignments between individuals and cannot be highly consistent with the tendency of the group-average parcellation (MBMv4) when the scanning runs are limited (Fig. 3D, E). Therefore, inspired by previous works[29,30], we trained a multi-layer deep learning network to classify parcels based on the fingerprints from MBMv4. There were two assumptions for this approach: (1) We assumed that individual cortical parcels were close to the group definition after the feature-based surface registration; (2) We assumed that every identified cortical parcel should be in a single class which was the combination of the target parcel and its spatially adjacent parcels (the "searchlight" for the candidate parcel). Thus, the setup of the classifier network was straightforward. Its architecture was as follows (for the graphic reference, see Fig. 5A): for each of the 96 parcels in each hemisphere, a multi-layer deep neural network was designed, which comprised three layers (one input, five hidden, and one output) and 384 hidden neurons (a reasonable compromise between accuracy and training speed for the classification). The whole-brain fingerprint of the candidate parcel from the MBMv4 worked as the training set for the network to classify whether or not each vertex in an individual ROI contained the parcel plus all of its neighbor parcels. Because of the spatial overlap of the "searchlight," we excluded the vertices belonging to multiple parcels. Then, we applied the same procedure of parcel creation as above, meaning that the borders of each identified parcel became the seeds to expand outward until they met other parcels using the "watershed by flooding" procedure. The whole process of individual parcellation was automatic and implemented using customized MATLAB codes (example codes are shared via www.marmosetbrainmapping.org/data.html) combined with the MATLAB Deep Learning Network toolbox.

**Evaluation by task-activation pattern.** We examined the functional relevance of the borders by evaluating the parcels contained within the fMRI activation pattern to a visual-task (Fig. 6A) from our previous study[32]. A subset of animals from the NIH dataset participated in the visual-choice task, which consisted of watching 20-s-long movies (visual field was 10 deg × 8 deg) and 16s resting periods (206 trials for marmoset-ID15 and 280 trials for marmoset-ID25). We performed a contrast comparison between the movie-presentation blocks and the resting blocks to generate visual-task-activation statistical maps for each session. A mixed-effects analysis was then applied to all statistical maps across sessions by the 3dMEMA command of AFNI to obtain a final statistical map. The map was

thresholded at a voxel-wise threshold of $p < 0.05$ and a cluster-wise threshold of $p < 0.05$ for multiple comparison corrections. To compare the similarity of the activation map and the parcellations in each hemisphere (for results, see the flat maps in Fig. 6A) we calculated the shortest Euclidean distance of every vertex/voxel in the boundary of the activation map to the vertexes/voxels in the boundary of parcels/regions from different parcellations. We considered the parcellation with the overall shortest distances of every vertex/voxel in the boundary of the activation map as the best border consistency (for results, see the scatterplots in Fig. 6A).

**Evaluation by functional connectivity gradient spectrum.** It is widely accepted that the cerebral cortex of multiple species, including both human and macaque primates, is organized along principal functional gradients that provide a spatial framework for the co-existence of multiple large-scale networks operating in a spectrum from unimodal to transmodal functional activity[34,69]. Therefore, if the MBMv4 parcellation created here accurately represents the functional organization of the marmoset cortex, we can presume that it will also reveal these principal functional gradients. Thus, as in previous studies[34,70], we followed a workflow for gradient identification: we first computed the rs-fMRI functional connectivity (RSFC) based on MBMv4. Next, the RSFC matrix $M_{x,y}$ with the same size as the atlas was made sparse (to a 10% sparsity), and a similarity matrix $A_{x,y}$ with the normalized angle was computed according to the following equation:

$$A(x,y) = 1 - \frac{\cos^{-1}(\text{cossim}(x,y))}{\pi} \quad (2)$$

Next, the similarity matrix was decomposed via Laplacian transformation into a set of principal eigenvectors describing the axes of most significant variance using the following equation:

$$Lg = \lambda Dg \quad (3)$$

Where $D_{x,y} = \sum_y A(x,y)$, L is the graph Laplacian matrix, and the eigenvectors g which corresponds to the m smallest eigenvalues $\lambda_k$ are used to build the new low-dimensional representation:

$$\varsigma_{LE} = [g_1, g_2, \ldots, g_m] \quad (4)$$

Finally, the first two axes $g_1, g_2$ of each parcel were plotted in 2D space. Meanwhile, we used the scores $g_1$ to sort the functional connectivity matrix (for results, see the heatmaps in Fig. 6B).

**The whole-brain modeling for the link between structural connectivity and functional connectivity.** As we know, structural connectivity and functional connectivity are closely related to each other. Therefore, the lack of structural evidence generally implies biological implausibility for functional connections. Testing whether the cortical parcels created above MBMv4 are accurate representations of the functional areas in the cerebral cortex requires investigation of the underlying structural connectivity. A computational model is a powerful approach to bridge structural and functional connectivity[69,71–74]. In the present study, we implemented a whole-brain model with only two free parameters from previous studies[35,36], as outlined below (for a graphic reference, see Fig. 7A, note that the fMRI data for the modeling part is frequently unfiltered, so the model used the full band of frequency):

According to the whole cortical parcellations (192 total parcels, 96 per hemisphere from MBMv4 or 232 total regions, 116 regions per hemisphere from the Paxinos atlas), the structural connectivity between parcels/regions $C_{i,j}$ was estimated from the structural datasets (see examples in Fig. 7B, C, E, F), either DTI data (in vivo or ex vivo) or the neuronal tracing data. Then, the local dynamics for every parcel/

region j can be properly approximated to the normal form of a Hopf bifurcation:

$$\frac{dz_j}{dt} = \left[a_j + iw_j\right]z_j + z_j|z_j|^2 \tag{5}$$

In this equation, $z_j$ is a complex-valued variable $z_j = x_j + y_j$, and $w_j$ is the intrinsic signal frequency of parcel/region j, which ranged from 0.04–0.07 Hz and was determined by the averaged peak frequency of the bandpass-filtered fMRI signals of the parcel/region j[36,75–78]. $a_j$ is a bifurcation free parameter controlling the dynamics of the parcel/region j. For $a_j < 0$, the phase space presents a unique stable and is governed by noise. For $a_j > 0$, the phase space presents the stable state, giving rise to a self-sustained oscillation. For $a_j \approx 0$ the phase presents an unstable state, switching back and forth and giving rise to a mixture of oscillation and noise.

The coordinated dynamics of the resting-state activity for parcel/region j could be modeled by coupling determined by the above structural connectivity $C_{i,j}$. To ensure the oscillatory dynamics for $a_j > 0$, the structural connectivity $C_{i,j}$ should be normalized and scaled to 0.2 in a weak coupling condition before starting the simulation. The coupled differential equations of the model are the following:

$$\frac{dx_j}{dt} = \left[a_j - x_j^2 - y_j^2\right]x_j - w_j y_j + G\sum_i C_{i,j}\left(x_i - x_j\right)\beta\eta_j(t) \tag{6}$$

$$\frac{dy_j}{dt} = \left[a_j - x_j^2 - y_j^2\right]y_j + w_j x_j + G\sum_i C_{i,j}\left(y_i - y_j\right)\beta\eta_j(t) \tag{7}$$

In this equation, G is another free parameter representing the fixed global coupling factor that scales structural connectivity $C_{i,j}$. $\eta_j$ represents additive Gaussian noise in each parcel/region and is scaled by a factor β fixed at 0.04 according to previous studies[36]. Euler-Maruyama algorithm integrated these equations with a time step of 0.1 seconds to accelerate simulation[79].

The free bifurcation parameter $a_j$ for parcel/region j could be locally optimized based on fitting the spectral information of the empirical BOLD signals. To achieve this, we filtered raw empirical BOLD data in the 0.04–0.25 Hz band and calculated the power spectrum $p_j(f)$ for each parcel j as below:

$$p_j = \frac{\int_{0.04}^{0.07} p_j(f)df}{\int_{0.04}^{0.25} p_j(f)df} \tag{8}$$

and updated the local bifurcation parameter $a_j$ by a gradient descendent strategy:

$$a_j = a_j + \eta\left(p_j^{empirical} - p_j^{simulated}\right) \tag{9}$$

We applied the above optimization process to receive the best bifurcation parameters $a_j$ of every parcel/region defined in the parcellations. Once we found the optimized set of bifurcation parameters $a_j$, we adjusted the free parameter G within the range of 0–8 in steps of 0.1 according to a reasonable compromise from previous studies[36,79] to simulate the same number of sessions for each animal and the same number of animals. To compare the performance in different atlases, we just needed to compare fitting (similarity) metrics, Pearson's correlation coefficient between the simulated functional connectivity and the one used for the empirical data, when we fixed the same value of parameter G.

Since the distributions of the optimal bifurcation parameter a are identical in different parcellations MBMv4 or Paxinos atlas (see Supplementary Fig. 13, one-way ANOVA $F_{(1,11986)} = 9.09$, $p = 0.26$), we selected the best free parameter G from the Paxinos atlas for comparison performance with our MBMv4 (see results in Fig. 7B, E, examples in Fig. 7C, F). Moreover, we also selected the group-averaged

functional connectivity from all individuals as the empirical observable for the ultra-high-resolution diffusion MRI and neuronal tracing datasets and the individual functional connectivity for the corresponding in vivo diffusion MRI.

The resulting simulation of functional connectivity demonstrated the influence of the distance, especially for the diffusion MRI (see Fig. 7D). To decrease this impact, we added an EDR for a more accurate estimation of structural connectivity before the simulation, which can be implemented as follows: we first normalized the structural connectivity estimated by diffusion tractography to 0–1, then calculated the probability of structural connectivity according to the EDR rule (p($d$) = ce$^{-\lambda d}$, $\lambda \approx 0.3$, $c \approx 0.94$ where $d$ is the distance[40]) and normalized to 0–1. Finally, we transformed the normalized structural connectivity to match the probability of structural connectivity. For simplicity to help us to identify abnormal values, we defined a threshold as the Median Absolute Deviation for the normalized structural connectivity

$$\hat{\sigma} = \frac{Median(|X_i - \widetilde{X}|)}{0.6745} \tag{10}$$

Where $X_i$ = each value, $\widetilde{X}$ = average value

If any values were larger than this threshold, we considered them abnormal values, and the corresponding probability of structural connectivity would replace them to reduce the impact of distance.

### Reporting summary

Further information on research design is available in the Nature Portfolio Reporting Summary linked to this article.

## Data availability

All of our datasets, including raw and preprocessed NIH and ION resting-state fMRI, diffusion MRI, and neuronal tracing datasets, are available on our specific webpage of the Marmoset Brain Mapping Project (www.marmosetbrainmapping.org/data.html). The volume data are in NIFTI format, and the surface data are in CIFTI format. The raw MRI data without processing is provided in the standard BIDS format for cross-platform sharing. The MBMv4 parcellations are also provided on the same webpage (www.marmosetbrainmapping.org/data.html), and the MBMv1 and Paxinos parcellations on our MRI template space are a part of the MBMv3 resource (marmosetbrainmapping.org/v3.html). The high-resolution ex vivo diffusion MRI data are a part of the MBMv2 resource (marmosetbrainmapping.org/atlas.html#v2). The raw neuronal tracing data are from Marmoset Brain Connectivity project (https://www.marmosetbrain.org/reference). Note that the MBMv4 datasets are only available for scientific purposes and are licensed under Creative Commons Attribution-NonCommercial-ShareAlike (CC BY-NC-SA 4.0). Source data are provided with this paper.

## Code availability

The codes for analyzing pipelines used in this study could be available on our webpage (www.marmosetbrainmapping.org/data.html). The codes package with examples includes the generation of cortical functional boundary maps, functional parcellation, and their evaluation metrics, as well as computational modeling which could link with structural imaging and neuronal tracing. Note that the codes and analyzing pipelines are only available for scientific purposes and are licensed under Creative Commons Attribution-NonCommercial-ShareAlike (CC BY-NC-SA 4.0).

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

## Acknowledgements

We thank Kaiwei Zhang and Binshi Bo at the 9.4T core facility (CEBSIT) for assistance in the data collection of the ION data, Lisa Zhang for the assistance in the data collection of the NIH 7T data, the Marmoset Animal Facility of CEBSIT for animal care, Xiaojia Zhu for the assistance in organizing MRI data in BIDS format, and the NIH Fellows Editorial Board for the editorial assistance on the early version of the manuscript. The study was supported by the grants from National Science and Technology Innovation 2030 Major Project of China (2022ZD0205000, 2021ZD0203900), the Pennsylvania Department of Health Commonwealth Universal Research Enhancement (CURE) Tobacco Settlement Appropriation – Phase 18 (Grant SAP4100083102 to ACS), the US National Institute on Aging (Grants R24AG073190 and U19AG074866 to ACS), the Shanghai Municipal Science and Technology Major Project (no. 2018SHZDZX05 to C.L. and Z.F.), the Lingang Laboratory (Grant no. LG-QS-202201-02 to C.L.), the National Natural Science Foundation of China (no. 32171088 to C.L.), the Australian Research Council (DP110101200, DP140101968, CE140100007) and National Health and Medical Research Council (APP1194206) to M.R., National Science Centre (2019/35/D/NZ4/03031 to P.M.), NIH Intramural Research Programs (ZIA NS003041 to A.S. and C.Y., ZICMH002888 to D.G.), International Neuroinformatics Coordinating Facility Seed Funding Grant (to P.M. and M.R.), a Spanish research project funded by the Spanish Ministry of Science, Innovation, and Universities (MCIU), State Research Agency (AEI), and European Regional Development Funds (FEDER) (ref. PID2019-105772GB-I00 AEI FEDER EU to G.D.), HBP SGA3 Human Brain Project Specific Grant Agreement 3 (grant agreement no. 945539 to G.D.), and the EU H2020 FET Flagship program and SGR Research Support Group support (ref. 2017 SGR 1545 to G.D.).

## Author contributions

C.L., X.T., A.C.S., and M.R. designed and supervised the study; Z.L. and Y.C. collected the ION MRI data; D.S., C.L., C.Y., and X.T. collected the NIH MRI data; M.R. and P.M. collected the neuronal tracing data; C.L., X.T., C.T., and Z.L. preprocessed and organized the MRI data; P.M., C.L., X.T., and H.J. preprocessed the neuronal tracing data; C.L. and X.T. constructed the functional-network maps; C.L., X.T., and M.R. constructed and evaluated the cortical parcellation maps; X.T., C.L., Y.P., and G.D. conducted the computational modeling; C.L. and F.F. developed the online web applications; D.G. and C.L. implemented the atlas

 

into AFNI/SUMA. X.T. and C.L. wrote the original draft. C.L., X.T., A.C.S., and M.R. revised the manuscript.

## Competing interests

The authors declare no competing interests.

## Additional information

[1]Department of Neurobiology, University of Pittsburgh Brain Institute, University of Pittsburgh, Pittsburgh, PA 15261, USA. [2]Center for Excellence in Brain Science and Intelligence Technology, Institute of Neuroscience, CAS Key Laboratory of Primate Neurobiology, Chinese Academy of Sciences, Shanghai, China. [3]Laboratory of Neuroinformatics, Nencki Institute of Experimental Biology of the Polish Academy of Sciences, 02-093 Warsaw, Poland. [4]Department of Physiology and Neuroscience Program, Biomedicine Discovery Institute, Monash University, Clayton, VIC 3800, Australia. [5]Center for Brain and Cognition, Computational Neuroscience Group, Department of Information and Communication Technologies, Universitat Pompeu Fabra, Roc Boronat 138, Barcelona 08018, Spain. [6]Universidad de San Andrés, Vito Dumas 284 (B1644BID), Buenos Aires, Argentina. [7]Cerebral Microcirculation Section, Laboratory of Functional and Molecular Imaging, National Institute of Neurological Disorders and Stroke, National Institutes of Health (NINDS/NIH), Bethesda, MD 20892, USA. [8]Department of Neurobiology, Affiliated Mental Health Center & Hangzhou Seventh People's Hospital, Zhejiang University School of Medicine, Zhe Jiang Sheng, China. [9]MOE Frontier Science Center for Brain Science and Brain-machine Integration, Zhejiang University, Hangzhou, China. [10]Scientific and Statistical Computing Core, National Institute of Mental Health, National Institutes of Health (NIMH/NIH), Bethesda, MD 20892, USA. [11]Institució Catalana de la Recerca i Estudis Avançats (ICREA), Passeig Lluís Companys 23, Barcelona 08010, Spain. [12]Department of Neuropsychology, Max Planck Institute for Human Cognitive and Brain Sciences, Leipzig 04103, Germany. [13]School of Psychological Sciences, Monash University, Melbourne, Clayton, VIC 3800, Australia. [14]Shanghai Center for Brain Science and Brain-Inspired Intelligence Technology Shanghai, Shanghai, China. [15]Lingang Laboratory, Shanghai 200031, China. [16]University of Chinese Academy of Sciences, Beijing, China. ✉e-mail: txgxp88@gmail.com; marcello.rosa@monash.edu; afonso@pitt.edu; zliang@ion.ac.cn; crliu@ion.ac.cn

