## [Peer Review File · Nature Communications]

An integrated resource for functional and structural connectivity of the marmoset brainREVIEWER COMMENTS

Reviewer #1 (Remarks to the Author):

This paper provides very important resources of marmoset MRI data including T2w, resting-state fMRI and in-vivo diffusion MRI. The strength of this study is excellent resting state fMRI dataset which are collected from a large sample of animals using two ultrahigh-field MRI scanners. The reviewer appreciates the authors' generosity to make these datasets publicly available and recommend the authors to make all the information including the scanning protocols publicly available. Then the readers can replicate the analysis and results of this study, and benefit from these datasets for future developments of non-invasive connectome & neuroimaging in this species. The reviewer makes critical comments below mostly on the preprocessing and network analysis. The reviewer recommends the authors to focus on appropriately publishing resource data as is (also including all the information of scanning parameters) and explicitly analyzing basic data quality to strengthen the value of the data and make it useful for the scientists in this field.

Major comments:

- 1) The dataset does not include the structural MRI image with high contrast between gray and white matter (e.g., MPRAGE), which is the most important image for cross-species standardization. T1-weighted images are not obtained in any of subjects. The structural normalization of the brain size and shape is done in general using T1-weighted images, which has the best contrast between gray and white matter in MRI images, allowing good alignment between subjects. The authors need to make a rational explanation not to do so and discuss limitation of this study, and possible degradation or bias of the group-wise connectivity.
- 2) The manuscript does not include the basic quality assessment of the resource data. The resource data does not accompany all the information on scanning parameters needed for preprocessing. Since the functional connectivity studies on the resting state fMRI is highly dependent on the quality of the data, explicit data-driven analysis is expected to strengthen this manuscript as an MRI resource paper. No quality assessment (except tSNR) was presented for fMRI, e.g., the contrast-to-noise ratio (CNR), unexpected noise and artifacts, as well as denoising of the fMRI data for the first-level analysis. No frequency and time series data are presented for all the 'neural' like components in group ICA, which are important for confirming the component as neural origin.
- 3) The manuscript does not provide any novel insights on the optimization of neuroimaging preprocessing for this species, which is behind those in the human MRI studies. It should be discussed as a limitation of this study. The higher-level analysis of the connectome depends on the preprocessing of

the data, while no established preprocessing pipeline is presented in this paper. The preprocessing is done just through a mixture of different software packages (AFNI, FSL, Workbench, TORTOISE), all of which are basically optimized for human brain imaging and not for marmoset data. The actual commands and parameter setting are not described in detail in the manuscript nor available at the resource website, so that the readers cannot replicate the analysis to confirm the results of the study. Different packages are used for solving the same imaging problem, which is not likely chosen based on the reasonable arguments and may confuse the readers who would like to find the best solution of preprocessing in this species. For example, the authors used the FSL Topup and TORTOISE for correcting B0-susceptibility induced distortion (i.e. blip-up blip-down method) of rfMRI and diffusion MRI, respectively. This difference can result in the bias of spatial accuracy between the two datasets, and potentially bias the link between functional and diffusion connectivity the authors analyzed.

4) The novelty of this paper is resting-state fMRI datasets from two institutions, as well as in-vivo diffusion MRI in addition to the ex-vivo dMRI that were made publicly available by the same authors previously. Therefore, a novel idea or analysis is expected for charactering and reducing bias, data harmonization and/or statistical harmonization between different datasets. However, nothing was explicitly performed unfortunately. Note that harmonization is a relatively new research interest in the human MRI large population studies (e.g., HCP, HCP, ABCD, UK biobank, Brain/MINDS-beyond) in order to understand subject variability. The explicit analysis for this approach to harmonize data across institutions/scanner and brain preparations (in-vivo vs ex-vivo) may also be valuable for the animal studies, as well as for its future international global neuroscience in NHP (see. PRIMatE Data and Resource Exchange (PRIME-DRE) Global Collaboration Workshop and Consortium, Neuron 2022).

5) The population-based boundary maps are likely similar between the two institutions, but they are not very close to each other. There seems to be dislocation or misregistration of the boundaries over the surfaces. Similarity of the functional boundary maps between hemispheres is also not very high. These findings are not reasonable since less hemispheric asymmetry or smaller intersubject variability are expected in the marmosets than in humans. The reviewer thinks that biases and artifacts are not removed completely for these datasets, thus it is too early to answer these naïve questions on neurobiological or network organization.

6) As for the link to the structural-functional connectivity, it is not understandable why use of neural tracing data did not achieve highest predictability of functional connectivity. What were these predictions if they were separately analyzed for short and long-distance connectivity? Is there anything that Hopf bifurcation neurodynamical function cannot model? High-resolution ex-vivo diffusion MRI data should be excellent as compared with any other diffusion MRI datasets, but it will not be complete nor have supreme SNR for the high b-value data. There are several issues on the accuracy of the diffusion tractography such as crossing vs kissing fibers, gyral bias, inaccuracy for long-distance tract, which are not yet solved completely in any model or algorithm. The neural tracing data using retrograde tracers may be absolutely reliable in terms of strength and specificity of connectivity and by far robust against distance.

Minor comments:

In abstract and many parts of the main text, the authors use a term 'MRI space'. This is a bit sloppy statement since it implies the MRI scanner coordinates, which is not meaningful for understanding the brain. The reviewer recommends the authors to use a more general term, something like 'standardized space of the marmoset brain' or 'marmoset brain template of ear-bar and eye-bar coordinates (MBMv3 or MBMv2)'. The abstract should be more specific about which resource data is shared in this publication. Ex-vivo diffusion MRI and neural tracing data were already made publicly available in the authors' previous publication.

Results

Line 109. For the quality assessment of fMRI between the two sites, the tSNR alone is insufficient at all. The tSNR calculated by simple mean divide standard deviation does show the quality of the images, but not for the quality of signal of interest (i.e., BOLD or neural signal). The variance of the BOLD activity in the resting-state fMRI is less than 10% among all even in the ultrahigh-field scanner. Therefore, for the quality of the fMRI data, analysis of variances obtained from data-driven analysis may provide useful measures including contrast-to-noise (CNR), motion-related variables, high-pass, and other artifacts (e.g., Marcus et al., 2013). For the quality assessment of diffusion MRI, the estimated motion, slice-by-slice outlier detections, CNR (for the diffusion weighted signal changes) are useful for presenting the data quality (e.g., Bastiani et al., 2009).

Line 139. The presented ICA components are very nice (Fig 2, S2). But the results are not showing any temporal features such as frequency spectrum. It is more informative to reveal the frequency spectrum for each component and show how they look like neural origin. It is questionable if the components E, F, P, S in S2 is really neural origin. The regressing ICA components to the unfiltered data can generate the time series data and frequency spectrum. It is also informative to demonstrate the representative noise or artifactual components in the supplements.

Line 150 'Default-model-like network (DMN)' Default mode network? In line 15 in the legend of Figure S2, 'default-network' should be default mode network.

Line 240 'DCBC'. The distance and areas over the surface may be largely different between subjects, which are not well controlled or regularized unless individual surface reconstruction and surface registration were not performed.

Line 335. 'MBMv4 provides 335 a more accurate reflection of the functional parcellation of the cortex than current histology-based atlases' - This is a bit sloppy statement, and any single modality of the brain metrics (including functional connectivity) may not provide perfect resolution of the cortical parcellations. For example, the histology data is more accurate than functional parcellations in some of the boundaries (e.g. V1/V2).

Data acquisition:

The functional MRI acquisition protocols are not well harmonized between two scanners (ION vs NIH). There are many different parameters that significantly affect the quality of the data including static magnetic field, gradient strength, which may cause difference in SNR, tissue T1 and T2* value, B0 inhomogeneity & distortion. The transmission coil is also different between two institutions, causing substantial difference in B1+ field, and hence bias the homogeneity of the signal, SNR across space etc. Difference in B0 field and in TE of fMRI (18ms vs 22.2ms) should cause the differences in CNR (contrast-to-noise ratio) of the fMRI data, which is not clearly and carefully investigated. There is no way for the readers to estimate how B0 inhomogeneity-induced distortion is look like between two institutions, unless the original (distorted) image or estimated shift images are not presented. Distortion correction method, Topup or TORTOISE, is optimized for human brain by default, thus the authors should carefully describe if it was indeed effective enough for correcting distortion of the fMRI in the small brain. There is no information available for detailed scanning parameters including dwell time, which is needed for B0-susceptibility induced distortion correction with a blip-up blip-down method. Since the resource data does not include any detailed info on imaging parameter (including json files usually stored in the BIDS format), so the data cannot be properly analyzed or replicated by the other users. The previous public data for ex-vivo diffusion MRI (V1 and V2) does not also provide all the information on the scanning parameters including dwell time.

The fMRI preprocessing is done by legacy method and likely not optimized for this species. Registration is an important treatment in the preprocess, and is highly dependent on the resolution of the data and object size, therefore it is obvious that the marm. The quality assessment of rfMRI is not carefully done. There are no descriptions on the animal's compliance during scanning and whether their head was moved or not during scanning. No explicit artifact or noise removal is done in the first-level analysis, which is commonly done in recent human resting-state fMRI studies. Bandpass filtering (0.01 – 0.1Hz) in the initial preprocessing (line 620) is a bit outdated approach, since it loses the significant neural signals at >0.1 Hz. The resampling of the fMRI data to the standard space (line 624) is not done by concatenating all the transformation at a single step and likely excluding the distortion correction by Topup. No exploratory data-driven analysis is applied, thus there is no way for the readers to see how much the rfMRI data is contaminated by any unexpected artifacts. The resting-state of subjects is hard to control the conditions of subjects particularly in animals. Therefore, data-driven approach like ICA is strongly recommended in the first-level analysis for estimating quality of the data, then needs to be applied for artifact removal. As long as the authors do not apply this approach, the group-wise ICA analysis should also be contaminated by a large number of unexpected artifacts from each scan. It is

strange that the authors apply a different or wider range of frequency in the post process analysis (0.04-0.07Hz in Line 858, 0.04-0.25Hz in Line 879).

Surface mapping of the volume data (e.g. fMRI) is likely not performed using the subject's surface, thus is potentially bias the results of a large group-based functional connectivity. Surface mapping of the subject's volume onto the average or the other subject's surface should not be very precise given the fact that there is a significant intersubject variability of brain size and shape of this primate species as compared with rodents. There is no detailed explanation of the volume-to-surface mapping (line 629). How were the surfaces created and how they are well aligned with the subject's fMRI and T2-weighted images?

Diffusion MRI. There are no careful descriptions whether the package of TORTOISE is used as is or optimized for the marmoset data. The image registration may be dependent on a global optimization technique that relies on the resolutions during iterative aligning processes. The optimality of the distortion correction by the blip-up and blip-down method may need to be carefully described. Nothing was described about what kind of tractography method was applied (line 639).

Neural tracing data: the mapping method of the neuronal tracing data onto the MTR template is not very clear. How accurate was the registration between the neuronal tracing data and MTR template? There are no visual presentations nor quantitative assessment on the validity of the registration. In particular, it is informative to present whether cortical boundaries of the tissue sections were properly aligned with the MRI template. No information is available about which cost function and registration method (linear vs nonlinear, DOF) was applied.

Group-ICA. Please properly describe which of NIFTI volume or CIFTI data the authors used for their group-wise ICA analysis using GIFT. Also please clarify if each scan data was normalized by the variance and its mean signal was removed before feeding into group ICA. The classification of the components (neural or noise) is likely done only based on the spatial distribution. This is a bit tricky because some of components even located within the brain may be associated with pulsation of the brain tissue (i.e., periodic movement). Frequency spectrum and time series signal may be presented for each component.

Line 850. 'while brain parcellations'. I'm wondering if the authors would really discuss whole brain parcellations including cortical and subcortical parcellations.

Line 851. '232 brain regions per hemisphere from Paxinos Atlas'. Is this number correct? Does this number include both cortical and subcortical structures?

Reviewer #2 (Remarks to the Author):

Review of “An integrated resource for functional and structural connectivity of the marmoset brain” by Tian and colleagues.

The authors provide a huge comprehensive data set acquired in marmosets encompassing awake resting state fMRI, in-vivo DTI, high resolution ex-vivo DTI and a large collection of neuronal tracing data. Moreover, the authors relied on this massive data set, in combination with deep learning and computational modelling, to propose a parcellation scheme of the cortex and to link anatomical properties with functional connectivity measures.

This is an exquisite data set that will become an important tool for many nonhuman and human primate researchers which will also improve homology research. Despite the complexity of the data and some of the analyses, the authors were able to summarize them in a succinct and very accessible manner. The analyses are state-of-the-art. I highly support publication of this manuscript and only have a few questions and/or suggestions which may improve the manuscript:

1. Fig. 2A-O: I am surprised by the ‘locality’ of the different functional networks. In humans and macaques, resting state based functional networks typically consist of multiple anatomically segregated subunits. In this case, mainly singular nodes are apparent (except for panel J?). How is this explained?

From an esthetic point of view: why is the background in panels 2I-O grey? I would use the same background in all panels.

2. Fig 2P, Q: It is unclear how the two network-parcellation maps are created (this was also not clear from the methods section) and more importantly how they relate to each other.

3. Line 200-207. I’m puzzled by the seemingly higher intra-subject than inter-subject variability. Given the fixed anatomy (within a subject), which may differ from that in other subjects, this is difficult to understand. Moreover, this also poses problems for ‘within’ subject research -e.g. where one aims to relate boundary maps with task-based information, injections, electrophysiological recordings, etc. Please discuss.

4. 229: Please explain exactly what is meant by ‘semi-manual optimization’. This was also unclear from reading the methods section. This may lead to reproducibility issues. Please discuss what exactly is done manually and whether this imposes methodological limitations.

5. Fig 6A and B (scatter plots). It is unclear what exactly is plotted. Also, what is the color (MBMv4) and grey-scale (Paxinos) mean in panel B?

6. In several locations (e.g. line 379) the authors mention that MBMv4 offers an alternative view on understanding the functional connectivity of a brain. This is a crucial statement. What exactly is meant? How can one relate both views? Is one ‘better’ (ground truth?) than the other? Is it reflecting the difference between anatomical versus functional connectivity? Or a difference in ‘sampling efficiency’? Or? Please discuss in depth.

7. Related to the previous point: Different parcellations can be obtained with different measures (e.g. rs functional connectivity, anatomical connectivity, cyto-and myelo-architectonics, etc..). It would be highly

informative if a few obvious parcels where all measures match and where measures don't match are described/shown in more detail (not using the whole-brain overviews).

8. The study revealed more functional networks and more parcels than previous marmoset studies. Does one expect these numbers to grow when higher spatio-temporal resolution data can be acquired in the future? In other words, to what extent are these parcels dependent on the type of measurements? This is important from a theoretical point of view as one (implicitly) creates the impression that the current processing modules are the ones determining cortical functioning. But what if one misses the really important modules (e.g. columnar like) simply because of current technical limitations?

9. Line 506: I guess inter-subject is meant instead of intra-subject?

10. Line 533: sentence is not clear.

11. Line 724: why is the 60th percentile used?

12. Line 793: Please explain the visual-choice task. I guess this was not a free-viewing movie task?

Dear reviewers:

We thank all the reviewers and the editor for their strong encouragement and support of our manuscript, and especially for providing insightful comments and suggestions, which we believe have significantly improved the quality of our publication. We conducted new analyses and revised the paper accordingly in response to the comments provided. We addressed all the reviewers' comments fully.

In what follows, we provide point-by-point responses to each reviewer's comments. The review comments are in black font, and our responses are in blue font. The resulting edits of the original manuscript are highlighted in red font in the revised manuscript.

REVIEWER COMMENTS

Reviewer #1 (Remarks to the Author):

This paper provides very important resources of marmoset MRI data including T2w, resting-state fMRI and in-vivo diffusion MRI. The strength of this study is excellent resting state fMRI dataset which are collected from a large sample of animals using two ultrahigh-field MRI scanners. The reviewer appreciates the authors' generosity to make these datasets publicly available and recommend the authors to make all the information including the scanning protocols publicly available. Then the readers can replicate the analysis and results of this study, and benefit from these datasets for future developments of non-invasive connectome & neuroimaging in this species. The reviewer makes critical comments below mostly on the preprocessing and network analysis. The reviewer recommends the authors to focus on appropriately publishing resource data as is (also including all the information of scanning parameters) and explicitly analyzing basic data quality to strengthen the value of the data and make it useful for the scientists in this field.

OUR RESPONSE: Thank you very much for your support and encouragement of our work.

We have now introduced an extensive analysis and made significant edits to the manuscript directly addressing your comments.

Major comments:

1) The dataset does not include the structural MRI image with high contrast between gray and white matter (e.g., MPRAGE), which is the most important image for cross-species standardization. T1-weighted images are not obtained in any of subjects. The structural normalization of the brain size and shape is done in general using T1-weighted images, which has the best contrast between gray and white matter in MRI images, allowing good alignment between subjects. The authors need to make a rational explanation not to do so and discuss limitation of this study, and possible degradation or bias of the group-wise connectivity.

OUR RESPONSE: Thank you very much for your suggestions.

In humans, due to the complicated gyrification, the gray-white contrasts provided by the T1w become essential for an accurate alignment, and it is more common to use T1w rather than T2w for segmentation and cross-subject registration.

However, in rodents and marmosets, the T2w image is more commonly used than the T1w image for the following reasons:

- 1) Similar as rodents, the marmoset has a smooth brain and is free from the registration inaccuracy caused by complex gyrification. Thus, the gray-white contrasts provided by T2w image is sufficient for accurate cross-subject registration.
- 2) Compared with T1w images, the T2w contrast is more similar as the T2*w contrast of fMRI data, allowing more accurate registration.
- 3) MP-RAGE sequences are not commonly used for T1-weighted structural imaging in ultra-high field (UHF, $\geq 7\text{T}$) MRI. With higher fields, the T1 relaxation time increases in both gray and white matter, leading to relatively smaller gray and whiter matter T1 differences¹⁻³, e.g., $T_1^{\text{WM}} / T_1^{\text{GM}} = \sim 1700/2000$ at 9.4T compared to $T_1^{\text{WM}} / T_1^{\text{GM}} =$

~800/1400 at 3.0T⁴⁻⁷. Below we provide a figure to show a simulation for the point (note that GM-WM signal contrast on 9.4T is much worse than that of 3.0T). MP-RARE signals were simulated using the algorithm in Kir et al.⁸. Therefore, it is commonly known that T1 weighted sequences like MP-RAGE is not an optimal and efficient way to acquire gray and white matter contrast at UHF animal MRI.

- [1] Graaf et al, *Magn Reson Med* . 2006 Aug;56(2):386-94.
- [2] Kuo et al, *J Magn Reson Imaging* . 2005 Apr;21(4):334-9.
- [3] van de Ven et al, *Magn Reson Med* . 2007 Aug;58(2):390-5.
- [4] Rooney et al, *Magn Reson Med* . 2007 Feb;57(2):308-18.
- [5] Stanis et al, *Magn Reson Med* . 2005 Sep;54(3):507-12.
- [6] Zhu et al, *Magn Reson Med* . 2005 Sep;54(3):725-31.
- [7] Gelman et al, *Magn Reson Med* . 2001 Jan;45(1):71-9

[8] Kir et al. *J Magn Reson Imaging* . 2012 Sep;36(3):748-55.

As above, it is common in rodent and marmoset neuroimaging (under high field MRI scanner) to use T2w image for spatial normalization; and the missing T1w contrast won't affect the alignment between marmosets.

Nevertheless, we fully agreed with the reviewer's points of the importance of T1w image for cross-species study. Although this paper is about a marmoset rs-fMRI resource, people would be interested in using it for comparative studies. Furthermore, since human neuroimaging adopted both T1w and T2w for structural normalization, the missing T1w image in our marmoset resource might degrade the cross-species standardization.

In addition, with the T1w image, the myelin map (T1w/T2w) can be calculated, which is an important contrast for gradient-based parcellation. We discussed the limitation in the revised manuscript (line 549-553):

“Fifth, our parcellation only used the resting-state functional connectivity information, as in many human studies^{21,28,40}. However, more advanced approaches incorporated structural contrasts, especially the T1w/T2w myelin map and multiple task-fMRI data for multi-modal brain parcellation. Thus, combining more image modalities to improve the parcellation of the marmoset brain becomes important in the future.”

2) The manuscript does not include the basic quality assessment of the resource data. The resource data does not accompany all the information on scanning parameters needed for preprocessing. Since the functional connectivity studies on the resting state fMRI is highly dependent on the quality of the data, explicit data-driven analysis is expected to strengthen this manuscript as an MRI resource paper. No quality assessment (except tSNR) was presented for fMRI, e.g., the contrast-to-noise ratio (CNR), unexpected noise and artifacts, as well as denoising of the fMRI data for the first-level analysis. No frequency and time series data are presented for all the 'neural' like components in group ICA, which are important for confirming the component as neural origin.

OUR RESPONSE: Thank you very much for your suggestions.

In the revised manuscript, we provided the following more quality assessments, including:

- a) As the reviewer suggested, we calculated more comprehensive QA measurements (tSNR, CNR, and head motions) for each run. In addition, for data harmonization purposes, we systematically compared the data quality between ION data and NIH data. The two datasets were similar in the head motion or SNR/tSNR/CNR. These figures of QA comparison across datasets are included in the revised Supplementary Materials (Fig. S2 and Fig. S3):

Figure S2. Similar quality measurements of the ION and the NIH datasets. (A) The raster plots and their histograms present the CNR (Contrast to Noise Ratio: the mean of the gray matter intensity values minus the mean of the white matter intensity values divided

by the standard deviation of the values outside the brain) and the Fiber (Foreground to Background Energy Ratio: the variance of voxels inside the brain divided by the variance of voxels outside the brain) of two datasets (the blue represents the results from the NIH dataset, and the yellow represents the results from the ION dataset); the results of the Wilcoxon rank test between two datasets (N-NIH =180 N-ION =172) are $p=0.45$ and $p=0.11$, respectively. (B) The raster plots and their histograms present the average SNR, median SNR and max SNR, average tSNR, median tSNR and max tSNR of the cortical gray matter from two datasets (N-NIH =180 N-ION =172). The Wilcoxon rank tests for SNR are $p=0.259$ $p=0.824$ and $p=0.968$; and for tSNR are $p=0.435$ $p=0.625$ and $p=0.2$, respectively. (C) presents the average SNR, median SNR and max SNR, average tSNR, median tSNR and max tSNR of cortical white matter from two datasets (N-NIH =180 N-ION =172). The Wilcoxon rank tests for SNR are $p=0.712$ $p=0.32$ and $p=0.42$; and for tSNR are $p=0.062$ $p=0.086$ and $p=0.908$, respectively. The above QA measurements show no significant differences between the NIH and the ION datasets.

Figure S3. Head motions of the ION and the NIH datasets. (A) the top panel presents head-motion (weighted euclidean norm of six motion parameters) across timepoints of different datasets (the blue represents the NIH dataset, and the yellow is the ION dataset). Each dot is the head-motion measure of each fMRI at a one-time point. The bottom panel

presents the histogram statistics from each dataset (error bar represents 95% confidence interval), which indicates head-motion levels are similar across datasets. (B) presents the percentage of censored time points (motion > 0.2mm and temporal outlier > 0.1) for each fMRI. Most animals and fMRI runs (710 runs) have low head-motion and censored time points, suggesting the effectiveness of our head-constrained and training approaches. Note that the three fMRI runs with extensive head motions (more than 10% time points were censored) were excluded from our analysis, and the total number of valid runs (710) was reported in our manuscript, although we included the three runs in the release of source (raw) data.

b) In addition to what the reviewer suggested, we also provided QA reports generated by the “APQC” function of AFNI for the fMRI data and the “DIFFPREP” function of TORTOISE for the dMRI data. Both functions are used as default quality control assessment, providing more user-friendly formats (for example, HTML web page) to visualize the data and data quality (including the head motion plots). The reports are available for download in the revised resource.

An HTML report example is shown below:

c) We provided each component's time series and frequency power plots for the ICA analysis in the supplementary Figure S5. All the components show neural-like patterns spatially (all peaks are in the cortical or subcortical gray matter) and temporally (no patterns of artifacts or noises).

3) The manuscript does not provide any novel insights on the optimization of neuroimaging preprocessing for this species, which is behind those in the human MRI studies. It should be discussed as a limitation of this study. The higher-level analysis of the connectome depends on the preprocessing of the data, while no established preprocessing pipeline is presented in this paper. The preprocessing is done just through a mixture of different software packages (AFNI, FSL, Workbench, TORTOISE), all of which are basically optimized for human brain imaging and not for marmoset data. The actual commands and parameter setting are not described in detail in the manuscript nor available at the resource website, so that the readers cannot replicate the analysis to confirm the results of the study.

Different packages are used for solving the same imaging problem, which is not likely chosen based on the reasonable arguments and may confuse the readers who would like to find the best solution of preprocessing in this species. For example, the authors used the FSL Topup and TORTOISE for correcting B0-susceptibility induced distortion (i.e. blip-up blip-down method) of rfMRI and diffusion MRI, respectively. This difference can result in the bias of spatial accuracy between the two datasets, and potentially bias the link between functional and diffusion connectivity the authors analyzed.

OUR RESPONSE: We agree with the reviewer that the preprocessing pipeline is essential for resting-state fMRI analysis. As the reviewer said, most existing software is optimized for human data, and thus we had to combine different software packages to achieve the best outcome for the following reasons:

- 1) We chose to use AFNI for the main preprocessing stream of the resting-state fMRI. Compared with FSL and SPM-based packages, AFNI provides the most animal-friendly commands, while FSL and SPM-based packages are mainly optimized for human neuroimaging data. For example, we were not able to achieve satisfied results with FSL ICA-FIX for denoising (which is a standard pipeline for HCP), thus we used the traditional regression method to denoise the data. AFNI is also written by an NIH-based team residing in the same NIH campus) and serving many NHP labs, including us, and helped solve many preprocessing issues of our marmoset data.
- 2) Although animal-friendly, AFNI didn't have a state-of-art tool for EPI distortion correction. Thus, we used the FSL-topup and incorporated into AFNI's main preprocessing stream. As the default topup config file involved many default parameters optimized for human data, we modified them to fit the marmoset brain size and data resolution. The modified config is also provided on the resource website ("code folder") for the revision.
- 3) TORTOISE is also another NIH-based software that is designed for diffusion MRI data preprocessing, which outperforms FSL-based tools in many aspects, including DWI denoising (not available in FSL), eddy current and EPI distortion correction (more flexible than FSL), EPI distortion correction, and tensor-based registration (not

available in FSL).

- 4) In our study, we used in-vivo dMRI data for computation modeling. As the modeling is ROI-based rather than voxel-based, the slight difference between TORTOISE (for dMRI) and FSL-topup (for fMRI) would not lead to an opposite conclusion.

As mentioned above, we made efforts to improve the preprocessing pipeline for marmosets, and the similar pipelines has been adopted by other marmoset imaging labs.

However, we admit that our manuscript does not bring a breakthrough for preprocessing method. In the revised manuscript, we discuss this limitation, which is also a common challenge, not only for marmoset neuroimaging but also for other animal species. As a matter of fact, our project (marmosetbrainmapping.org) also plans to develop such an analysis package in future versions. More importantly, we hope that by releasing the large raw fMRI data of marmosets, the whole community can come together to develop better data analysis packages and pipelines for non-human primate neuroimaging.

- 4) The novelty of this paper is resting-state fMRI datasets from two institutions, as well as in-vivo diffusion MRI in addition to the ex-vivo dMRI that were made publicly available by the same authors previously. Therefore, a novel idea or analysis is expected for charactering and reducing bias, data harmonization and/or statistical harmonization between different datasets. However, nothing was explicitly performed unfortunately. Note that harmonization is a relatively new research interest in the human MRI large population studies (e.g., HCP, HCP, ABCD, UK biobank, Brain/MINDS-beyond) in order to understand subject variability. The explicit analysis for this approach to harmonize data across institutions/scanner and brain preparations (in-vivo vs ex-vivo) may also be valuable for the animal studies, as well as for its future international global neuroscience in NHP (see. PRIMatE Data and Resource Exchange (PRIME-DRE) Global Collaboration Workshop and Consortium, Neuron 2022).

OUR RESPONSE: Thanks for the reviewer's valuable suggestions. As we answered the

first question, we comprehensively compared two datasets (tSNR, CNR, and head motions) and there was no significant difference between two datasets. See the response to the major comment point-2.

5) The population-based boundary maps are likely similar between the two institutions, but they are not very close to each other. There seems to be dislocation or misregistration of the boundaries over the surfaces. Similarity of the functional boundary maps between hemispheres is also not very high. These findings are not reasonable since less hemispheric asymmetry or smaller intersubject variability are expected in the marmosets than in humans. The reviewer thinks that biases and artifacts are not removed completely for these datasets, thus it is too early to answer these naïve questions on neurobiological or network organization.

OUR RESPONSE: Thank you for this comment. Like humans, marmosets are outbred and thus genetically diverse, and thus their inter-individual variability is quite large. Their dramatic intersubject variability can be observed and quantified not only in the current fMRI data, but also in structural images (brain size/shape and region size) published previously (*Liu, et al, Neuroimage, 2021*):

Cited from Liu, et al, Neuroimage, 2021

As for the issue of inter-hemispheric asymmetry, we checked the original publication describing the boundary map method (Gordon et al. Cerebral Cortex 2016), which analyzed 120 healthy young adult human subjects (60 females, mean age = 25 years, age range = 19–32 years) during relaxed eyes–open fixation. All subjects were English native speakers and right-handed. The resulting population-based boundary map between hemispheres was not similar (see below).

Cited from Gordon et al. Cerebral Cortex 2016

Since the original work did not show the comparison Dice value across the hemisphere, it is hard to estimate whether marmosets have less hemispheric asymmetry than humans. However, their results (*Gordon et al. Cerebral Cortex 2016*) might indirectly suggest that that might be the case. For example, Gordon et al. found 422 cortical parcels (206 in the left hemisphere and 216 in the right hemisphere). In contrast, we obtained the same number of parcels (96) for both marmoset hemispheres after blindly processing each hemisphere independently. Furthermore, as answered in the previous question, we have tried our best to ensure consistency of the data across the two marmoset populations (NIH and ION) and performed the same preprocessing of both datasets.

6) As for the link to the structural-functional connectivity, it is not understandable why use of neural tracing data did not achieve highest predictability of functional connectivity. What were these predictions if they were separately analyzed for short and long-distance connectivity? Is there anything that Hopf bifurcation neurodynamical function cannot model? High-resolution ex-vivo diffusion MRI data should be excellent as compared with any other diffusion MRI datasets, but it will not be complete nor have supreme SNR for the high b-value data. There are several issues on the accuracy of the diffusion tractography such as crossing vs kissing fibers, gyral bias, inaccuracy for long-distance tract, which are not yet solved completely in any model or algorithm. The neural tracing data using retrograde tracers may be absolutely reliable in terms of strength and specificity of connectivity and

by far robust against distance.

OUR RESPONSE: We appreciate the reviewer for making this suggestion. We have re-run simulations using our model considering the distance. As you predicted, the neural tracing data is robust against distance (see panel **D** below). However, the overall performance is influenced because the simulation based on tracing data did not perform well in the short-range distance.

We added the figure above to the supplementary information (Fig. S9) in our revised manuscript and also reminded the reader of the importance of this consideration (line 431-439):

“Since the accuracy of the diffusion tractography may be influenced by the lengths of tracts, we used the MBMv4 to further evaluate the structural reliability of different connective distances for each type of data. The fitting results of our model are consistent with the prediction that the distance affects the accuracy of diffusion tractography, as both in vivo

and ex vivo dMRI showed with lower structural-functional fitting correlations for long-range compared to short-range connections (Supplementary Fig. S9A-C). On the contrary, as expected, the neuronal tracing data are more reliable and robust to model connectivity against distance (Supplementary Fig. S9D). In summary, the MBMv4 presents a reasonable framework for examining discrepancies between structural and functional connectivity.”

Minor comments:

In abstract and many parts of the main text, the authors use a term ‘MRI space’. This is a bit sloppy statement since it implies the MRI scanner coordinates, which is not meaningful for understanding the brain. The reviewer recommends the authors to use a more general term, something like ‘standardized space of the marmoset brain’ or ‘marmoset brain template of ear-bar and eye-bar coordinates (MBMv3 or MBMv2)’. The abstract should be more specific about which resource data is shared in this publication. Ex-vivo diffusion MRI and neural tracing data were already made publicly available in the authors’ previous publication.

OUR RESPONSE: Thank the reviewer for pointing them out. In the revision, we removed these sloppy statements; and made it clear which resource data is previously published in the abstract.

*“Here we present a comprehensive resource that integrates the largest awake non-human primate resting-state fMRI available to date (39 marmoset monkeys, 710 runs, 12117 mins) with **previously published** cellular-level neuronal-tracing (52 marmoset monkeys, 143 injections), and multi-resolution diffusion MRI datasets. The combination of these data allowed us to ...”*

Results

Line 109. For the quality assessment of fMRI between the two sites, the tSNR alone is

insufficient at all. The tSNR calculated by simple mean divide standard deviation does show the quality of the images, but not for the quality of signal of interest (i.e., BOLD or neural signal). The variance of the BOLD activity in the resting-state fMRI is less than 10% among all even in the ultrahigh-field scanner. Therefore, for the quality of the fMRI data, analysis of variances obtained from data-driven analysis may provide useful measures including contrast-to-noise (CNR), motion-related variables, high-pass, and other artifacts (e.g., Marcus et al., 2013). For the quality assessment of diffusion MRI, the estimated motion, slice-by-slice outlier detections, CNR (for the diffusion weighted signal changes) are useful for presenting the data quality (e.g., Bastiani et al., 2009).

OUR RESPONSE: We thank the reviewer for this good suggestion. In the revised manuscript, we provide these quality assessments as suggested in the supplementary materials (see the response to major comment #2 above).

Line 139. The presented ICA components are very nice (Fig 2, S2). But the results are not showing any temporal features such as frequency spectrum. It is more informative to reveal the frequency spectrum for each component and show how they look like neural origin. It is questionable if the components E, F, P, S in S2 is really neural origin. The regressing ICA components to the unfiltered data can generate the time series data and frequency spectrum. It is also informative to demonstrate the representative noise or artifactual components in the supplements.

OUR RESPONSE: We provided the timeseries and frequency power plots of each component in supplementary Figure S5. All the components show neural-like patterns spatially (all peaks are located in the cortical or subcortical gray matter) and temporally (no patterns of artifacts or noises). See the previous response to major comment #2.

Line 150 'Default-model-like network (DMN)' Default mode network? In line 15 in the legend of Figure S2, 'default-network' should be default mode network.

OUR RESPONSE: Thank the reviewer for pointing it out. We have corrected it in the revision.

Line 240 'DCBC'. The distance and areas over the surface may be largely different between subjects, which are not well controlled or regularized unless individual surface reconstruction and surface registration were not performed.

OUR RESPONSE: Individual surface reconstruction and surface-based registration were not performed for the following reasons:

- 1) Because of gyrification, surface-based registration is a better solution than volume-based registration for the human brain. However, in rodents and marmosets, the smooth brain didn't provide sufficient contrast for surface-based registrations; The volume information is essential for the registration. Meanwhile, the inaccurate spatial smoothing issue caused by the gyrification for the volume-based method is not apparent for the smooth marmoset brain, and thus the advantage provided by the surface-based registration is limited.
- 2) The current surface reconstruction tool (for example, Freesurfer) was deeply optimized for the human brain. The surface reconstruction methods for animal data are far from robust, accurate, and automatic. They require extensive manual intervention to generate a useable but imperfect surface, including the most state-of-art tools and pipelines published in the recent special issue of Neuroimage (for NHP neuroimaging tools). This issue has been emphasized as a critical challenge in the PRIME-DE paper and workshop.

Because of the great technical difficulty and low benefits of surface-based methods, almost all rodent and marmoset neuroimaging studies adopted a volume-based approach for registration.

Indeed, the individual surface could improve the accuracy of the estimation of spatial distance. However, the DCBC comparison is across-parcellations (Fig. 4C) or within the

same individual (Fig. 5D) rather than across-subjects. Each parcellation received a consistent influence from the individual surface variability and thus the conclusion that the MBMv4 has better DCBC than other parcellations is valid.

Line 335. 'MBMv4 provides 335 a more accurate reflection of the functional parcellation of the cortex than current histology-based atlases' - This is a bit sloppy statement, and any single modality of the brain metrics (including functional connectivity) may not provide perfect resolution of the cortical parcellations. For example, the histology data is more accurate than functional parcellations in some of the boundaries (e.g. V1/V2).

OUR RESPONSE: Thank you for pointing this out. We removed this statement in the revised manuscript.

Data acquisition:

The functional MRI acquisition protocols are not well harmonized between two scanners (ION vs NIH). There are many different parameters that significantly affect the quality of the data including static magnetic field, gradient strength, which may cause difference in SNR, tissue T1 and T2* value, B0 inhomogeneity & distortion. The transmission coil is also different between two institutions, causing substantial difference in B1+ field, and hence bias the homogeneity of the signal, SNR across space etc. Difference in B0 field and in TE of fMRI (18ms vs 22.2ms) should cause the differences in CNR (contrast-to-noise ratio) of the fMRI data, which is not clearly and carefully investigated. There is no way for the readers to estimate how B0 inhomogeneity-induced distortion is look like between two institutions, unless the original (distorted) image or estimated shift images are not presented. Distortion correction method, Topup or TORTOISE, is optimized for human brain by default, thus the authors should carefully describe if it was indeed effective enough for correcting distortion of the fMRI in the small brain. There is no information available for detailed scanning parameters including dwell time, which is needed for B0-susceptibility induced distortion correction with a blip-up blip-down method. Since the resource data

does not include any detailed info on imaging parameter (including json files usually stored in the BIDS format), so the data cannot be properly analyzed or replicated by the other users. The previous public data for ex-vivo diffusion MRI (V1 and V2) does not also provide all the information on the scanning parameters including dwell time.

OUR RESPONSE: The parameter file used for EPI distortion correction are modified according to the marmoset brain size, spatial resolution and dwell time to achieve the best correction for the marmoset brain. We provided the topup config file in the revision so that users can adopted the config file to re-run and test the topup. Meanwhile, in the resource, we provided the raw data and the preprocessed data with topup correction, so that the user can directly evaluate the outcome of the EPI distortion correction. We also added a supplementary Figure S10 to show the data before v.s. after EPI distortion correction.

In the revised resources (under "sequences folder"), we provide the raw method file from the Bruker scanner and generated the json file (by bruker-api), so that other users can

replicate our results or propose better analysis methods than the ones used in the current paper.

The fMRI preprocessing is done by legacy method and likely not optimized for this species. Registration is an important treatment in the preprocess, and is highly dependent on the resolution of the data and object size, therefore it is obvious that the marm. The quality assessment of rfMRI is not carefully done. There are no descriptions on the animal's compliance during scanning and whether their head was moved or not during scanning. No explicit artifact or noise removal is done in the first-level analysis, which is commonly done in recent human resting-state fMRI studies. Bandpass filtering (0.01 – 0.1Hz) in the initial preprocessing (line 620) is a bit outdated approach, since it loses the significant neural signals at >0.1 Hz. The resampling of the fMRI data to the standard space (line 624) is not done by concatenating all the transformation at a single step and likely excluding the distortion correction by Topup. No exploratory data-driven analysis is applied, thus there is no way for the readers to see how much the rfMRI data is contaminated by any unexpected artifacts. The resting-state of subjects is hard to control the conditions of subjects particularly in animals. Therefore, data-driven approach like ICA is strongly recommended in the first-level analysis for estimating quality of the data, then needs to be applied for artifact removal. As long as the authors do not apply this approach, the group-wise ICA analysis should also be contaminated by a large number of unexpected artifacts from each scan. It is strange that the authors apply a different or wider range of frequency in the post process analysis (0.04-0.07Hz in Line 858, 0.04-0.25Hz in Line 879).

OUR RESPONSE: Thanks for the reviewer's question. We did a similar analysis on data without band-passing filtering, the results of brain parcellation and evaluation are similar as the current reported data with band-passing filtering (0.01-0.1Hz), and thus the signals of 0.1 – 0.25 Hz didn't provide significant information for the parcellation. In our resources, we provided raw data and preprocessed data with/without temporal filtering. Thus, the users can easily use or reprocess the data for their specific purposes.

The ANTs are a general, flexible, and robust software for spatial registration and has been widely used in spatial normalization and template construction in all kinds of species (for example: marmoset (Liu, et al. 2021), macaque (Seidlitz, et al. 2018), tree shrew (Wang S, 2013), rat (Johnson, et al, 2021), mouse, dog (Datta, et al, 2012)). and compatible well with our marmoset data. We concatenated the transformation of different stages of registration as much as possible. However, the afni and fsl transformation are not directly compatible with ANTs. Thus, we didn't include the head motion correction and topup deformation in the concatenation. This would cause extra smoothing as multiple resampling were conducted, but considering the better volume-based registration outcome provided by the ANTs, the gains outweighed the losses.

Ref:

- 1) Liu C, Yen CC, Szczupak D, Tian X, Glen D, Silva AC. Marmoset Brain Mapping V3: Population multi-modal standard volumetric and surface-based templates. *Neuroimage*. 2021;226:117620. doi:10.1016/j.neuroimage.2020.117620
- 2) Seidlitz J, Sponheim C, Glen D, et al. A population MRI brain template and analysis tools for the macaque. *Neuroimage*. 2018;170:121-131. doi:10.1016/j.neuroimage.2017.04.063
- 3) Wang S, Shan D, Dai J, et al. Anatomical MRI templates of tree shrew brain for volumetric analysis and voxel-based morphometry. *J Neurosci Methods*. 2013;220(1):9-17. doi:10.1016/j.jneumeth.2013.08.023
- 4) Johnson GA, Laoprasert R, Anderson RJ, et al. A multicontrast MR atlas of the Wistar rat brain. *Neuroimage*. 2021;242:118470. doi:10.1016/j.neuroimage.2021.118470
- 5) Datta R, Lee J, Duda J, et al. A digital atlas of the dog brain. *PLoS One*. 2012;7(12):e52140. doi:10.1371/journal.pone.0052140

We tried the FSL ICA-FIX on the data for artifact removal, but we failed to obtain good results with ICA-FIX, as the tool is mainly designed for human data. Thus, we chose the traditional regression-based methods for artifact removal. In the future, if there are more delicatated tools developed for marmosets, we will try them and update the resource if better results are achieved.

For the head movement, we provided the raw head motion files in the resource websites for each fMRI. Note that we used the same helmet-based method to constrain the animal, and the head-motion level has been evaluated and compared with headpost approach in our previous paper (Schaeffer, et al, 2021).

Cited from Schaeffer DJ, Liu C, Silva AC, Everling S. Magnetic Resonance Imaging of Marmoset Monkeys. ILAR J. 2021 Feb 26;ilaa029. doi: 10.1093/ilar/ilaa029. Epub ahead of print. PMID: 33631015.

“It is strange that the authors apply a different or wider range of frequency in the post process analysis (0.04-0.07Hz in Line 858, 0.04-0.25Hz in Line 879).”

OUR RESPONSE: This two-frequency band is used for the modeling part, and we did not do any frequency filtering of our rsfMRI data when we applied it to our modeling simulation so that it would be in the full band. In our revised manuscript, we further remind the readers of this point in the Methods section.

Therefore, to find the optimal parameter alpha in our Hopf modeling, we selected the power

spectrum in the relevant band (0.04 to 0.07Hz) normalized by the power spectrum of the full range without low frequency (considering a fMRI acquisition of TR=2 s, 0.25Hz is the maximum frequency and 0.04 Hz lower boundary is acting as a high pass filter avoiding the low frequencies).

Surface mapping of the volume data (e.g. fMRI) is likely not performed using the subject's surface, thus is potentially bias the results of a large group-based functional connectivity. Surface mapping of the subject's volume onto the average or the other subject's surface should not be very precise given the fact that there is a significant intersubject variability of brain size and shape of this primate species as compared with rodents. There is no detailed explanation of the volume-to-surface mapping (line 629). How were the surfaces created and how they are well aligned with the subject's fMRI and T2-weighted images?

OUR RESPONSE: Thank you for your suggestions. Individual surface reconstruction and surface-based registration were not performed for the reasons described in our response to a previous comment about DCBC.

This study uses the Marmoset Brain Mapping V3 (MBMv3) template for volume-based registration and the Connectome Workbench (`wb_command -volume-to-surface-mapping` function and ribbon constrained mapping algorithm) for the volume-to-surface mapping. The MBMv3 has a population-based T2-weight template image and brain surfaces. First, the subject fMRI was co-registered to the subject T2w image, and the subject T2w was registered to the T2w template of MBMv3. Then, the transformations were concatenated to normalize the subject fMRI data to the MBMv3 template space. Finally, the normalized subject fMRI data were mapped to the MBMv3 brain surface.

The method of the volume-to-surfacing had been described in lines 672-678 and discussed the limitation in the revision manuscript regarding the individual surface mapping (lines 543-548): *“Fourth, as robust surface reconstruction tools were not available for marmoset brains, we didn't perform analysis on individual surfaces. Pooling all data onto the*

population-based surface may lose the information of individual variability on brain morphology and reduce the accuracy of individual functional connectivity calculation and evaluation, for example, the DCBC index. Thus, automatic surface reconstruction is highly demanding for marmoset neuroimage studies.”

Diffusion MRI. There are no careful descriptions whether the package of TORTOISE is used as is or optimized for the marmoset data. The image registration may be dependent on a global optimization technique that relies on the resolutions during iterative aligning processes. The optimality of the distortion correction by the blip-up and blip-down method may need to be carefully described. Nothing was described about what kind of tractography method was applied (line 639).

OUR RESPONSE: Thank you for pointing this out. TORTOISE was used as is. The TORTOISE was developed by a lab at NIH that also performed many animal-based studies and thus it was developed to be used as an animal-friendly package. Although small bugs existed in the earlier version for the marmoset data, they have been fixed in the current version used here (v3.1). For the tractography, we used the default iFOD2 (Second-order Integration over Fiber Orientation Distributions) methods of Mrtrix3. The tracking method details were described in the revised manuscript (lines 687-698):

“The response function of each preprocessed diffusion MRI data was calculated by the “dhollander” method of the “dwi2response” command, and then the fibre orientation distributions (FOD) were estimated using spherical deconvolution by the multi-shell multi-tissue CSD method of the “dwi2fod” command. Finally, region-to-region tractography was performed using the iFOD2 method of the “tckgen” command. For each pair of cortical regions, diffusion tractography was conducted by using one region as the seed and the other as the target, and vice-versa. Thus, each pair of regions generated two sets of tracking probability maps, which were normalized by total streamlines selected, and the two probability maps were averaged into a single map to represent the final map of the connection of the two regions. Finally, all pairs of connections formed the whole cortical

structural connectome for computational modeling.”

Neural tracing data: the mapping method of the neuronal tracing data onto the MTR template is not very clear. How accurate was the registration between the neuronal tracing data and MTR template? There are no visual presentations nor quantitative assessment on the validity of the registration. In particular, it is informative to present whether cortical boundaries of the tissue sections were properly aligned with the MRI template. No information is available about which cost function and registration method (linear vs nonlinear, DOF) was applied.

OUR RESPONSE: Thanks for your suggestions. The registration methods between MRI and Nissl templates were described in our previous method paper (*Majka et al, 2021*). We adopted a similar approach in the current study, using SyN transformations, CC similarity metric (cost function), and three-stage iteration alignments (rigid alignment (dof 6), affine alignment (dof 12) and non-linear SyN transformations). In the revised manuscript, we added these details (lines 713-717) and a new supplementary Figure S11 to demonstrate the validity of the registration visually.

Figure S11. *The registration of the histological NM template to the MBMv3 MRI template. The underlay is the T2w template of the MBMv3, and the overlay is the outline of the histological NM template that is transformed on the MBMv3 template space. The outline is generated by the @AddEdge function of the AFNI (using default setting).*

Ref: Majka, Piotr et al. "Histology-Based Average Template of the Marmoset Cortex With Probabilistic Localization of Cytoarchitectural Areas." NeuroImage vol. 226 (2021): 117625.

Group-ICA. Please properly describe which of NIFTI volume or CIFTI data the authors used for their group-wise ICA analysis using GIFT. Also please clarify if each scan data was normalized by the variance and its mean signal was removed before feeding into group ICA. The classification of the components (neural or noise) is likely done only based on the spatial distribution. This is a bit tricky because some of components even located within the brain may be associated with pulsation of the brain tissue (i.e., periodic movement). Frequency spectrum and time series signal may be presented for each component.

OUR RESPONSE: We used the NIFTI volume data for the group-wise ICA analysis. We adopted the default normalization method of GIFTI ("remove means per timepoints"). We also tested other normalization methods implemented in the GIFTI, which generated similar results. All components can be detected regardless of the normalization methods. As a software aimed for group-ICA analysis, the GIFTI handled the normalization well. We provided each component's time series and frequency power plots in the supplementary Figure S5. All the components show neural-like patterns spatially (all peaks are located in the cortical or subcortical gray matter) and temporally (no patterns of artifacts or noises).

Line 850. 'while brain parcellations'. I'm wondering if the authors would really discuss whole brain parcellations including cortical and subcortical parcellations.

OUR RESPONSE: I am sorry for our misleading. The parcellation didn't involve subcortical regions. Therefore, we modified the manuscript "whole-brain parcellations" to "cortical parcellation".

Line 851. '232 brain regions per hemisphere from Paxinos Atlas'. Is this number correct? Does this number include both cortical and subcortical structures?

OUR RESPONSE: Thank the reviewer for pointing this out. I am sorry for our unclear description. We have corrected in the manuscript: "116 cortical regions per hemisphere from Paxinos Atlas." For the Paxinos atlas, there are 232 cortical regions (116 per hemisphere). This number does not include the subcortical structures.

Reviewer #2 (Remarks to the Author):

Review of "An integrated resource for functional and structural connectivity of the marmoset brain" by Tian and colleagues.

The authors provide a huge comprehensive data set acquired in marmosets encompassing awake resting state fMRI, in-vivo DTI, high resolution ex-vivo DTI and a large collection of neuronal tracing data. Moreover, the authors relied on this massive data set, in combination with deep learning and computational modelling, to propose a parcellation scheme of the cortex and to link anatomical properties with functional connectivity measures. This is an exquisite data set that will become an important tool for many nonhuman and human primate researchers which will also improve homology research. Despite the complexity of the data and some of the analyses, the authors were able to summarize them in a succinct and very accessible manner. The analyses are state-of-the-art. I highly support publication of this manuscript and only have a few questions and/or suggestions which may improve the manuscript:

OUR RESPONSE: Thank you very much for your encouragement. We have rewritten the Results and Discussion sections of the manuscript in addressing all your comments.

Fig. 2A-O: I am surprised by the 'locality' of the different functional networks. In humans and macaques, resting state based functional networks typically consist of multiple anatomically segregated subunits. In this case, mainly singular nodes are apparent (except for panel J?). How is this explained?

OUR RESPONSE: Thank you for your suggestions. Among the 15 cortical networks, there are 7 functional networks (G, I, J, K, L, N, O) involving long-distance connectivity (anatomically segregated subunits). The reason that the networks appeared to be "locality" (but not really) is due to the relatively small size of the prefrontal cortex compared with macaque and in particular, humans. Thus, many small components didn't catch one's eye.

However, for all marmoset studies published so far (including the current study), the identified high-level association networks in marmosets are not as diverse as in humans. For example, we believe the networks I and J can be decomposed into default-mode, dorsal, ventral, frontal-parietal, and central executive networks in humans. But we were not

able to dissociate these networks in marmosets. This concatenation of marmoset networks may be attributed to evolutionary differences. Alternatively, it may be influenced by the limited fMRI spatial resolution (0.5 mm isotropic) relative to the size of the marmoset brain, although this is presently the highest resolution available.

From an esthetic point of view: why is the background in panels 2I-O grey? I would use the same background in all panels.

OUR RESPONSE: We appreciate the reviewer for making this suggestion. We changed all panels into the same white background.

2. Fig 2P, Q: It is unclear how the two network-parcellation maps are created (this was also not clear from the methods section) and more importantly how they relate to each other.

OUR RESPONSE: Thank you for pointing this out. We apologize for our unclear description in Methods, which we have revised in the manuscript (lines 739-746). We described the creation of the network-parcellation methods as follows: First, we combined the 15 cortical networks according to their spatial locations. Second, we took the highest values according to their normalized Z scores from ICA if they have spatial overlapping. We obtained short-range network parcellation based on the above steps (Fig. 2P). However, short-range (local) connectivity usually is stronger than long-range connectivity. Therefore, when overlapping two ICA components, regions with long-range connectivity may be covered by the short-range network parcellation. Thus, we created a second map for the long-range parcellation to cover all network components. We repeated the above step but only applied to networks with long-range connections (Fig. 2I-K) to obtain a second network parcellation.

3. Line 200-207. I'm puzzled by the seemingly higher intra-subject than inter-subject variability. Given the fixed anatomy (within a subject), which may differ from that in other subjects, this is difficult to understand. Moreover, this also poses problems for 'within' subject research -e.g. where one aims to relate boundary maps with task-based

information, injections, electrophysiological recordings, etc. Please discuss.

OUR RESPONSE: Thanks for the reviewer's valuable suggestions. This is because the boundary map has a big variation within a subject with a different number of runs (see Fig. 2E). Thus, here mapping the population-based atlas to a single session within a subject would result in a relatively large variation because of the different number of runs between sessions. However, as we demonstrated in supplementary Fig. S8, it does not influence the functionally related information change in the parcel. Thanks for the reviewer's suggestions, in the future, we will make more effort to figure out how many runs will be enough for the data reliability.

4. 229: Please explain exactly what is meant by 'semi-manual optimization'. This was also unclear from reading the methods section. This may lead to reproducibility issues. Please discuss what exactly is done manually and whether this imposes methodological limitations.

OUR RESPONSE: We apologize for our unclear description, and we have updated the manuscript in lines 796-799. A detailed description of manual adjustments is as follows:

1) Since the thicknesses of boundaries are different, we might need to manually adjust the parcel borders after the automatic parcel generation.

2) The uneven region growing method might result in incorrect attributions within a parcel. So we have to examine and manually correct these errors.

3) According to the boundary map, we need to perform a spatial smoothing of parcel borders (here, we used the 8-neighbor spatial smoothing method).

Because the above procedures are all related to computer vision, we will need to improve our algorithms for processing 3D imaging data and introduce more efficient automatic processing methods in the future.

5. Fig 6A and B (scatter plots). It is unclear what exactly is plotted. Also, what is the color (MBMv4) and grey-scale (Paxinos) mean in panel B?

OUR RESPONSE: We apologize for our unclear description. In Fig. 6A (left panels), these gray curves on the flat brain map present the borders of brain regions or the generated parcels from different atlases (Paxinos atlas and our MBMv4 atlas). The foreground of red regions is the fMRI activation patterns when the animals watched 20s visual movie clips. To demonstrate the functional prediction of our atlas, we calculated the match between the border of functional activation and brain regions/the generated parcels. In other words, for every voxel from the border of the functional activation pattern, we can calculate its shortest Euclidean distance with the borders of brain regions from Paxinos atlas or the borders of generated parcels from MBMv4 atlas. The results are shown in the right panels (scatter plot). To quantify the effect, we fitted all rasters and made a regression. The dashed black line represents the diagonal line, and the red line represents the linear fitting line. The linear fitting line is toward the side of MBMv4 atlas, so it means that MBMv4 indeed has a high match compared with the Paxinos atlas.

For Fig. 6B (gradient show), we labeled the voxels in the same spatial gradient with a color scale (from light to dark) according to the scores of the 1st gradient. You can find three colors in MBMv4 (yellow, green, and purple). However, for the Paxinos atlas, we cannot reconstruct such gradient effects so that they are all in one color (gray).

6. In several locations (e.g. line 379) the authors mention that MBMv4 offers an alternative view on understanding the functional connectivity of a brain. This is a crucial statement. What exactly is meant? How can one relate both views? Is one 'better' (ground truth?) than the other? Is it reflecting the difference between anatomical versus functional connectivity? Or a difference in 'sampling efficiency'? Or? Please discuss in depth.

OUR RESPONSE: Thank you for these comments.

1) Since the brain function is generally implemented across spatial-temporal dimensions, measuring it from different scales is necessary. Therefore, our MBMv4 offers the brain description on a scale of functional-connectivity architecture.

2) Structure-function connectivity is related to each other. Our MBMv4 might offer an option for us to investigate this relationship. As the reviewer said, MBMv4 reflects the difference between anatomical features versus functional connectivity (the related discussion has been shown in lines 485-486).

3) The classic atlas, such as Paxinos Atlas, is based on cytoarchitectonics. Since that, it cannot reflect individual differences. However, our MBMv4 using a non-invasive MRI imaging technique catches the individual “functional differences” to some extent. Therefore, it offers an option for us to investigate individual brain functions.

In sum, in our opinion MBMv4, to some extent, reflects some combination of direct and indirect structural connectivity according to our modeling simulation and also the statistical history of functional coactivations according to the task verification.

7. Related to the previous point: Different parcellations can be obtained with different measures (e.g. rs functional connectivity, anatomical connectivity, cyto-and myelo-architectonics, etc..). It would be highly informative if a few obvious parcels where all measures match and where measures don't match are described/shown in more detail (not using the whole-brain overviews).

OUR RESPONSE: We have discussed the parcels that do or do not match the structure (see our discussion line 486-497). For example, in our MBMv4, the somatomotor cortex is parcellated into parcels across multiple areas of the facial, forelimb, and trunk musculatures. If we look at V1/V2 region, some parcels follow the representation of eccentricity in the visual field, especially for the foveal, and some are not. We consider this is because our MBMv4 is only based on resting-state fMRI. Some topographically organized cytoarchitectonic areas could be dissociated from the resting-state functional responses. In the future, we will make more effort to detect these differences.

8. The study revealed more functional networks and more parcels than previous marmoset studies. Does one expect these numbers to grow when higher spatio-temporal resolution

data can be acquired in the future? In other words, to what extent are these parcels dependent on the type of measurements? This is important from a theoretical point of view as one (implicitly) creates the impression that the current processing modules are the ones determining cortical functioning. But what if one misses the really important modules (e.g. columnar like) simply because of current technical limitations?

OUR RESPONSE: We believe with higher spatiotemporal resolution and more fMRI modality (task-based fMRI like human studies), we can get a greater number of parcels (more detailed parcellation). Although we were able to acquire most state-of-art awake resting-state fMRI dataset of marmosets, the current 0.5mm isotropic spatial resolution was still not enough to map the functional network and parcellation in the small marmoset brain, especially for the complex but small prefrontal cortex. We discussed the limitation in the revised manuscript (lines 541-543): “Third, although the resource provided the most state-of-art awake resting-state fMRI, the *0.5mm isotropic resolution may not fully capture the functional-connectivity patterns of the small marmoset brain, because of current MRI technical limitations.*”

9. Line 506: I guess inter-subject is meant instead of intra-subject?

OUR RESPONSE: Thanks for pointing out this typo. It was corrected in the revision.

10. Line 533: sentence is not clear.

OUR RESPONSE: The sentence was revised to “*Because neuronal tracing data revealed true directional anatomical connections, which is the unidirectional diffusion tractography may not capture. the intactness of the neuronal tracing data is helpful for an accurate mapping of the structural connectome.*”

11. Line 724: why is the 60th percentile used?

OUR RESPONSE: We followed the rule in the original paper of boundary map and found it has a good performance with our marmoset dataset (Gordon, cerebral cortex 2016).

12. Line 793: Please explain the visual-choice task. I guess this was not a free-viewing movie task?

OUR RESPONSE: The data was recycled from our recently published paper in *Cerebral Cortex* 2021 (The Brain Circuits and Dynamics of Curiosity-Driven Behavior in Naturally Curious Marmosets). We designed the delayed free-choice task (Fig. 1A in the manuscript), and Marmosets had to choose between two target images after watching a short movie clip (20sec). Here, we only took the fMRI activation pattern during 20s movie watching, during which the animal cannot make eye movements out of the screen (eye field is 10 deg x 8 deg).

REVIEWER COMMENTS

Reviewer #1 (Remarks to the Author):

The reviewer appreciates the authors' efforts in revising the manuscript. The manuscript now added the quality assessment of the data (S2, S3, S6), which are informative for those who want to use data. Unfortunately, there are no attempts in this revision to applying denoise and harmonization of the fMRI datasets across sites/scanners, which significantly loses the impact of this manuscript. Figures added in this revision indeed suggest the noise or bias are significantly left unremoved in these datasets. Despite of their importance, preprocessing including registration and standardization is not pipelined, thus users who download MBMv4 parcellation cannot replicate embedding of their own imaging data to the standard coordinates where MBMv4 parcellation was carried out. The parcellations are done on purely data-driven approach without any validation by comparing with histology and tracer data. Overall, data looks great, but preprocessing is not complete, thus it is not clear what is the scientific novelty. Therefore, the reviewer recommends the authors to share the current codes of preprocessing and explicitly describe the limitation of the MRI preprocessing, when the data is made publicly available.

As for structural imaging of T1w and T2w, the authors presented simulation data in the rebuttal that the T1w scanning does not work well in 9.4T as compared with 3T due to elongation of T1 at ultra-high field, which is well known phenomena. However, the point is whether T2w image at ultra-high field really has enough high contrasts of brain structures (e.g. gray matter/CSF contrast, gray/white matter contrast) for achieving good cross-subjects alignments of the brain areas. The dMRI may be much better for gray/white matter contrast, and probably useful for cross-subject registration. It is good to demonstrate the histogram of the brain signals of T1w, T2w and dMRI volumes separately in the same subject. In addition, T2w signal may be affected mainly by (tissue) water fraction, T2* signal may be affected by the iron deposition and myelination, thus T2w may not be a good indicator of brain structure.

The results added in this revision (Fig. S2 and S6) demonstrate incompleteness of denoising and ambiguity of site/scanner differences. In Fig S2, it is not well understandable why the signal-to-noise ratio (SNR) is same for the same resolution (0.5mm) of EPI sequence between two sites (NIH vs ION) despite of differences in strength of static magnetic field (7T vs 9.4T, respectively). Interestingly, CNR of functional MRI used contrast of gray/white signals, which is not a good indicator of the quality of the functional MRI. Grey/white contrast is widely used for structural MRI, because the structural images are commonly used for registration, segmentation, and surface estimation based on the contrast of gray/white matter. In case of fMRI, it is common to have interests in the blood-oxygen dependent 'neural' signals, e.g., beta of 'neural' activity in the task fMRI statistical model or the variance of the resting state fMRI component are optimal contrasts of interests for estimating CNR of fMRI. In addition, there are not any assessments of 'noise'-like features in the fMRI datasets, which can be estimated by

applying the first-level ICA and estimating variance of noise-like components. Indeed, Fig S1 shows that there are significant slice-by-slice artifacts in the temporal SNR maps in both sites, which can be estimated by applying the first-level ICA and even removed by regressing out the spatio-temporal component. In Fig S6, the authors revealed the signal-time series data, frequency spectrum data for signal components of group-wise ICA. However, the frequency spectrum does not very clearly show the 'neural like' features and likely have contamination of very low frequency noise. In particular, D, E, F, R are questionable if they are really neural, while B, I, J, K seem most promising neural components. With so much thresholding, it is hard to know what is going on in most of areas. It is also strange that there is significant frequency band > 0.1 Hz although the authors described that band pass filtering (0.01 – 0.1 Hz) (but w/o nuisance regression) was applied before feeding into ICA. When applying ICA, removal of the brain signal mean is not very good idea because of difference in the biasfield across runs, and potential bias by global signal removal. The mean brain signal needs to be normalized to a fixed value (e.g. 100 in SPM or 10000 in FSL, HCP), followed by removal of the timeseries mean, detrending of signals and cross-run variance normalization are recommended.

Although the reviewer pointed out the potential difference of functional connectivity between sites/scanners (Fig. 3, S6), it is unfortunate that the authors did not assess the effect of the sites/scanners on the FC connectome or similarity of FC connectome between sites/scanners in this revision and no attempts were made for denoising and harmonization of the data across sites/scanners. This is probably the most significant drawback of this paper, since the valid analysis of the resting-state fMRI data relies on the ability to remove any potential noises and biases, since a major part of the fMRI signal variance ($> 90\%$) is dominated by structured and random noise.

Dependence of the relationship between simulated and empirical FC on distance (Fig S9) is interesting. It demonstrates that major part of the variance of simulated functional connectivity (based on dMRI) is explained by the distance while this is not true when FC was simulated using neural tracer. This indicates correlation between simulated and empirical FC (Fig 7E) is largely dependent on distance via the error propagation from dMRI tractography. However, since a part of the authors demonstrated in their recent study that neural tracer data of marmosets reveals exponential distance rule (EDR) (Theodoni et al., 2022) like in other species (macaque and rodent), the issue of distance-dependency of dMRI tractography may be carefully managed before simulating FC.

As for asymmetry of the marmoset brain, reviewer described in the prior revision that it may be to a much smaller extent than in humans. However, the data in the revised manuscript does not address this issue. While number of parcellations was the same between hemispheres, there is likely left vs right differences in the cortical boundary maps in Fig 3. There is possibility that the areal size of the parcels may be different between hemispheres, but it was not evaluated in this revision. This can be achieved by invert warping the standard average surfaces to the subject's native space and calculating the surface areas of each parcel in subject's native space. The authors may want to make it clear if asymmetry is really smaller than in humans.

Reviewer #2 (Remarks to the Author):

The authors carefully addressed all concerns of myself and the other reviewer. I don't have further questions

We thank all the reviewers and the editor for their strong encouragement and support of our manuscript. In what follows, we provide point-by-point responses to reviewer#1's concerns. The review comments are in black font, and our responses are in blue. The resulting edits of the original manuscript are highlighted in red font in the revised manuscript.

Besides resolving the reviewer's comments, we also improved our resource by adding an online connectome viewer to allow the user to explore different connectomes reported in our paper: connectome.marmosetbrainmapping.org

REVIEWER COMMENTS

Reviewer #1 (Remarks to the Author):

The reviewer appreciates the authors' efforts in revising the manuscript. The manuscript now added the quality assessment of the data (S2, S3, S6), which are informative for those who want to use data. Unfortunately, there are no attempts in this revision to applying denoise and harmonization of the fMRI datasets across sites/scanners, which significantly loses the impact of this manuscript. Figures added in this revision indeed suggest the noise or bias are significantly left unremoved in these datasets.

We should express many thanks to the reviewer. With valuable comments, we made more assessments and quality control in the previous and current revisions to guarantee the usability of our data. In fact, we realized the "harmonization" or "reproducibility" when we started the data acquisition in 2018, right after the PRIME-DE data was released (Milham et al., 2018). Therefore, we made a lot of efforts to make the datasets comparable and compatible, including similar awake training and data acquisition protocols. As a result, if compared with the early PRIME-DE, there has been a great improvement in our research work. Furthermore, in the latest manuscript version, we also made a considerable effort to

provide new results as much as possible to demonstrate the reproducibility and denoising between two datasets, including the results of a previous revision (different types of QA: SNR, tSNR, head motion, QA reports from AFNI and TORTOISE), the quantification of ICA components (CNR of components), the ICA components under different preprocessing (different normalization and noise estimation), and the comparison of the functional connectome across sites/scanners, etc. Therefore, the "harmonization" or "reproducibility" might not be the weakness, but the strength of our resources compared with previous NHP fMRI resources, especially after the improvement by following the reviewer's valuable suggestions.

However, we must admit that data harmonization and denoise is a quickly developing field, especially with the open resource of the primate MRI from multiple sites. In particular, a recent review paper published by *Autio et al., 2021*, comprehensively describes the minimal specifications for the non-human primate MRI, providing many useful guides for the non-human primate MRI data collection and analysis. Unfortunately, compared to the latest guides, our research work cannot fully follow the same practices because of practical difficulties (we completed the data collection before 2021 and the technical support), such as the lack of T1w images and parts of QA assessment in our original submission.

In our latest manuscript version, we purposely opened a subtitle in Discussion (Specific limitations) to discuss these limitations and emphasized the importance of this effort we should make in the future. Lines 557-606.

Despite of their importance, preprocessing including registration and standardization is not pipelined, thus users who download MBMv4 parcellation cannot replicate embedding of their own imaging data to the standard coordinates where MBMv4 parcellation was carried out. The parcellations are done on purely data-driven approach without any validation by comparing with histology and tracer data. Overall, data looks great, but preprocessing is not complete, thus it is not clear what is the scientific novelty. Therefore, the reviewer recommends the authors to share the current codes of preprocessing and explicitly

describe the limitation of the MRI preprocessing when the data is made publicly available.

We are sorry for our unclear in the last version. Indeed, we have provided the instructions for the data preprocessing pipeline on the resource website (ReadMe sections). In addition, in this latest revised manuscript, we uploaded the preprocessing code (“MBMv4_preprocessing_pipeline.zip”) into the “Codes” folder so that the users could replicate our protocol (https://marmosetbrainmapping.org/data_for_MBMv4_reviews.html).

Codes	Codes only (without data and 3rd party tools)	Codes with exemplar data and all tools
-------	---	--

As for the preprocessing pipeline, we have closely collaborated with AFNI for a long time. The AFNI provided an easy-to-use automatical python wrapper `afni_proc.py` to generate different fMRI pipelines for different kinds of data. AFNI programs are generic image processing and mathematical analysis tools. Since few places need to define the species, the whole processing is friendly for the animal dataset. The wrapper automatically generated our pipeline (except that we added FSL-topup into the pipeline). All of them can be found in our example codes or instructions.

However, we admitted the shortages of our current preprocessing pipeline regarding our denoising approach compared with the HCP pipeline and widely discussed them in our latest manuscript version (Line 586-593).

"A third refers to data preprocessing and denoising. The human connectome project (HCP) has released a sophisticated and standardized pipeline for denoising by ICA-FIX (*Glasser MF, et al. 2013*). Recently, the HCP-style pipeline was successfully transferred to the application of the macaque brain (*Autio JA et al. 2020, 2021*) and significantly accelerated the comparative studies between NHP and humans. As a marmoset version of ICA-FIX was unavailable, we adopted the traditional preprocessing method to denoise resting-state fMRI data. Therefore, developing HCP-style pipeline of the marmosets will be important to fully reveal the functional connectivity patterns of marmoset brains"

Reference:

Glasser MF, Sotiropoulos SN, Wilson JA, et al. The minimal preprocessing pipelines for the Human Connectome Project. *Neuroimage*. 2013;80:105-124.

Autio JA, Glasser MF, Ose T, et al. Towards HCP-Style macaque connectomes: 24-Channel 3T multi-array coil, MRI sequences and preprocessing. *Neuroimage*. 2020;215:116800.

Autio JA, Zhu Q, Li X, et al. Minimal specifications for nonhuman primate MRI: Challenges in standardizing and harmonizing data collection. *Neuroimage*. 2021;236:118082.

As for structural imaging of T1w and T2w, the authors presented simulation data in the rebuttal that the T1w scanning does not work well in 9.4T as compared with 3T due to elongation of T1 at ultra-high field, which is well known phenomena. However, the point is whether T2w image at ultra-high field really has enough high contrasts of brain structures (e.g. gray matter/CSF contrast, gray/white matter contrast) for achieving good cross-subjects alignments of the brain areas. The dMRI may be much better for gray/white matter contrast, and probably useful for cross-subject registration. It is good to demonstrate the histogram of the brain signals of T1w, T2w and dMRI volumes separately in the same subject. In addition, T2w signal may be affected mainly by (tissue) water fraction, T2* signal may be affected by the iron deposition and myelination, thus T2w may not be a good indicator of brain structure.

We appreciated the reviewer's concern regarding the registration and the value of T1w images, considering the large anatomical variability of the marmoset brain. We also agreed with the reviewer that T1w is a better indicator of brain structures than T2w; Recently, Ose et al. also published a valuable and significant paper in this field to indicate that T1w will help classify brain structures and registration as well as the evaluation of the marmoset brain parcels.

Unfortunately, not all of our marmosets in our fMRI resources have the T1w scanned for practical reasons since this project started in 2018 and ended during COVID pandemics. However, as the reviewer concerning registration accuracy, based on what we have, we purposely compared the performance of the registration of the T1w image and T2w image. Based on our multi-modal surface template of MBMv3 (Liu et al., 2021 Neuroimage), the registration results of the T1w image and T2w image are almost identical by using ANTs, regardless of whether the skull was removed (see the below figure for an example subject).

We also uploaded all exemplar data and registration test results under the link "T1w and T2w Registration Rest"

(https://marmosetbrainmapping.org/data_for_MBMv4_reviews.html). So It is easier to

load these images into one image viewer and check the registration results.

The underlay is the subject T2w image transformed in the MBMv3 space, and the overlay is the outline of the T2w template of the MBMv3. The outline is generated by the @AddEdge function of the AFNI (using the default setting).

In addition, because of this shortage of our resources, we reported the research progress in solving the anatomical variability of marmoset brains by T1w image and reminded the necessity of T1w image in our revised manuscript:

In the discussion, Line 569-576

"In addition, the resource did not contain the T1w images, as the T2w images were more commonly used for small animal imaging using ultra-high field MRI. However, with better tissue contrasts than the T2w images, the T1w images are more commonly used in human neuroimaging studies and can improve brain tissue segmentation, spatial registration, and myelin maps' estimation (Ose T et al. 2022). Therefore, the T1w images should be considered in the future for a more accurate preprocessing and comparative studies across

species.”

Reference:

Ose T, Autio JA, Ohno M, et al. Anatomical variability, multi-modal coordinate systems, and precision targeting in the marmoset brain. *Neuroimage*. 2022;250:118965.

The results added in this revision (Fig. S2 and S6) demonstrate incompleteness of denoising and ambiguity of site/scanner differences. In Fig S2, it is not well understandable why the signal-to-noise ratio (SNR) is same for the same resolution (0.5mm) of EPI sequence between two sites (NIH vs ION) despite of differences in strength of static magnetic field (7T vs 9.4T, respectively).

Although the 9.4T has a stronger magnetic field than the 7T, our 7T scanner comes with stronger and smaller gradients (450 mT/m, 15 cm) than the 9.4T scanner (300 mT/m, 20 cm). Therefore, we pushed the TR/TE to a limit for both scanners to obtain the best fMRI data. Under the current TR/TE, however, the gradient of the 9.4T scanner is not strong enough to implement the Bruker sampling technique (double sampling) because of the duty cycle. Thus, the 7T gradient compensated for the weaker magnetic field and improved the SNR. In addition, we provided the raw Bruker method file from both scanners in the resources, and the MRI physicists can easily import the method file into their Bruker scanner for usage.

https://marmosetbrainmapping.org/data_for_MBMv4_reviews.html

Sequences	Sequences_(Bruker_Method_Files)
-----------	---------------------------------

Meanwhile, the RF coil of 7T is slightly better than the 9.4T coil due to a small difference in the coil case. Both 7T and 9.4T RF coils were made by the same design and person (Daniel Papoti from Silva lab; Papoti, et al, 2017). For the ION 9.4T coil, we added an extra 3D-printed plastic case to protect the coil loop. In contrast, we removed the case for the NIH 7T coil, and the coil loops were directly wrapped with soft plastic skin; thus, it was

closer to the animal head and provided better SNR, although the coils were almost identical.

Because of these differences in MRI gradient and coil protection cases, the weaker 7T achieved similar fMRI imaging quality to the 9.4T.

Reference:

Papoti D, Yen CC, Hung CC, Ciuchta J, Leopold DA, Silva AC. Design and implementation of embedded 8-channel receive-only arrays for whole-brain MRI and fMRI of conscious awake marmosets. *Magn Reson Med* 78, 387-398 (2017).

Interestingly, CNR of functional MRI used contrast of gray/white signals, which is not a good indicator of the quality of the functional MRI. Grey/white contrast is widely used for structural MRI, because the structural images are commonly used for registration, segmentation, and surface estimation based on the contrast of gray/white matter. In case of fMRI, it is common to have interests in the blood-oxygen dependent 'neural' signals, e.g., beta of 'neural' activity in the task fMRI statistical model or the variance of the resting state fMRI component are optimal contrasts of interests for estimating CNR of fMRI.

Thanks for the reviewer's corrections. Here we calculated the variance of the resting-state fMRI components to estimate the CNR of fMRI. They do not have apparent differences across sites/scanners. Below is the CNR of the four exemplar components that the reviewer labeled neural components (also as shown in the supplementary Fig. S7).

We selected four ICA components from our network results (B: the dorsal somatomotor I: the frontoparietal, J: the default-model, and K: the visual-related network) and compared their temporal CNR between the two datasets. No significant difference was found between the two sites (The Wilcoxon rank tests, $p > 0.05$, $N_{\text{ION}}=346$, $N_{\text{NIH}}=364$).

In addition, there are not any assessments of 'noise'-like features in the fMRI datasets, which can be estimated by applying the first-level ICA and estimating variance of noise-like components. Indeed, Fig S1 shows that there are significant slice-by-slice artifacts in the temporal SNR maps in both sites, which can be estimated by applying the first-level ICA and even removed by regressing out the spatio-temporal component.

Thanks for the reviewer's suggestions. For the first-level ICA, we provided an example of first-level ICA on a single run, under the link "Single Run FSL-ICA" of (https://marmosetbrainmapping.org/data_for_MBMv4_reviews.html).

For the 3rd Round Review

Single Run FSL-ICA

T1w and T2w Registration Rest

The ICA-FIX has been used to remove noise-like components in an HCP-Style sequence for macaques brain imaging (Autio et al., 2020). This is significant progress for non-human primate neuroimaging. Indeed, when we started this project, we made a lot of effort in this direction. But unfortunately, as we mentioned in the last manuscript version, we cannot get the satisfying results by FSL ICA for denoising our marmoset brain dataset.

The reviewer may directly view the HTML report generated by the FSL of the exemplar ICA analysis on a single run:

The reasons might be:

- 1) the FSL ICA-FIX was highly optimized for the human brain, which needs the specific optimization of marmoset datasets;
- 2) Most importantly, ICA-FIX also required training datasets where marmoset versions were still unavailable.

Therefore, we chose the traditional regression method based on AFNI to denoise the data, which had been used widely in many studies of different species and was recommended by the AFNI group. Meanwhile, we could also get technical support from the AFNI team to solve issues when working with the marmoset dataset. However, since optimizing ICA-FIX or HCP-style preprocessing for the marmoset dataset requires more technical support from

FSL or HCP rather than AFNI-team and us, it will be an actual research effort for our future. Therefore, we will widely connect with other labs working on an HCP-style preprocessing pipeline and keep updating the resource.

In addition, we admitted the shortages of our current preprocessing pipeline regarding our denoising approach compared with the HCP pipeline and widely discussed them in our latest manuscript version (Line 586-593).

"A third refers to data preprocessing and denoising. The human connectome project (HCP) has released a sophisticated and standardized pipeline for denoising by ICA-FIX (*Glasser MF, et al. 2013*). Recently, the HCP-style pipeline was successfully transferred to the application of the macaque brain (*Autio JA et al. 2020, 2021*) and significantly accelerated the comparative studies between NHP and humans. As a marmoset version of ICA-FIX was unavailable, we adopted the traditional preprocessing method to denoise resting-state fMRI data. Therefore, developing HCP-style pipeline of the marmosets will be important to fully reveal the functional connectivity patterns of marmoset brains "

References:

Glasser MF, Sotiropoulos SN, Wilson JA, et al. The minimal preprocessing pipelines for the Human Connectome Project. *Neuroimage*. 2013;80:105-124.

Autio JA, Glasser MF, Ose T, et al. Towards HCP-Style macaque connectomes: 24-Channel 3T multi-array coil, MRI sequences and preprocessing. *Neuroimage*. 2020;215:116800.

Autio JA, Zhu Q, Li X, et al. Minimal specifications for nonhuman primate MRI: Challenges in standardizing and harmonizing data collection. *Neuroimage*. 2021;236:118082.

In Fig S6, the authors revealed the signal-time series data, frequency spectrum data for signal components of group-wise ICA. However, the frequency spectrum does not very

clearly show the 'neural like' features and likely have contamination of very low frequency noise. In particular, D, E, F, R are questionable if they are really neural, while B, I, J, K seem most promising neural components. With so much thresholding, it is hard to know what is going on in most of areas.

Thanks for the reviewer's consideration. We had provided the un-thresholded ICA images for every component (named "atlas_MBMv4_networks_raw.nii.gz") in our MBMv4 resources: https://marmosetbrainmapping.org/data_for_MBMv4_reviews.html

Atlases Produced	Atlas V4 only	Atlas V4 with V3 templates
 atlas_MBMv4_networks_parcellation_pri	2022-05-13	95.7 KB
 atlas_MBMv4_networks_parcellation_ser	2022-05-13	35.9 KB
 atlas_MBMv4_networks_raw.nii.gz	2021-11-08	4.09 MB
 atlas_MBMv4_networks_thresholded.nii	2021-11-08	534 KB
 surffs.lh.MBMv4_cortex_parcellation.lat	2022-05-13	25.2 KB

The example results of "questionable" components D, E, F, R are shown below:

We believed these components are neural-like signals, as these components were located in cortical gray matter regions and showed bilateral patterns (spatial patterns).

To further verify the possibility of contamination of very low-frequency noise, we also purposely re-performed the ICA on data with a high-pass filter (>0.01Hz).

Below are two examples of neural components, D (the “questionable” component) and B (the “promising” component), from the data with a high-pass filter (>0.01Hz).

Below are the same examples of neural components D and B from the data without a high-pass filter (from raw figures S6).

Since the power peaks of these components are all above 0.01Hz, it is hard to say the contamination of very low-frequency noises. Therefore, we can still detect these four components after a high-pass filter (>0.01Hz). Furthermore, we have already tried to take the strict rule combining spectrum and spatial patterns to determine whether a component is neural or noise and our identified components should be reliable.

It is also strange that there is significant frequency band > 0.1 Hz although the authors described that band pass filtering (0.01 – 0.1 Hz) (but w/o nuisance regression) was applied before feeding into ICA. When applying ICA, removal of the brain signal mean is not very good idea because of difference in the biasfield across runs, and potential bias by global signal removal. The mean brain signal needs to be normalized to a fixed value (e.g. 100 in SPM or 10000 in FSL, HCP), followed by removal of the timeseries mean, detrending of

signals and cross-run variance normalization are recommended.

Thank you for being so concerned. We did not apply any band-passing filtering on ICA. Instead, we fed into group-ICA for the data with minimal preprocessing (slice-time correction, motion correction, topup EPI distortion correction, and spatial registration). This part of the method had been described in the original manuscript, Lines 777-778 "*First, preprocessed data without regression of nuisance covariates were group-ICA analyzed with increasing numbers of ICA components from 20 to 80 in steps of 10...*". The nuisance-covariate regression included the band-pass filtering, which was treated as a regressor with other covariates. Sorry that the description was not clear and confused the reviewer. We modified it to a clearer statement: "*First, preprocessed data without bandpassing and regression of nuisance covariates were group-ICA analyzed with increasing numbers of ICA components from 20 to 80 in steps of 10...*"

For the data normalizing, we also tested what the reviewer suggested, as well as other options of GIFTI, including "Remove Mean Per Timepoint - At each time point, image mean is removed.", "Remove Mean Per Voxel - Time-series mean is removed at each voxel", "Intensity Normalization - At each voxel, time-series is scaled to have a mean of 100." or "Variance Normalization - At each voxel, time-series is linearly detrended and converted to Z-scores (with mean brain signal scaled to 100)".`

Despite preprocessing choices, we could obtain similar components described in the paper. We demonstrated four components (one local network B, one global network J, and two questionable networks D and E suggested by the reviewers) as examples here. The "Variance Normalization" has similar results as the "Remove Mean Per Voxel" and "Remove Mean Per Timepoint".

Although the reviewer pointed out the potential difference of functional connectivity between sites/scanners (Fig. 3, S6), it is unfortunate that the authors did not assess the effect of the sites/scanners on the FC connectome or similarity of FC connectome between sites/scanners in this revision and no attempts were made for denoising and harmonization of the data across sites/scanners. This is probably the most significant drawback of this paper, since the valid analysis of the resting-state fMRI data relies on the ability to remove any potential noises and biases, since a major part of the fMRI signal variance (> 90 %) is dominated by structured and random noise.

We appreciate the reviewer's suggestions. To demonstrate the similarity of FC connectome between sites/scanners, we calculated the FC similarity (2-D correlation coefficient) across the subjects and the sessions from different sites/scanners (see below figure, also in manuscript supplementary figure S4).

(A) The left heatmap represents the functional connectivity similarity matrix across subjects (we used the 2D Pearson correlation coefficient as the similarity metric). Subjects No. 1-13 come from the ION dataset, and the rest subjects (No. 14-39) are from the NIH dataset. To demonstrate the high similarity across sites, we further made a histogram plot of the similarity metric for the subjects from the ION dataset (light blue), the NIH dataset (orange), cross ION-NIH dataset (Green). They present high similarity (ANOVA multiple comparison tests, $df=2$, $F=0.92$, $p = 0.4008$). (B) Same as the (A), we present the functional connectivity similarity matrix across sessions, Sessions No. 1-55 come from the ION dataset, and the rest sessions (No. 56-107) are from the NIH dataset. They present high similarity (ANOVA multiple comparison tests, $df=2$, $F=97.63$, $p = 1.717$)

Dependence of the relationship between simulated and empirical FC on distance (Fig S9) is interesting. It demonstrates that major part of the variance of simulated functional connectivity (based on dMRI) is explained by the distance while this is not true when FC was simulated using neural tracer. This indicates correlation between simulated and empirical FC (Fig 7E) is largely dependent on distance via the error propagation from dMRI

tractography. However, since a part of the authors demonstrated in their recent study that neural tracer data of marmosets reveals exponential distance rule (EDR) (Theodoni et al., 2022) like in other species (macaque and rodent), the issue of distance-dependency of dMRI tractography may be carefully managed before simulating FC.

We appreciate the reviewer's suggestions. Before simulating FC, we now re-calculated the structural connectivity based on the exponential distance rule (EDR) (Theodoni et al., 2022). The procedure was presented in the method part (lines 1024-1039)

“The resulting simulation of functional connectivity demonstrated the influence of the distance, especially for the diffusion MRI (see Fig. 7D). To decrease this impact maximumly, we added the Exponential Distance Rule (EDR) for a more accurate estimation of structural connectivity before the simulation. We first normalized the structural connectivity estimated by diffusion tractography to 0-1, then calculated the probability of structural connectivity according to the EDR rule ($p(d)=ce^{-\lambda d}$, $\lambda\approx 0.3$, $c\approx 0.94$ where d is the distance⁴⁰) and also normalized to 0-1. Finally, we transformed the normalized structural connectivity to match the probability of structural connectivity. For simplicity, we defined a threshold as the Median Absolute Deviation for the normalized structural connectivity. If any value was larger than this threshold, it would be replaced by the corresponding probability of structural connectivity to reduce the impact of distance.”

The simulated results are shown in revised Fig. 7 (E,F), and described in results lines 440-452:

Figure 7. A computational framework links the structural-functional connectivity according to different parcellation. (A) The application of the whole-brain modeling, including the estimation of structural connectivity from the neuronal tracing, diffusion MRI (in-vivo or ex-vivo) according to the Paxinos atlas or MBMv4, the simulation of functional connectivity from structural connectivity by the Hopf bifurcation neurodynamical functions, and the similarity measure with empirical connectivity from resting-state fMRI. **(B)** Comparing the fitting effect based on Paxinos atlas and MBMv4 in different spatial scales. The round dot represents an example from individual in-vivo diffusion MRI, the polygon is from ex-vivo diffusion MRI, the star is from neuronal tracing, and the solid red line represents the diagonal line. **(C)** Examples of correlation between the simulated and

empirical functional connectivity from (B); solid black lines represent marginal regression lines. (D) Comparing the fitting effect with (red) and without (blue) the correction by exponential distance rule (EDR). From left to right present the simulation results of the in-vivo diffusion MRI from an example subject (solid circle in B), in-vivo diffusion MRI from all subjects (all circles in B), the ex-vivo diffusion MRI (the polygon in B), and the neuronal-tracing dataset (the star in B). All dashed lines represent a 95% confidence interval. (E) Comparing the fitting effect with EDR correction based on Paxinos atlas and MBMv4 in different spatial scales. (F) Examples of correlation between the simulated and empirical functional connectivity from (E).

As the connection distance may affect the simulation results of the functional connectivity, we introduced the exponential distance rule (EDR) in the model to account for the distance effect (Fig. 7D), where the brain connection followed an exponential distribution of the projection length^{38, 39, 40}. Due to its inherent limitation for tracking long connections, the diffusion tractography were more affected by the distance for the modeling, where longer connections had lower FC fitting values (Fig. 7D). On the contrary, the modeling from neural tracing data was not affected by the distance, which suggested that the tracing data might be a more reliable bridge to link the structural and functional connections. Importantly, as shown in Fig. 7E-F, the EDR-constrained modeling based on MBMv4 fits the empirical functional data better than the Paxinos atlas (Wilcoxon paired signed-rank test: N=27, p=0.01947), similar as the modeling without EDR (Fig. 7B-C). Thus, MBMv4 preserves a crucial bridge for examining the structural and functional connectivity discrepancy.

As for asymmetry of the marmoset brain, reviewer described in the prior revision that it may be to a much smaller extent than in humans. However, the data in the revised manuscript does not address this issue. While number of parcellations was the same between hemispheres, there is likely left vs right differences in the cortical boundary maps in Fig 3. There is possibility that the areal size of the parcels may be different between hemispheres, but it was not evaluated in this revision. This can be achieved by invert warping the standard average surfaces to the subject's native space and calculating the

surface areas of each parcel in subject's native space. The authors may want to make it clear if asymmetry is really smaller than in humans.

Thanks for the helpful suggestion. As the reviewer suggested, we registered the MBMv4 atlas into every subject's native space and calculated the size of each parcel. Then, a paired two-sample t-test was performed between the left and the right for each parcel across subjects, and we found no significant differences between hemispheres (see the below figure and supplementary figure S9-C).

In addition, the issue regarding asymmetry had been investigated previously (*James K, et al, 1999*), which specifically pointed out that the brains become more asymmetrical as they grow in size during evolution:

James K, et al, Differential expansion of neural projection systems in primate brain evolution. Neuroreport. 1999 May 14; 10(7):1453-9.

In the revision, we cited this influential paper in the brain evolution literature and made it clear regarding the asymmetry:

(Line 483-485)

“Therefore, our results may reflect that the asymmetry in marmosets may be smaller than in humans, as expected from previous analyses based on anatomical measurements (James K, et al, 1999) and the evidence that the number of cortex subdivisions increases with brain volume (Changizi MA et al., 2005).”

James K, et al, Differential expansion of neural projection systems in primate brain evolution. Neuroreport. 1999 May 14; 10(7):1453-9.

Changizi MA, Shimojo S. Parcellation and area-area connectivity as a function of neocortex size. Brain Behav Evol 66, 88-98 (2005).

Reviewers' comments:

Reviewer #1 (Remarks to the Author):

The reviewer appreciates the authors' great efforts in improving the manuscript. Overall, data acquisition of fMRI is in high quality and higher-level analysis is sophisticated (e.g. deep-learning parcellations, FC simulations). Registration of subject's T2w volume to template was very nice. The reviewer encourages the authors to make further improvements as follows:

1. The FIX-ICA was initially developed for resting-state MRI data in humans but can be used in any species. Indeed, it is also used in rodents (Zerbi et al 2015; Diao et al. 2021). The first-level ICA results shared with the reviewer in this version of rebuttal indicates that ICA is working well thus FIX-ICA should also work if the authors trained the classification of the components. The reviewer identified N=12 components (1, 6, 7, 9, 11, 12, 13, 15, 16, 21, 22, 28) as neural, comprising 5.8% of total variance, and the rest of components (N=46) as structural artifacts (19.1% of total variance) and random noise (~75% of total variance), which means ~94% of total variance in the data is likely of uninterest. Training of the ICA components classifier on users' own data is in general recommended (see usage of ICA-FIX), and if it was made publicly available, those who use animal MRI scanners in marmoset neuroscience may benefit greatly from it. Also, the fMRI input data has significant left vs right asymmetry of signals probably due to expansion and shrinkage of B0-distortion, which may also cause some issues when combining across scans for gICA but can be reduced by taking account into Jacobian when running topup.
2. It is better to run group ICA (gICA) using cleaned fMRI datasets. The authors shared 'raw 15 cortical network' components (atlas_MBMv4_networks_raw.nii.gz). Among the '15-cortical network', components 3, 4, 5 and 14 are likely structural artifacts (3: motion/registration error/bias, 4: imaging artifacts or registration error, 5: CSF, 14: CSF) and others are questionable or mixture of signal and noise (11: CSF and visual). Therefore, it is recommended to use uncleaned fMRI data to feed into gICA and to detail neurobiological and fair interpretations of each signal and artifactual components, which will have a significant impact in this field. It is also recommended to share the sub-cortical and artifactual components of gICA, as well as time signal, frequency spectrum data for all the components, since the classifying the components is not easy without having these data. The authors should consistently label (or number) all the ICA components including signals and noises in figures and data shared. It is not clear which dimensionality was actually applied in the final results (20?), although the authors applied "from 20 to 80 in step of 10" as written in the method section.
3. As for the optimal settings of gICA (variance normalization, remove per time point etc.). only spatial maps are shown but no data for time series and spectrum. Thus, it is difficult to see which is likely the optimal.

Minor points

It is not clear how CNR was calculated. Please describe it in method section.

It is not clear why “adding the exponential distance rule to diffusion tractography should make more accurate estimation of structural connectivity”. Please elaborate it more in method section.

In Fig. S5, the figure legend states that “We found 19 networks by group-ICA analysis”, but only 18 are presented in the figure. There are no labels that can be identified in the ICA components data shared. As suggested above, four are likely artifacts among ‘15 cortical-network’ of ICA.

Line 605, “Finally, our parcellation only used the resting-state functional connectivity information, as in many human studies^{22, 29, 45.}” - This is not correct. Glasser et al., (ref 29) used myelin maps, thickness, task fMRI and resting-state fMRI for parcellating human cerebral cortex.

Data resource: It is not clear how CIFTI data was created, how symmetrization of the hemisphere, what surface registration was applied, and what number of meshes was used. There is no description on the label or number of ICA components that are consistent between the data and figure. No data for gICA components of sub-cortical structures and structural artifacts are shared.

Response letter to Reviewer

In what follows, we provide point-by-point responses to reviewer#1's concerns. The review comments are in black font, and our responses are in blue. The resulting edits of the original manuscript are highlighted in red font in the revised manuscript.

The Reviewer appreciates the authors' great efforts in improving the manuscript. Overall, data acquisition of fMRI is in high quality and higher-level analysis is sophisticated (e.g. deep-learning parcellations, FC simulations). Registration of subject's T2w volume to template was very nice. The Reviewer encourages the authors to make further improvements as follows:

1. The FIX-ICA was initially developed for resting-state MRI data in humans but can be used in any species. Indeed, it is also used in rodents (Zerbi et al 2015; Diao et al. 2021). The first-level ICA results shared with the Reviewer in this version of rebuttal indicates that ICA is working well thus FIX-ICA should also work if the authors trained the classification of the components. The Reviewer identified N=12 components (1, 6, 7, 9, 11, 12, 13, 15, 16, 21, 22, 28) as neural, comprising 5.8% of total variance, and the rest of components (N=46) as structural artifacts (19.1% of total variance) and random noise (~75% of total variance), which means ~94% of total variance in the data is likely of uninterest. Training of the ICA components classifier on users' own data is in general recommended (see usage of ICA-FIX), and if it was made publicly available, those who use animal MRI scanners in marmoset neuroscience may benefit greatly from it.

We created and shared the ICA-FIX version to meet the Reviewer's request

For the ICA-FIX, we followed the Reviewer's request to provide the version of ICA-FIX preprocessed data (and the associated manually training datasets and training-weighted files), which will be released online as a supplementary resource of our MBMv4 upon the publication of the manuscript. In addition, for review purposes, we uploaded the gICA results with/without ICA-FIX cleaning to facilitate the Reviewer for the comparison (both under the setting of ICA=30).

To create the ICA-FIX version of data, we ran first-level ICA on each fMRI run, randomly selected 24 runs from 24 animals (12 runs from the ION dataset and 12 runs from the NIH dataset), and manually classified the noise components based on spatial patterns and the power spectrum. Note that the ICA-FIX recommended a training dataset of at least 10; here, we made 24 training datasets to improve our ICA-FIX classifier significantly, which means that the trained-weights file (trainingMBMv4.RData) is comprised of two trained-weights files from the ION dataset (trainingION.RData, using 12 ION training datasets) and the NIH (trainingNIH.RData, using 12 NIH training datasets) respectively. Then, the trained-weighted file (trainingMBMv4.RData) was used to clean all fMRI data using three different sensible-value thresholds (5, 10, and 20; the ICA-FIX recommended a threshold in the range of 5-20, so for a fair comparison, we select three thresholds) and thus created three different versions of ICA-FIX cleaned datasets. In addition, we provided the mask files to allow the ICA-FIX to work on the marmoset data since the ICA-FIX has several steps that use human-default settings and files, which are incompatible with the marmoset data. The ICA-FIX data can be downloaded via (https://marmosetbrainmapping.org/data_for_MBMv4_reviews.html) and will be publically accessed after the paper's publication.

It is worth noting that, due to the inherent limitation of the single-run ICA on our data, the ICA failed to decompose the unsmoothed data (and data with 0.5mm FWHM smoothing from our animal MRI imaging resolution) into enough components for us to classify noise components. We also uploaded an example of a single-run ICA on unsmoothed data to prove this point as below:

(https://marmosetbrainmapping.org/data_for_MBMv4_reviews.html), which showed that the single-run ICA only decomposed the data into 6 components with a mixture of single and noises. Thus, all the ICA-FIX here we provide requires data with 1mm FWHM smoothing (which was also for the example we shared with the Reviewer in the previous round of revision).

Nevertheless, we have successfully applied ICA-FIX in our dataset and **still obtained similar results as the report without ICA-FIX**. For example, the following figure directly demonstrates these components that the Reviewer regarded as artifacts because we did not use ICA-FIX can still be robustly detected and high consistent with our result without ICA-FIX reporting in our paper.

With ICA-FIX

Without ICA-FIX (in our paper)

Component D
frontal pole

Component E
orbital frontal cortex

Component F
the parahippocampus

Component H
Salient-related network

Also, the fMRI input data has significant left vs right asymmetry of signals probably due to expansion and shrinkage of B0-distortion, which may also cause some issues when combining across scans for gICA but can be reduced by taking account into Jacobian when running topup.

The distortion issue was answered in the previous round.

We kindly refer the editor to our answer in the first round (see below) for the question of distortion, which has already demonstrated a good spatial correction.

428 **OUR RESPONSE:** The parameter file used for EPI distortion correction are modified
429 according to the marmoset brain size, spatial resolution and dwell time to achieve the best
430 correction for the marmoset brain. We provided the topup.conf file in the revision so that
431 users can adopted the config file to re-run and test the topup. Meanwhile, in the resource,
432 we provided the raw data and the preprocessed data with topup correction, so that the user
433 can directly evaluate the outcome of the EPI distortion correction. We also added a
434 supplementary Figure S10 to show the data before v.s. after EPI distortion correction.

2. It is better to run group ICA (gICA) using cleaned fMRI datasets. The authors shared 'raw 15 cortical network' components (atlas_MBMv4_networks_raw.nii.gz). Among the '15-cortical network', components 3, 4, 5 and 14 are likely structural artifacts (3: motion/registration error/bias, 4: imaging artifacts or registration error, 5: CSF, 14: CSF) and others are questionable or mixture of signal and noise (11: CSF and visual). Therefore, it is recommended to use uncleaned fMRI data to feed into gICA and to detail neurobiological and fair interpretations of each signal and artifactual components, which will have a significant impact in this field. It is also recommended to share the sub-cortical and artifactual components of gICA, as well as time signal, frequency spectrum data for all the components, since the classifying the components is not easy without having these data. The authors should consistently label (or number) all the ICA components including signals and noises in figures and data shared. It is not clear which dimensionality was actually applied in the final results (20?), although the authors applied "from 20 to 80 in step of 10" as written in the method section.

We must say that the Reviewer's question confuses us a lot since it has internally conflicting suggestions. Initially, he/she suggests that, "It is better to run group ICA (gICA) using **cleaned** fMRI datasets." Later, he/she indicates that "Therefore, it is recommended to use **uncleaned** fMRI data to feed into gICA and to detail neurobiological and fair interpretations of each signal and artifactual components, which will have a significant impact in this field." Still, we have run all of these analyses back and forth for several rounds of the revisions, and these components, which the Reviewer considers artifacts, can be detected robustly no matter which method is adopted.

In the first round of revisions, it was requested from us to run gICA on uncleaned data preprocessing with/without bandpassing. These components are robust (as below).

304 We believed these components are neural-like signals, as these components were located
305 in cortical gray matter regions and showed bilateral patterns. To further verify the possibility
306 of contamination of very low-frequency noise, we also purposely re-performed the ICA on
307 data with a high-pass filter ($>0.01\text{Hz}$). Below are two examples of neural components, D
308 (the “questionable” component) and B (the “promising” component), from the data with a
309 high-pass filter ($>0.01\text{Hz}$).

310
311 Below are the same examples of neural components D and B from the data without a high-
312 pass filter (from raw figures S6).

313
314 Since the power peaks of these components are all above 0.01Hz , it is hard to say the
315 contamination of very low-frequency noises. We still detect these four components after a
316 high-pass filter ($>0.01\text{Hz}$). Therefore, we have already tried to take the strict rule combining
317 spectrum and spatial patterns to determine whether a component is neural or noise. As a
318 result, our identified components should be reliable.

In the second round of revisions, it was requested from us to run gICA on uncleaned fMRI data preprocessing under different normalizations. Again, these components are robust (as below).

340 For the data normalizing, we also tested what the reviewer suggested, as well as other
341 options of GIFTI, including "Remove Mean Per Timepoint - At each time point, image mean
342 is removed.", "Remove Mean Per Voxel - Time-series mean is removed at each voxel",
343 "Intensity Normalization - At each voxel, time-series is scaled to have a mean of 100." or
344 "Variance Normalization - At each voxel, time-series is linearly detrended and converted to
345 Z-scores (with mean brain signal scaled to 100)".`

346

347 Despite preprocessing choices, we could obtain similar components described in the paper.
348 We demonstrated four components (one local network B, one global network J, and two
349 questionable networks D and E suggested by the reviewers) as examples here. The
350 "Variance Normalization" has similar results as the "Remove Mean Per Voxel" and
351 "Remove Mean Per Timepoint".

352

353

Now, in the third round of revision, the reviewer asks us to run ICA-FIX because we did not use this denoise method. Following the request, we created the ICA-FIX version of preprocessed data and ran the gICA on data with the ICA-FIX cleaning dataset. Again, these components are robust (as below). As mentioned above, we uploaded the raw ICA results (with 30 components setting) with/without ICA-FIX for the comparison by the Reviewer (https://marmosetbrainmapping.org/data_for_MBMv4_reviews.html).

At this point, it is fair to conclude that our detected components are not artifacts, as the Reviewer assumed. Besides that, we also provide further evidence, as detailed below.

1) The components are located in meaningful cortical areas, given present knowledge of primate brain anatomy and physiology: They are located in gray matter regions with clear anatomical structures, which are unaffected by imaging distortion. Also, their spatial ICA patterns matched their connectivity patterns.e.g. the frontal-pole area had a low correlation with any distant area (reference Liu et al., 2019). If these components were removed as the Reviewer thinks, a large portion of the cortical regions (highlighted in red overlay) would not have a functional network (see below):

References:

Liu C, et al. Anatomical and functional investigation of the marmoset default mode network. Nature Communications. 2019, 10(1), 1-8.

2) Reported and replicated in the previous paper: The components have been reported in previous papers including ICA analysis on marmoset data. For example, similar components of the frontal pole, the orbitofrontal component, and the cerebellum have been reported in awake marmosets (Belcher, et al, 2013), and a similar orbitofrontal component has been reported in anesthetized marmoset (Ghahremani, et al, 2016).

References:

Belcher AM, et al. Large-scale brain networks in the awake, truly resting marmoset monkey. *J. Neurosci.* 2013;33:16796–16804. doi: 10.1523/JNEUROSCI.3146-13.2013.

Ghahremani M, Hutchison RM, Menon RS, Everling S. Frontoparietal Functional Connectivity in the Common Marmoset. *Cereb Cortex.* 2017;27(8):3890-3905.

3) Statistical thresholds required for ICA: The Reviewer argued that noise and artifacts contaminated these components. However, since any ICA method cannot 100% separate signals from noises, our ICA software implemented a stringent statistical threshold by default to show the most significant parts of the components. The results with or without ICA-FIX provide a good example. Although ICA-FIX denoising reduced noise level, the spatial pattern of the group ICA results is virtually identical to that obtained without ICA-FIX denoising. After the statistical thresholds, these components were all in the gray matter, and CSF or artifact parts were removed

from our final network parcellations.

3. As for the optimal settings of gICA (variance normalization, remove per time point etc.), only spatial maps are shown but no data for time series and spectrum. Thus, it is difficult to see which is likely the optimal.

We must kindly remind the Reviewer that the different normalizations for the data (variance normalization, removal per time point, etc.) could not change the frequency property of the data because the data did not undertake any frequency filter. So the components (under different normalization) with similar spatial patterns should result in a similar power spectrum and time series after ICA decomposing. Thus, we have already provided the default normalization power spectrum in the previous response letter and supplementary data (as below). They should be similar.

"It is not clear which dimensionality was actually applied in the final results (20?), although the authors applied "from 20 to 80 in step of 10" as written in the method section."

Different dimensionality should result in similar patterns of components, although some components might be separated into two or more at high dimensionality. Therefore, the question becomes which dimensionality is the better presentation for a network. For us, the final network parcellation was based on the ICA results of the 20 and 30. We also did manual correction before creating the final network parcellation; for example, when the left and right parts of the same network were separated into two components, we would merge them into one network by averaging. We added the above details in the revised methods (Line 755-770).

Minor points

It is not clear how CNR was calculated. Please describe it in method section.

Following the Reviewer's suggestion in the last round that CNR should be calculated as the variance of the resting state fMRI component after ICA, which are optimal contrasts of interests (see the Reviewer's previous common below),

Interestingly, CNR of functional MRI used contrast of gray/white signals, which is not a good indicator of the quality of the functional MRI. Grey/white contrast is widely used for structural MRI, because the structural images are commonly used for registration, segmentation, and surface estimation based on the contrast of gray/white matter. In case of fMRI, it is common to have interests in the blood-oxygen dependent 'neural' signals, e.g., beta of 'neural' activity in the task fMRI statistical model or the variance of the resting state fMRI component are optimal contrasts of interests for estimating CNR of fMRI.

We selected the fMRI ICA components which the review thinks are "optimal" (see the Reviewer's suggestions in the last round, where he/she indicates that components B, I, J, K are optimal). Then, we calculated the variance of these components in contrast with the noise component for each dataset (ION and NIH) and compared their similarity.

clearly show the 'neural like' features and likely have contamination of very low frequency noise. In particular, D, E, F, R are questionable if they are really neural, while B, I, J, K seem most promising neural components. With so much thresholding, it is hard to know what is going on in most of areas.

We have added this description to our figure legend and to our Methods section (L737-748, see below).

We calculated a series of indices to test the data harmonization across different datasets (NIH and ION). They include the Single time points, mean images (average across time for one fMRI run), temporal Signal Noise Ratio (tSNR, from one fMRI run), contrast to Noise Ratio (CNR, the mean of the gray matter intensity values minus the mean of the white matter intensity values divided by the standard deviation of the values outside the brain), temporal contrast to Noise Ratio (tCNR, the variance of optimal resting-state fMRI components after ICA contrast to the noise), the Fiber (Foreground to Background Energy Ratio: the variance of voxels inside the brain divided by the variance of voxels outside the brain), head motion and the whole-brain functional connectivity across subjects and sessions.

It is not clear why "adding the exponential distance rule to diffusion tractography should make more accurate estimation of structural connectivity". Please elaborate it more in method section.

We have already elaborated on the procedure in our Methods section (L1037-1053) in the previous round

1037 The resulting simulation of functional connectivity demonstrated the influence of the
 1038 distance, especially for the diffusion MRI (see Fig. 7D). To decrease this impact, we
 1039 added an Exponential Distance Rule (EDR) for a more accurate estimation of structural
 1040 connectivity before the simulation, which can be implemented as follows: we first
 1041 normalized the structural connectivity estimated by diffusion tractography to 0-1, then
 1042 calculated the probability of structural connectivity according to the EDR rule ($p(d)=ce^{-\lambda d}$,
 1043 $\lambda \approx 0.3$, $c \approx 0.94$ where d is the distance ⁴⁰) and normalized to 0-1. Finally, we transformed
 1044 the normalized structural connectivity to match the probability of structural connectivity.
 1045 For simplicity to help us to identify abnormal values, we defined a threshold as the Median
 1046 Absolute Deviation for the normalized structural connectivity.

1047

$$1048 \quad \hat{\sigma} = \frac{\text{Median}(|X_i - \bar{X}|)}{0.6745}$$

40

1049 Where X_i = each value, \bar{X} = average value

1050

1051 If any values were larger than this threshold, we considered them abnormal values, and
 1052 the corresponding probability of structural connectivity would replace them to reduce the
 1053 impact of distance.

1054

So the procedure can be summarized as below:

- 1) Normalization of structural connectivity matrix estimated by diffusion MRI into 0-1.
- 2) Calculate another matrix of structural connectivity probability in the same size according to the EDR rule since we have already known the surface distance between regions and normalized it into 0-1.
- 3) Through the threshold calculated by the Median Absolute Deviation of the normalized structural connectivity matrix, we replace the abnormal values from the normalized structural connectivity matrix with the values from the structural connectivity probability.

In Fig. S5, the figure legend states that "We found 19 networks by group-ICA analysis", but only 18 are presented in the figure. There are no labels that can be identified in the ICA components data shared. As suggested above, four are likely artifacts among '15 cortical-network' of ICA.

Since our paper focus on cortical connectivity, we removed all the subcortical networks from the supplementary data to avoid confusion in the revision and reported the 15 cortical networks only. Again, in response to the previous questions, these components are not artifacts but significant biological components reported in the

previous marmoset ICA studies.

Line 605, "Finally, our parcellation only used the resting-state functional connectivity information, as in many human studies^{22, 29, 45.}" - This is not correct. Glasser et al., (ref 29) used myelin maps, thickness, task fMRI and resting-state fMRI for parcellating human cerebral cortex.

We corrected the description of the citation in the revised manuscript.

Data resource: It is not clear how CIFTI data was created, how symmetrization of the hemisphere, what surface registration was applied, and what number of meshes was used. There is no description on the label or number of ICA components that are consistent between the data and figure. No data for gICA components of sub-cortical structures and structural artifacts are shared.

The surface data were created directly using `wb_command`, similar to what the HCP data did. First, We have a description for this part in Methods 665-694 of the previous revision, and in the current revision, we added more details to the manuscript (Lines 668-690). Second, we had already answered the question twice regarding the surface registration in the previous response letter (first round), in which we used volume registration to map all data to the MBMv3 template and then mapped all data to the MBMv3 surface (see below). Third, the details of how MBMv3 surface creation have been detailed described in the published article (Liu, et al, 2021), with a total of 76068 meshes for the cortex in the CIFTI data and the subcortical data can be directly distinguished from the cortical data by the connectome workbench. Thus, we kindly refer the Reviewer to the original research articles.

Reference: C Liu, CCC Yen, D Szczupak, X Tian, D Glen, AC Silva, Marmoset Brain Mapping V3: Population multi-modal standard volumetric and surface-based templates, Neuroimage 226, 117620, 2021

OUR RESPONSE: Thank you for your suggestions. Individual surface reconstruction and surface-based registration were not performed, for the reasons described in our response to a previous comment about DCBC.

This study uses the Marmoset Brain Mapping V3 (MBMv3) template for volume-based registration and the Connectome Workbench (`wb_command -volume-to-surface-mapping` function and ribbon constrained mapping algorithm) for the volume-to-surface mapping. The MBMv3 has a population-based T2-weight template image and brain surfaces. First, the subject fMRI was co-registered to the subject T2w image, and the subject T2w was registered to the T2w template of MBMv3. Then, the transformations were concatenated to normalize the subject fMRI data to the MBMv3 template space. Finally, the normalized subject fMRI data were mapped to the MBMv3 brain surface.

OUR RESPONSE: Thank you for your suggestions. Individual surface reconstruction and surface-based registration were not performed, for the reasons described in our response to a previous comment about DCBC.

This study uses the Marmoset Brain Mapping V3 (MBMv3) template for volume-based registration and the Connectome Workbench (wb_command -volume-to-surface-mapping function and ribbon constrained mapping algorithm) for the volume-to-surface mapping. The MBMv3 has a population-based T2-weight template image and brain surfaces. First, the subject fMRI was co-registered to the subject T2w image, and the subject T2w was registered to the T2w template of MBMv3. Then, the transformations were concatenated to normalize the subject fMRI data to the MBMv3 template space. Finally, the normalized subject fMRI data were mapped to the MBMv3 brain surface.

Furthermore, all the component files (volume, surface, etc.) in the resource use the same number between the label and the labeled files to describe which component is which.

Our paper is all about the "cortical" ICA parcellation and connectivity. Thus we showed only cortical ICA results in the manuscript, and the resource provided the raw ICA maps for the cortical network parcellation. We also removed the sub-cortical components from the supplementary data to avoid confusion. The raw/preprocessed data have been all released to allow replication of our results.

Again, for the Reviewer's interests in structural artifacts, we uploaded the raw group-ICA results of a setting of 30 components, with and without ICA-FIX denoising (https://marmosetbrainmapping.org/data_for_MBMv4_reviews.html). This shows the structural artifacts and the components/timecourses/spectrum, which can be directly viewed via GIFT software (<https://trendscenter.org/software/gift>).